# LLMs Lean on Priors, Not Programming Language Semantics

**Aditya Thimmaiah** [1]  **Jiyang Zhang** [1]  **Jayanth Srinivasa** [2]  **Junyi Jessy Li** [1]  **Milos Gligoric** [1]

## Abstract

Recent work asks whether large language models (LLMs) condition their reasoning on explicit rules rather than statistical regularities from pretraining. Program execution provides a canonical instance: formal semantics define behavior through symbolic transition rules that can be systematically altered under distribution shift. We investigate whether LLMs can condition their reasoning on formal semantics through program execution and introduce PLSEMANTICSBENCH, pairing featherweight C programs with two semantic systems—small-step operational semantics and K semantics—and probing four capabilities: composing rules for final states, selecting rules when state is unmutated, sustaining such conditioning over long traces, and following supplied rules under novel semantics. To decouple semantic reasoning from syntactic familiarity, we redefine familiar operators to induce symbol-meaning conflict and introduce novel symbols defined only through the supplied rules, and stress-test models on Human-Written, LLM-Translated, and Fuzzer-Generated splits with increasing structural complexity. Across 11 frontier LLMs, strong final-state accuracy under standard semantics (up to 90%) drops sharply—by as much as 40–60% points—under semantic mutations and increasing structural complexity. Only a handful of models achieve non-zero long-horizon conditioning accuracy, and even the best systems reach just 35%. Together, these results suggest that contemporary LLMs often rely on pretrained lexical associations rather than systematically conditioning on supplied formal rules. PLSEMANTICSBENCH is publicly available at https://EngineeringS oftware.github.io/PLSemanticsBench.

## 1. Introduction

Modern large language models (LLMs) increasingly solve programming tasks that appear to require reasoning about program behavior, from predicting outputs (Gu et al., 2024; Ni et al., 2024) to repairing and generating (Huang et al., 2025) code. This raises a natural question: do such models rely primarily on statistical regularities acquired during pretraining, or can they flexibly condition their reasoning on explicitly provided behavioral rules?

Consider an integer arithmetic operation with two alternative semantics for the standard '+' symbol, a scenario frequently encountered in operator overloading (Ravasi & Vasconcelos, 2020; Triton, 2026). Under the first, `2+2` behaves conventionally as *addition*; under the second, the same symbol is defined to perform *subtraction*. A system that reasons from syntax alone (learned priors) would produce the same answer in both cases. A system that conditions on the supplied semantic definitions would change its behavior immediately. This contrast captures a broader scientific issue:

*Can LLMs adapt their reasoning to externally specified formal systems, even when those systems conflict with entrenched priors learned from data?*

Formal semantics (Pierce, 2002) offers a uniquely controlled setting for studying this question. The semantics of a programming language consist of symbolic transition rules governing program-state evolution. Correct execution requires repeatedly selecting and composing such rules over many steps. Crucially, these rules can be modified without altering surface syntax, allowing one to separate reliance on lexical cues from genuine conditioning on semantics. Furthermore, formal semantic rules: (1) are atomic with uniform granularity, enabling systematic comparison across programs and model predictions, and (2) specify behavior mathematically rather than in natural language, reducing ambiguity between intended execution and model instructions.

**Tested hypotheses**. We use program execution as a lens for analyzing *formal semantic rule-conditioned reasoning in LLMs*. Rather than asking whether models can execute programs in familiar languages, we ask whether they can—*alter* their reasoning under novel formal semantics, apply individual rules at fine granularity, and sustain such conditioning across long execution horizons (e.g., loops and nested control flow). This yields four concrete hypotheses

---

[1]The University of Texas at Austin [2]Cisco Research. Correspondence to: Aditya Thimmaiah <auditt@utexas.edu>.

*Proceedings of the 43rd International Conference on Machine Learning*, Seoul, South Korea. PMLR 306, 2026. Copyright 2026 by the author(s).

about model capabilities:

★ **H1** (Global Rule Conditioning): Models can combine many rule applications to correctly predict final states.

★ **H2** (State-Free Rule Conditioning): Models can follow rules correctly under state-mutation free execution.

★ **H3** (Long-Horizon Rule Conditioning): Models can follow formal rules consistently across long execution traces.

★ **H4** (Rule Conditioning Under Semantic Shift): Models continue to follow supplied rules under novel semantics.

To test these hypotheses, we introduce PLSEMANTICSBENCH, which pairs a featherweight (Harper, 2016) C programming language C$^\star$ with two formal semantic systems—the fine-grained small-step structural operational semantics ($\mathbb{S}$) and the coarser rewriting-based $\mathbb{K}$ semantics (Roșu & Șerbănută, 2010). The benchmark probes the hypotheses via three complementary tasks—predicting final program states (**PredState**), selecting semantic rules governing execution in absence of state mutation (**PredRule**), and generating full execution traces to probe long-horizon rule application (**PredTrace**)—while using semantic mutations and program-complexity splits as stressors for robustness.

**Reliance on learned priors vs supplied rules**. We disentangle reliance on supplied rules versus learned priors along two orthogonal axes: semantic mutation and program-complexity shifts. A key feature enabled by formal semantics is *nonstandard* variants that systematically perturb symbol meanings. In KeywordSwap, common operators exchange their behavior, creating direct conflicts with pretrained priors. In KeywordObf, familiar syntax is replaced with novel symbols whose meanings are defined only through the supplied rules. Models are additionally evaluated on human-written, LLM-translated, and fuzzer-generated programs with varied structural complexity, stressing deep control flow and unusual data-flow patterns.

**Choice of programming language**. We use C$^\star$ rather than indentation-sensitive languages such as Python, whose concrete syntax requires recovering block structure (Adams, 2013) from layout before abstract syntax can be constructed, thereby entangling syntactic recovery with semantic reasoning. Explicit block delimiters '{ }' in C$^\star$ avoid this confound, allowing us to isolate the model's ability to condition on formal semantics defined over the abstract syntax.

Our experiments across a broad set of frontier and open-weight models show that while several benefit from access to formal rules under standard semantics, performance deteriorates sharply under semantic mutations, increased rule granularity, and long execution horizons, exposing systematic limits in current models' ability to sustain reasoning conditioned on externally specified formal systems.

By framing program execution as a controlled probe of rule-conditioned reasoning, we provide a semantics-driven

```
1  <program>    ::= <stmt>
2  <stmt>       ::= <assgn_stmt>
3  <assgn_stmt> ::= <id>'='<exp>';'
4  <exp>        ::= <literal> | <exp>'+'<literal>
5  <literal>    ::= <digit> | <literal> <digit>
6  <id>         ::= <letter> | <id> <letter>
7  <digit>      ::= '0' | ... | '9'
8  <letter>     ::= 'a' | ... | 'z'
```

*(a)* Syntax in Backus-Naur Form (McCracken & Reilly, 2003).

$$\text{S-ASSIGNSTEP} \quad \frac{\langle e,\sigma\rangle \to_E \langle e',\sigma\rangle}{\langle x=e,\sigma\rangle \to_S \langle x=e',\sigma\rangle}$$

$$\text{S-ASSIGN} \quad \frac{\sigma' = \sigma[x \mapsto v]}{\langle x=v,\sigma\rangle \to_S \langle \epsilon,\sigma'\rangle}$$

$$\text{E-ADD} \quad \frac{v_3 = v_1 + v_2}{\langle v_1+v_2,\sigma\rangle \to_E v_3}$$

$$\text{E-ADDLEFTSTEP} \quad \frac{\langle e_1,\sigma\rangle \to_E \langle e_1',\sigma\rangle}{\langle e_1+e_2,\sigma\rangle \to_E \langle e_1'+e_2,\sigma\rangle}$$

$$\text{E-ADDRIGHTSTEP} \quad \frac{\langle e_2,\sigma\rangle \to_E \langle e_2',\sigma\rangle}{\langle v_1+e_2,\sigma\rangle \to_E \langle v_1+e_2',\sigma\rangle}$$

$$\text{E-ID} \quad \frac{\sigma(x) = v}{\langle x,\sigma\rangle \to_E v}$$

*(b)* Semantics in Small-step operational semantics (Plotkin, 2004).

*Figure 1.* Formal syntax and semantics of an example language $\mathcal{L}$.

benchmark for assessing when LLMs adapt their behavior to externally specified formal systems.

## 2. Background

The semantics of a programming language defines program behavior. Structural operational semantics specifies semantics via inference rules that govern transitions between *configurations*, each pairing a program fragment with its execution state. We use small-step semantics ($\mathbb{S}$), where each rule represents one atomic computation and execution arises from repeated rule applications. Rules are written in Gentzen-style inference notation (Gentzen, 1964), with premises and side conditions above the fraction bar and conclusions below.

We illustrate semantics formalization in $\mathbb{S}$ using a simple imperative language $\mathcal{L}$ whose syntax (Figure 1a) includes assignments and integer expressions with addition. Table 1 summarizes the notation used in its formalization (Figure 1b). Configurations take the form $\langle c,\sigma\rangle$, where $c$ ranges over statements

*Table 1.* Notation primer.

| Notation | Definition |
|---|---|
| $\sigma$ | Program state |
| $s$ | Statement |
| $x$ | Int variable |
| $e$ | Int expression |
| $v$ | Int literal |
| $\langle operation, \sigma\rangle$ | Configuration |
| $\sigma[x \mapsto v]$ | Store $v$ in $x$ |
| $\langle e,\sigma\rangle \to_E \langle e',\sigma\rangle$ | Expression-step |
| $\langle s,\sigma\rangle \to_S \langle s',\sigma'\rangle$ | Statement-step |
| $\langle \epsilon,\sigma\rangle$ | NOP |

($s$) and expressions ($e$), and the state maps variables ($x$) to integer values ($v$).

Expression transitions $\langle e,\sigma\rangle \xrightarrow{r}_E \langle e',\sigma\rangle$ apply a single rule to reduce an expression and record the ordered list of rules $r$ used so far; they do not mutate state and terminate in a literal under transitive–reflexive closure. Statement transitions $\langle s,\sigma\rangle \xrightarrow{r}_S \langle s',\sigma'\rangle$ may update the state and iterate until reaching NOP ($\langle \epsilon,\sigma\rangle$). For example, S-ASSIGNSTEP propagates evaluation through an assignment by stepping the right-hand-side expression, while S-ASSIGN applies once that expression reduces to a literal and commits the value to the state; together, such rules illustrate how programs execute by repeated configuration transitions.

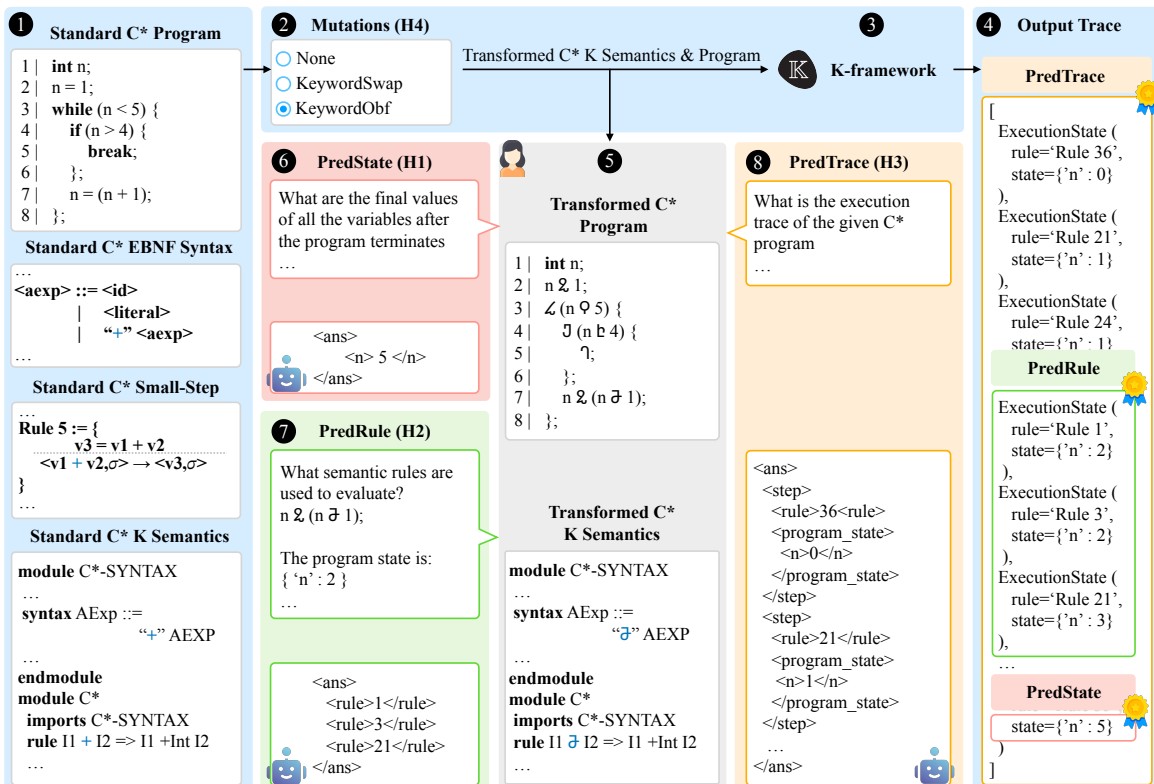

*Figure 2.* Overview of the PLSEMANTICSBENCH construction workflow and evaluation tasks designed to probe rule-conditioned reasoning under formal semantics. Each program is written in C* with EBNF syntax and paired with two semantic systems—small-step operational semantics (S) and K semantics (K) (①). The standard semantics are systematically transformed into two nonstandard variants, KeywordSwap and KeywordObf, which preserve rule structure while perturbing the symbol–meaning mapping (②). These semantics are used to derive ground-truth execution traces for each transformed program (③-④). The transformed program and its semantics (⑤) are then provided to models as prompts. Tasks (⑥-⑧) range from predicting final states (**PredState**; **H1**), to selecting applicable rules when state does not mutate (**PredRule**; **H2**), to generating full execution traces to test long-horizon rule conditioning (**PredTrace**; **H3**). An analogous pipeline is used for S by replacing the execution engine, enabling controlled comparisons across semantic formalisms and shifts (**H4**).

**Definition 2.1** (Statement Execution). Let $s$ be a statement derived from a given grammar $G$ that is semantically valid under an S formalization $\Psi$, and let $\mathcal{R}$ be the set of all rule names in $\Psi$. Suppose that under $\Psi$ and an initial program state $\sigma_0$, the statement $s$ reduces to a NOP configuration in $n$ statement-steps: $\langle s, \sigma_0 \rangle \xrightarrow{r_1}_S \langle s', \sigma_1 \rangle \xrightarrow{r_2}_S \ldots \xrightarrow{r_n}_S \langle \epsilon, \sigma_n \rangle$. For each $i \in [1, n]$, $r_i \in \mathcal{R}^*$ denotes the ordered list of rules required for the $i^{th}$ statement-step, with elements indexed as $r_{i,j}$ where $j \in [0, |r_i|)$, and $\sigma_i$ denotes the program state after the $i^{th}$-step. Let '$\oplus$' denote the standard list concatenation operator. We then define statement execution of $s$ under $\Psi$ and an initial program state $\sigma_0$ as a pair:

$$\llbracket s \rrbracket_\Psi(\sigma_0) \triangleq (\ \underbrace{\sigma_n}_{\text{resulting state}}, \ \underbrace{\bigoplus_{i=1}^{n} \bigoplus_{j=0}^{|r_i|-1} [(\sigma_i, r_{i,j})]}_{\text{execution trace}}), \ r_{i,j} \in \mathcal{R}$$

**Definition 2.2** (Program Execution). Let $\mathcal{P}$ be a program derived from a given grammar $G$ that can be parsed into an ordered list of statements $[s_0, \ldots, s_n]$. We define program

execution of $\mathcal{P}$ under a given formalization $\Psi$ and an initial program state $\sigma_0$ compositionally using Definition 2.1 as:

$$\llbracket \mathcal{P} \rrbracket_\Psi(\sigma_0) = \llbracket [s_0, \ldots, s_n] \rrbracket_\Psi(\sigma_0) \triangleq (\llbracket s_0 \rrbracket_\Psi \otimes \ldots \otimes \llbracket s_n \rrbracket_\Psi)(\sigma_0)^{\dagger\ddagger}$$

Here, '$\otimes$' is a left-associative sequencing operator defined by direct style *Kleisli composition* (Wadler, 1995) as follows:

$$(\llbracket s_A \rrbracket_\Psi \otimes \llbracket s_B \rrbracket_\Psi)(\sigma) \triangleq \text{let } (\sigma_A, \tau_A) = \llbracket s_A \rrbracket_\Psi(\sigma) \text{ in}$$
$$\text{let } (\sigma_B, \tau_B) = \llbracket s_B \rrbracket_\Psi(\sigma_A) \text{ in}$$
$$(\sigma_B, \tau_A \oplus \tau_B)$$

where $\sigma_A$ and $\sigma_B$ are the resulting states of $\llbracket s_A \rrbracket_\Psi(\sigma)$ and $\llbracket s_B \rrbracket_\Psi(\sigma_A)$ respectively, and $\tau_A$ and $\tau_B$ are the corresponding execution traces. Consequently, $\llbracket \mathcal{P} \rrbracket_\Psi(\sigma_0) = (\sigma_n, \oplus_{i=0}^{n} \tau_i)$, where $\sigma_n$ is the final state—obtained by exe-

---

[†] Statements $s_0, \ldots, s_n$ are sequenced at the program level as single units, regardless of whether their execution expands into nested statements (e.g., loops).

[‡] The empty program denotes the base case of the compositional definition: $\forall \sigma. \llbracket [\ ] \rrbracket_\Psi(\sigma) \triangleq (\sigma, [\ ])$.

*Table 2.* Median code-complexity statistics of our dataset splits. Control-flow complexity is characterized using extended cyclomatic complexity ($\Omega_{CC}$), maximum nested if–else ($\Omega_{If}$) and nested loop ($\Omega_{Loop}$) depths, maximum taken nested if–else ($\hat{\Omega}_{If}$), and taken nested loop ($\hat{\Omega}_{Loop}$) depths. Program size complexity is measured using lines of code ($\Omega_{Loc}$), Halstead metrics Volume ($\Omega_{Vol}$) and Vocabulary ($\Omega_{Voc}$), and execution trace length ($\hat{\Omega}_{Trace}$). Data-flow complexity is analyzed using DepDegree ($\Omega_{DD}$) and the total number of assignments to variables in execution traces ($\hat{\Omega}_{Assign}$). Metrics computed under dynamic-analysis are shown with a hat.

| Dataset Split | #Programs | #Tokens* | | | Control-flow | | | | | Data-flow | | Size | | | |
|---|---|---|---|---|---|---|---|---|---|---|---|---|---|---|---|
| | | $\Psi^{\Box,std}$ | $\Psi^{\Box,swap}$ | $\Psi^{\Box,obf}$ | $\Omega_{CC}$ | $\Omega_{If}$ | $\Omega_{Loop}$ | $\hat{\Omega}_{If}$ | $\hat{\Omega}_{Loop}$ | $\Omega_{DD}$ | $\hat{\Omega}_{Assign}$ | $\Omega_{Loc}$ | $\Omega_{Vol}$ | $\Omega_{Voc}$ | $\hat{\Omega}_{Trace}$ |
| Human-Written | 162 | 81 | 81 | 142** | 3 | 1 | 1 | 1 | 1 | 12 | 9 | 19 | 320 | 22 | 20 |
| LLM-Translated | 165 | 538 | 538 | 873** | 9 | 1 | 1 | 1 | 1 | 48 | 62 | 106 | 2K | 35 | 180 |
| Fuzzer-Generated | 165 | 9183 | 9183 | 19016** | 100 | 7 | 6 | 2 | 1 | 6K | 86 | 794 | 63K | 112 | 190 |

\* Median token counts using GPT-4o-mini; not used for complexity metrics. \*\* We study tokenization impact on rule-conditioning via a controlled ablation (Appendix F).

cuting the terminal statement $s_n$. This holds true for both $\mathbb{K}$ and $\mathbb{S}$ formalizations. We use the notations $[\![\cdot]\!]_\Psi^\sigma$ and $[\![\cdot]\!]_\Psi^\tau$ to denote accessing the final state and execution trace respectively. We also introduce the projection ($\pi$) based notation for tuple element access: For a tuple $t = (x_1, \ldots, x_k)$, $\pi_i(t) = x_i$ and its compositional extension to an ordered list of tuples $T = [t_1, \ldots, t_n]$ as $\pi_i(T) = [\pi_i(t_1), \ldots, \pi_i(t_n)]$.

# 3. Benchmark Construction

Figure 2 shows the benchmark construction process. We formalize $C^\star$ in both $\mathbb{S}$ and $\mathbb{K}$ semantics (❶). We use the $\mathbb{K}$-framework (❸) to obtain ground-truths (❹) for $\mathbb{K}$ experiments and a custom ANTLR4-based interpreter for those with $\mathbb{S}$. The $C^\star$ program along with the $\mathbb{K}$ or $\mathbb{S}$ derived formalization is used to prompt the LLMs (❺).

## 3.1. Dataset Curation

PLSEMANTICSBENCH contains three splits, the Human-Written, the LLM-Translated, and the Fuzzer-Generated.

**Human-Written**. This set of $C^\star$ programs we manually adapted from C++ solutions to coding problems sourced from LeetCode (LeetCode, 2024), HumanEval (Chen et al., 2021; Zheng et al., 2023), CodeContests (Li et al., 2022), and MBPP (Austin et al., 2021; Orlanski et al., 2023). We use public test cases as input and their corresponding oracles as expected outputs. Variable names are obfuscated by replacing semantically meaningful identifiers (e.g., maxIter) with random strings (Appendix C.1). We validate correctness by executing the programs with $\mathbb{K}$-framework and verifying outputs against the test oracles.

**LLM-Translated**. This set of $C^\star$ programs are translated from C++ programs using LLMs. Specifically, we collect the C++ programs from the CodeForces solutions published on Hugging Face (Penedo et al., 2025). We prompt QWEN 2.5-INST 32B with the $C^\star$ syntax, semantics constraints, the C++ solution and one corresponding public test case to generate a valid $C^\star$ program which we subsequently filter via the $\mathbb{K}$-framework based on successful test execution.

**Fuzzer-Generated**. We construct this with a depth-controlled, semantics-aware, grammar-based fuzzer (Yang

et al., 2011; Han et al., 2019); a fuzzer is a tool that automatically generates programs and it is commonly used for testing compilers and interpreters. The fuzzer samples statements from—assign, if–else, while, break, continue, halt—using depth-tapered probabilities—a cosine decay reduces the chance of generating new if/while as nesting grows—and legality masks that enforce syntactic and semantic validity (Appendix C.2).

**Program complexity and data statistics**. We characterize program complexity along three axes—control-flow, data-flow, and size. For control-flow, we use extended cyclomatic complexity ($\Omega_{CC}$) (McCabe, 1976); the *static* maximum nesting depths of if–else and while ($\Omega_{If}$, $\Omega_{Loop}$); and their *dynamic* counterparts measured along executed paths ($\hat{\Omega}_{If}$, $\hat{\Omega}_{Loop}$). For data-flow, we use DepDegree ($\Omega_{DD}$), which quantifies uses and redefinitions of declared variables (Beyer & Fararooy, 2010), and the total number of executed assignments ($\hat{\Omega}_{Assign}$). For size, we use Halstead Vocabulary and Volume ($\Omega_{Voc}$, $\Omega_{Vol}$) (Halstead, 1977) which captures the symbol variety and program information in bits respectively, lines of code ($\Omega_{Loc}$), and execution-trace length ($\hat{\Omega}_{Trace}$).

Table 2 reports median values of the complexity metrics per dataset. Across $\approx$165 programs per split, the median complexity increases progressively from Human-Written to LLM-Translated to Fuzzer-Generated along all three axes (complexity metrics distributions given in Appendix D).

## 3.2. Semantic Shifts

To test whether models condition their reasoning on explicitly supplied semantics—rather than rely on pretraining-induced associations between surface syntax and behavior—we introduce two semantic shifts, KeywordSwap and KeywordObf. These are transformations of the standard $C^\star$ semantics ($\Psi^{\Box,std}$; $\Box \in \{\mathbb{S}, \mathbb{K}\}$) that preserve rule structure while perturbing the mapping between syntactic symbols and their conventional meanings, enabling controlled tests of whether models follow the supplied inferential rules when syntactic familiarity is disrupted.

**KeywordSwap ($\Psi^{\Box,swap}$).** KeywordSwap swaps the semantic interpretations of selected syntactic operators in the standard semantics with their KeywordSwap counterparts

*Table 3.* Mutations and obfuscations applied to the standard semantics to derive the nonstandard semantics KeywordSwap and KeywordObf.

| Type | Assignment | Arithmetic | | | | | Relational | | | | | | Logical | | | Keyword | | | | |
|---|---|---|---|---|---|---|---|---|---|---|---|---|---|---|---|---|---|---|---|---|
| Standard | = | + | − | ⋆ | / | % | < | <= | > | >= | == | != | ! | && | \|\| | if-else | while | halt | break | continue |
| KeywordSwap [*] | = | − | + | / | ⋆ | % | > | >= | < | <= | != | == | ! | \|\| | && | if-else | while | halt | break | continue |
| KeywordObf [**] | Ⅼ | Ⴎ | ↳ | Ⴟ | ↲ | Ꙩ | Ⴕ | Ⴗ | Ⴆ | Ⴐ | Ⴞ | Ⴄ | ↄ | ↳ | Ⴥ | Ⴑ-Ⴣ | ↙ | Ⴆ | ⅂ | Ⴓ |

[*] Swaps the semantics of standard operator/keyword symbols; [**] Assigns semantics of standard operators/keywords to novel symbols (from the Caucasian-Albanian script).

(Table 3); for example, it exchanges addition (+) and subtraction (−), so that an expression written as x+y is evaluated according to the subtraction rule. Because KeywordSwap preserves surface syntax while altering operational meaning, correct reasoning requires conditioning on the explicit transition rules rather than defaulting to pretraining-derived interpretations of common symbols.

**KeywordObf ($\Psi^{\square,\mathbf{obf}}$).** KeywordObf probes the complementary case in which syntactic familiarity is removed altogether by systematically replacing standard keywords and operators in the standard semantics with symbols drawn from the rarely encountered Caucasian-Albanian script (Gippert & Schulze, 2023) (Table 3). Under KeywordObf, expressions such as x ↳ y execute identically to x+y under standard semantics, but without relying on familiar symbolic cues. By eliminating conventional symbol associations while preserving rule structure, KeywordObf isolates a model's ability to follow explicit operational definitions in the absence of syntactic priors.

# 4. Evaluation Setup

**Models and inference settings**. We evaluate eleven frontier LLMs divided into two classes. The *non-reasoning* group consists of LLAMA-3.3 70B (Grattafiori et al., 2024), QWEN2.5-INST 14B and QWEN2.5-INST 32B (Hui et al., 2024), and GPT-4o-mini (Achiam et al., 2023). The *reasoning* group includes DeepSeek variants DS-LLAMA 70B, DS-QWEN 14B, and DS-QWEN 32B (Guo et al., 2025), as well as o3-mini and GPT-5-mini (OpenAI, 2025), GEMINI-2.5-pro (Kavukcuoglu, 2025), and QwQ 32B (Team, 2025b). We average all reasoning-model runs (and GPT-4o-mini) over three trials. The temperature for all non-reasoning models (except GPT-4o-mini) is set to 0 (prompts and additional details in Appendix E).

**Preliminary validation: formal-notation understanding**. Before testing whether models can condition their reasoning on explicit formal semantic rules, we verify that they can interpret the notation used to express those rules; otherwise downstream failures could reflect superficial misunderstanding of the formalism rather than limitations in rule-conditioned reasoning. We perform this validation using two auxiliary classification tasks: NL → Rule and Rule → NL. In NL→Rule, models select the correct formal rule (out of five choices) given its natural-language description (human-written); conversely, in Rule→NL they identify the correct description for a given rule. Together, these tasks isolate

notation-level understanding at the granularity of individual inference rules.

*Dataset.* Multiple-choice distractors are generated via a hierarchical sampling strategy to prevent reliance on surface lexical cues (e.g., random sampling could produce distractors involving unrelated operators or constructs, enabling pattern matching rather than semantic discrimination). Rules are grouped—in descending order of sampling preference—into families, constructs, and semantic roles (Appendix G.1). We generate 200 samples per task and semantic variant (Standard, KeywordSwap, and KeywordObf).

*Analysis.* Figures 3a and 3b show the results averaged over three runs under *zero-shot* prompting for $\mathbb{S}$ and $\mathbb{K}$. Under all semantic variants and formalizations, most models achieve near-ceiling performance on both NL → Rule and Rule → NL tasks. The performance of the QWEN2.5-INST 14B (Figure 3a, ①) and GPT-4o-mini (Figure 3a, ②) models on the NL → Rule task under $\Psi^{\mathbb{S},\diamond}$ are the exceptions, maxing out at ≈80-90%. Figures 3c and 3e, and Figures 3d and 3f, show the confusion matrices for the top three most mispredicted rules for ⋄=**std** and **swap** respectively, for QWEN2.5-INST 14B and GPT-4o-mini on NL → Rule. Both models primarily confuse structurally adjacent rules that govern small-step reduction of expressions and computations for $\Psi^{\mathbb{S},\mathbf{std}}$and $\Psi^{\mathbb{S},\mathbf{swap}}$formalizations. Most rule mispredictions fall within the *Arithmetic Expression* (7-23) and the *Relational Comparison* (28-51) families.

In summary, most frontier models exhibit stable notation-level competence across semantic formalizations and shifts, indicating that subsequent failures primarily reflect limitations in rule-conditioned reasoning rather than inability to parse the formalism itself. When errors occur, the dominant failure mode is *imprecise discrimination among fine-grained semantic roles within a construct* (e.g., step vs. compute cases), rather than global breakdown or random guessing. $\Psi^{\mathbb{K},\diamond}$ exhibits fewer such confusions, consistent with its coarser rule inventory per construct, which reduces the density of near-miss distractors relative to $\Psi^{\mathbb{S},\diamond}$.

# 5. Experiments and Results

We now evaluate our hypotheses concerning whether LLMs can condition their reasoning on explicitly specified formal semantic rules. Specifically, we test whether models can (H1) compose rules to obtain correct final states (§ 5.1), (H2) select appropriate rules when execution does not mutate

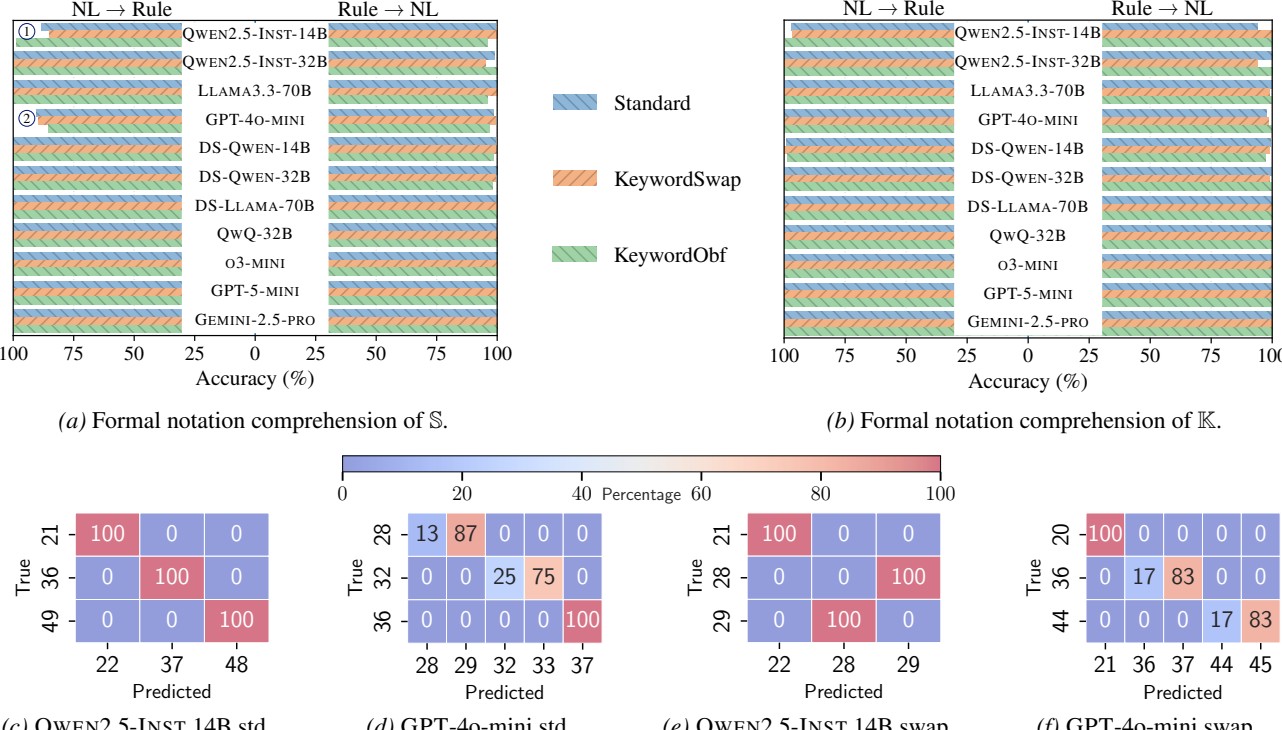

*Figure 3.* Formal-semantics notation comprehension results. Models solve two multiple-choice (five choices) tasks: mapping natural-language descriptions to semantic rules ($\mathtt{NL \to Rule}$) and vice-versa ($\mathtt{Rule \to NL}$). Panels (a,b) show results on $\mathtt{NL \to Rule}$/$\mathtt{Rule \to NL}$ tasks under $\Psi^{\Box,\Diamond}$ ($\Diamond \in \{\mathbf{std}, \mathbf{swap}, \mathbf{obf}\}$) for $\Box = \mathbb{S}$ and $\mathbb{K}$ respectively, averaged over three runs of 200 samples. Panels (c,d) and (e,f) plot $\mathtt{NL \to Rule}$ confusion matrices under $\Psi^{\mathbb{S},\mathbf{std}}$ and $\Psi^{\mathbb{S},\mathbf{swap}}$ respectively for the three most frequently mispredicted rules for QWEN2.5-INST 14B and GPT-4o-mini. For example in (c), QWEN2.5-INST 14B under $\Psi^{\mathbb{S},\mathbf{std}}$ selects rule 22 for rule 21's description in 100% of cases.

state (§ 5.2), (H3) sustain such conditioning across long execution traces (§ 5.3), and (H4) remain faithful to supplied rules under semantic shifts that conflict with learned priors concerning symbol–meaning associations.

We first introduce two task scoped metrics to test our hypotheses. Without loss of generality, consider a model's task-scoped accuracy score $\mathbf{Acc}^{\Box}_{\Diamond}$, potentially** realizable under the set of formalizations: $\{\Psi^{\Box,\Diamond} \mid \Box, \Diamond \in (\{\mathbb{S}, \mathbb{K}\} \times \{\mathbf{std}, \mathbf{swap}, \mathbf{obf}\}) \cup \{\perp^{\dagger\dagger}\}\}$. We then define:

▶ **Semantic Conditioning** ($\Delta_{\mathbf{cnd}}$) $\triangleq \mathbf{Acc}^{\Box}_{\mathbf{std}} - \mathbf{Acc}_{\mathbf{na}}$ quantifies the effect of supplying semantics formalization explicitly on a model's accuracy score. $\mathbf{Acc}_{\mathbf{na}}$ is the model's accuracy when $\Box, \Diamond = \perp$ i.e., when provided with no formalization while $\mathbf{Acc}^{\Box}_{\mathbf{std}}$ is the accuracy under $\Psi^{\Box,\mathbf{std}}$. A positive score is indicative of a model's ability to condition its reasoning on explicitly supplied formal semantics.

▶ **Semantic Shift Sensitivity** ($\Delta_{\mathbf{is}}$) $\triangleq \mathbf{Acc}^{\Box'}_{\Diamond} - \mathbf{Acc}^{\Box}_{\mathbf{std}}$ quantifies the effect of semantic shifts on a model's accuracy

---

** Not all of our tasks support the entire *combination* of pairings of formalizations and semantic shifts.

†† We denote '$\perp$' as the absence of formalization and assume that only the final state of program execution is computable under $\perp$ while its execution-trace is not.

score. $\mathbf{Acc}^{\Box'}_{\Diamond}$ is the model's accuracy when $\Diamond \in \{\mathbf{swap}, \mathbf{obf}\}$ i.e., under semantic shifts while $\mathbf{Acc}^{\Box}_{\mathbf{std}}$ is the accuracy under standard semantics as before, with both being realized under the same semantic framework i.e., $\forall \Box, \Box' \in \{\mathbb{S}, \mathbb{K}\}; \Box = \Box'$. A large negative score is indicative of a model's inability to override pretrained symbol priors when operator meanings are perturbed.

### 5.1. Global Rule-Conditioned Reasoning (H1)

**Motivation**. H1 posits that models can condition their reasoning using explicitly supplied formal semantic rules to determine program final states (program execution reasoning at *coarser granularity*). We introduce the **PredState** task requiring predicting final states by composing rule applications across control and data flow, thereby testing H1 by probing whether semantics guide multi-step execution reasoning rather than acting as inert context.

**Dataset**. From Definition 2.2 the final program state of a program $\mathcal{P}$, with an initial program state $\sigma_0$, and under a semantic formalization $\Psi^{\Box,\Diamond}$ is $[\![\mathcal{P}]\!]^{\sigma_0}_{\Psi^{\Box,\Diamond}}(\sigma_0)$ which we use as the ground-truth for **PredState**. Supposing $\mathcal{D}'$ be a subset of a **PredState** dataset $\mathcal{D}$ for which a model's results are well formed, then we define accuracy over the $\mathcal{D}$ as:

*Table 4.* **Global and state mutation-free rule-conditioned reasoning.** *Left:* predict final program state accuracy via rule composition. *Right:* predict semantic-rules for program execution in absence of state mutation. Best per column is bold.

| | | PredState ($\mathcal{D}$=Human-Written) | | | | | | PredRule | | | | | |
| | | $\mathbb{K}$-Formalization | | | $\mathbb{S}$-Formalization | | | $\mathbb{K}$-Formalization | | | $\mathbb{S}$-Formalization | | |
| **Models** | $Acc_{na}$ | $Acc_{std}$ ($\Delta_{cnd}$) | $Acc_{swap}$ ($\Delta_{is}$) | $Acc_{obf}$ ($\Delta_{is}$) | $Acc_{std}$ ($\Delta_{cnd}$) | $Acc_{swap}$ ($\Delta_{is}$) | $Acc_{obf}$ ($\Delta_{is}$) | $Acc_{std}$ | $Acc_{swap}$ ($\Delta_{is}$) | $Acc_{obf}$ ($\Delta_{is}$) | $Acc_{std}$ | $Acc_{swap}$ ($\Delta_{is}$) | $Acc_{obf}$ ($\Delta_{is}$) |
|---|---|---|---|---|---|---|---|---|---|---|---|---|---|
| *Non-reasoning* | | | | | | | | | | | | | |
| QWEN2.5-INST 14B | 33 | 27(−06) | 6(−21) | 14(−13) | 28(−05) | 6(−22) | 8(−20) | 49 | 45(−04) | 45(−04) | 19 | 19(000) | 17(−02) |
| QWEN2.5-INST 32B | 50 | 29(−21) | 4(−25) | 12(−17) | 33(−17) | 4(−29) | 19(−14) | 58 | 52(−06) | 46(−12) | 17 | 24(+07) | 19(+02) |
| LLAMA-3.3 70B | 32 | 29(−03) | 4(−25) | 12(−17) | 25(−07) | 5(−20) | 12(−13) | 45 | 42(−03) | 45(000) | 32 | 32(000) | 27(−05) |
| GPT-4o-mini | 31 | 26(−05) | 6(−20) | 8(−18) | 24(−07) | 6(−18) | 8(−16) | 38 | 34(−04) | 27(−11) | 27 | 27(000) | 21(−06) |
| *Non-reasoning + Chain-of-thought* | | | | | | | | | | | | | |
| QWEN2.5-INST 14B-CoT | 73 | 70(−03) | 2(−68) | 48(−22) | 68(−05) | 4(−64) | 41(−27) | 50 | 32(−18) | 27(−23) | 12 | 10(−02) | 6(−06) |
| QWEN2.5-INST 32B-CoT | 81 | 77(−04) | 8(−69) | 56(−21) | 69(−12) | 3(−66) | 33(−36) | 64 | 47(−17) | 47(−17) | 29 | 26(−03) | 24(−05) |
| LLAMA-3.3 70B-CoT | 75 | 75(000) | 3(−72) | 56(−19) | 77(+02) | 2(−75) | 48(−29) | 69 | 46(−23) | 50(−19) | 28 | 28(000) | 17(−11) |
| GPT-4o-mini-CoT | 68 | 78(+10) | 2(−76) | 38(−40) | 65(−03) | 3(−62) | 27(−38) | 57 | 46(−11) | 37(−20) | 27 | 26(−01) | 24(−03) |
| *Reasoning* | | | | | | | | | | | | | |
| DS-QWEN 14B | 65 | 81(+16) | 2(−79) | 40(−41) | 58(−07) | 2(−56) | 29(−29) | 57 | 45(−12) | 48(−09) | 22 | 21(−01) | 20(−02) |
| DS-QWEN 32B | 84 | 93(+09) | 21(−72) | 72(−21) | 95(+11) | 3(−92) | 77(−18) | 79 | 66(−13) | 65(−14) | 47 | 38(−09) | 38(−09) |
| DS-LLAMA 70B | 80 | 88(+08) | 2(−86) | 58(−30) | 89(+09) | 2(−87) | 59(−30) | 34 | 10(−24) | 27(−07) | 1 | 1(000) | 1(000) |
| QwQ 32B | 93 | 98(+05) | 71(−27) | 82(−16) | 98(+05) | 7(−91) | 86(−12) | 92 | 85(−07) | 76(−16) | 49 | 44(−05) | 41(−08) |
| o3-mini | 94 | **100**(+06) | 41(−59) | 84(−16) | **100**(+06) | 63(−37) | 95(−05) | 93 | 65(−28) | 84(−09) | 80 | 72(−08) | 67(−13) |
| GPT-5-mini | **100** | 99(−01) | 79(−20) | **94**(−05) | **100**(000) | 79(−21) | 99(−01) | 92 | 83(−09) | 82(−10) | 80 | 81(+01) | 81(+01) |
| GEMINI-2.5-pro | 93 | **100**(+07) | **97**(−03) | **94**(−06) | 99(+06) | **98**(−01) | **100**(+01) | **99** | **98**(−01) | 90(−09) | **94** | **96**(+02) | **98**(+04) |

$$\mathrm{Acc}_{\diamond}^{\square} \triangleq \frac{1}{|\mathcal{D}|} \sum_{(\mathcal{P},\sigma_0)\in\mathcal{D}'} \mathbf{1}\left[ [\![\mathcal{P}]\!]_{\Psi^{\square,\diamond}}^{\sigma}(\sigma_0) = \sigma_{pred}\big|_{\Psi^{\square,\diamond},\mathcal{P},\sigma_0} \right]^{\dagger\ddagger}$$

Where $\square, \diamond \in (\{\mathbb{S}, \mathbb{K}\} \times \{\mathbf{std}, \mathbf{swap}, \mathbf{obf}\}) \cup \{\perp\}$, $\mathbf{dom}(\sigma_0) = \varnothing$. $\mathcal{D} \in$ {Human-Written, LLM-Translated, Fuzzer-Generated} and $\sigma_{pred}\big|_{\Psi^{\square,\diamond},\mathcal{P},\sigma_0}$ is the model's final state prediction for the program $\mathcal{P}$, with an initial state $\sigma_0$, and formalization $\Psi^{\square,\diamond}$.

**Analysis**. Table 4 (left-side) and Table 5 show the accuracy percentages for the Human-Written dataset, and the structurally more complex LLM-Translated and Fuzzer-Generated datasets respectively under *one-shot* prompting. Partial correctness percentage discussed in Appendix G.2.3.

**Does providing formal rules change global composition ($\Delta_{cnd}$)?** On the Human-Written split (Table 4, left-side), $\Delta_{cnd}$ sharply separates model classes: reasoning-oriented models gain 9–16% points (e.g., DS-QWEN 32B, DS-LLAMA 70B), pushing GEMINI-2.5-pro to ≥99% accuracy, while non-reasoning models lose 5–25% points. On the Fuzzer-Generated split (Table 5), even frontier models show generally negative $\Delta_{cnd}$ indicating that structural scale overwhelms global rule-conditioned composition.

**Do models override pretrained symbol biases ($\Delta_{is}$)?** Across both tables, **swap** causes far larger drops than **obf**—often 40–70% points—even when standard accuracy is high. On Human-Written programs (Table 4, left-side), GPT-5-mini and DS-QWEN 32B lose 20–70% points under **swap**, while **obf** causes modest degradation. Because **swap** *preserve surface syntax while changing operator meaning*, these gaps show that most models fail to override pretrained

symbol associations in favor of supplied rules. GEMINI-2.5-pro stands out with ≥98% accuracy even under **swap**.

**Structural factors limiting rule composition ($\Omega_{\square}$).** Moving from Human-Written to LLM-Translated and Fuzzer-Generated programs (Table 5) induces systematic accuracy collapses—often exceeding 40% points—highlighting the fragility of long-horizon rule composition under scale. Multivariate regression (Appendix G.2.1) isolates distinct stressors: control-flow depth dominates on human programs, while data-flow and size-related metrics govern translated and fuzzed inputs, implicating long execution traces and global state tracking as primary bottlenecks.

**Impact of CoT prompting.** On Human-Written programs (Table 4, left-side), chain-of-thought (CoT) boosts non-reasoning models under standard semantics by nearly 50 points. However, these gains vanish under **swap** and shrink to ≈40 points for **obf** indicating that CoT aids long-horizon execution but does not overcome pretrained operator biases.

---

**Key Findings**

⋆ Reasoning-oriented models benefit substantially from access to formal semantics, while non-reasoning models often degrade.
⋆ KeywordSwap causes far larger drops than KeywordObf, revealing persistent reliance on pretrained symbol–semantics associations rather than strict conditioning on provided rules.
⋆ Performance drops with program complexity— deep control flow, heavy data dependencies—highlighting structural limits in global rule-conditioned reasoning.
⋆ CoT prompting boosts non-reasoning models on standard semantics but fails under semantic swaps, indicating reasoning traces alone do not overcome pretrained operator biases.

---

†We generally drop the superscript '$\square$' for Acc when the type of formalization is readily inferrable.

‡ $\sigma_A = \sigma_B \Leftrightarrow \mathbf{dom}(\sigma_A) = \mathbf{dom}(\sigma_B) \wedge \forall x \in \mathbf{dom}(\sigma_A).\sigma_A(x) = \sigma_B(x)$

*Table 5.* **PredState** results on the LLM-Translated and Fuzzer-Generated datasets. Best per column within each dataset is bold.

| Models[*] | $\mathbb{K}$-Formalization | | | | $\mathbb{S}$-Formalization | | |
|---|---|---|---|---|---|---|---|
| | Acc$_{na}$ | Acc$_{std}$ ($\Delta_{cnd}$) | Acc$_{swap}$ ($\Delta_{is}$) | Acc$_{obf}$ ($\Delta_{is}$) | Acc$_{std}$ ($\Delta_{cnd}$) | Acc$_{swap}$ ($\Delta_{is}$) | Acc$_{obf}$ ($\Delta_{is}$) |
| *LLM-Translated* | | | | | | | |
| QwQ 32B | 82 | 83 (+01) | 31 (−52) | 61 (−22) | 82 (000) | 4 (−78) | 63 (−19) |
| GPT-5-mini | 94 | **96** (+02) | 76 (−20) | 86 (−10) | **95** (+01) | 65 (−30) | 90 (−05) |
| GEMINI-2.5-pro | 91 | 94 (+03) | **85** (−09) | **91** (−03) | 94 (+03) | **87** (−07) | **93** (−01) |
| *Fuzzer-Generated* | | | | | | | |
| QwQ 32B | 16 | 16 (000) | 0 (−16) | 3 (−13) | 15 (−01) | 0 (−15) | 1 (−14) |
| GPT-5-mini | 57 | 51 (−06) | 14 (−37) | 23 (−28) | 55 (−02) | 17 (−38) | 23 (−32) |
| GEMINI-2.5-pro | **73** | **69** (−04) | **26** (−43) | **49** (−20) | **69** (−04) | **39** (−30) | **47** (−22) |

\* Only the best scoring models on **PredState** for the Human-Written are considered.

## 5.2. State-Free Rule-Conditioned Reasoning (H2)

**Motivation**. H2 targets a more elementary capability than its predecessor: selecting the correct operational rules at individual steps when program state does not mutate. By removing long-horizon state propagation, this setting isolates whether models ground local decisions in supplied formal semantics rather than surface syntax or pretrained operator associations. We test this hypothesis via the **PredRule** task, asking whether semantic shifts still disrupt rule selection when execution does not mutate state.

**Dataset**. From Definition 2.2 the execution trace $\tau_{\mathcal{P}}$ of a program $\mathcal{P}$, an initial program state $\sigma_0$, under a semantic formalization $\Psi^{\square,\diamond}$ is $[\![\mathcal{P}]\!]^{\tau}_{\Psi^{\square,\diamond}}(\sigma_0)$. The programs ($\mathcal{P}$) in the **PredRule** dataset are constructed from those in the Human-Written split satisfying the invariant $\forall \sigma' \in \pi_1(\tau_{\mathcal{P}}) \wedge \sigma' \neq [\![\mathcal{P}]\!]^{\sigma}_{\Psi^{\square,\diamond}}(\sigma_0), \sigma' = \sigma_0$ i.e., the initial program state is unmutated throughout program execution barring the terminal statement-step. We use the ordered list of semantic rules $\oplus \pi_2(\tau_{\mathcal{P}})$ as the ground-truth. Supposing $\mathcal{D}'$ is a subset of a **PredRule** dataset $\mathcal{D}$ for which the model's predictions are well formed, then we define accuracy over $\mathcal{D}$ as:

$$\text{Acc}^{\square}_{\diamond} \triangleq \frac{1}{|\mathcal{D}|} \sum_{(\mathcal{P},\sigma_0) \in \mathcal{D}'} \mathbf{1}\left[\oplus \pi_2([\![\mathcal{P}]\!]^{\tau}_{\Psi^{\square,\diamond}}(\sigma_0)) = \text{R}_{\text{pred}}\big|_{\Psi^{\square,\diamond},\mathcal{P},\sigma_0}\right]$$

$\square, \diamond \in \{\mathbb{S}, \mathbb{K}\} \times \{\textbf{std}, \textbf{swap}, \textbf{obf}\}$ and $\textbf{dom}(\sigma_0)$ may not be $\varnothing$. $\text{R}_{\text{pred}}\big|_{\Psi^{\square,\diamond},\mathcal{P},\sigma_0}$ is the model predicted ordered list of semantic rules for the program $\mathcal{P}$.

**Analysis**. Table 4 (right-side) shows the accuracy percentages for **PredRule** under *one-shot* prompting. Details about **PredRule** split construction and rule prediction failure rates can be found in Appendix G.3.1 and G.3.2

**Pretrained symbol biases during local rule selection ($\Delta_{is}$)**
Across **PredRule** (Table 4, right-side), most systems exhibit strongly negative $\Delta_{is}$, implying that even when state evolution is removed, local rule selection remains dominated by symbol priors rather than formal definitions. Only a narrow subset of frontier models maintain or improve accuracy under mutation (e.g., only GEMINI-2.5-pro under $\Psi^{\mathbb{S},\diamond}$ shows modest improvement), indicating that faithful local rule conditioning under distribution shift is rare.

*Table 6.* **PredTrace** results on the Human-Written (least complex) dataset split. Best per column is bold.

| Models[*] | $\mathbb{K}$-Formalization | | | $\mathbb{S}$-Formalization | | |
|---|---|---|---|---|---|---|
| | Acc$_{std}$ | Acc$_{swap}$ ($\Delta_{is}$) | Acc$_{obf}$ ($\Delta_{is}$) | Acc$_{std}$ | Acc$_{swap}$ ($\Delta_{is}$) | Acc$_{obf}$ ($\Delta_{is}$) |
| QwQ 32B | 18 | 16 (−02) | 15 (−03) | 0 | 0 (000) | 0 (000) |
| o3-mini | 19 | 3 (−16) | 13 (−06) | 5 | 3 (−02) | 2 (−03) |
| GPT-5-mini | 20 | 14 (−06) | 17 (−03) | 17 | 15 (−02) | 17 (000) |
| GEMINI-2.5-pro | **25** | **25** (000) | **25** (000) | **32** | **35** (+03) | **35** (+03) |

\* Only models with non-zero scores on **PredTrace** are shown.

**Is robust local rule conditioning a general capability?**
The pattern of $\Delta_{is}$ reveals sharp stratification rather than smooth scaling: reasoning-oriented models are typically more stable than non-reasoning ones, but large drops persist even among strong systems (e.g., o3-mini under **swap** drops by ≈30%). This heterogeneity suggests that local rule-conditioned reasoning is not yet a broadly learned behavior across LLM families.

> **Key Findings**
>
> ⋆ **PredRule** isolates rule select/app w/o state tracking; most models nevertheless fail under KeywordSwap, showing persistent reliance on pretrained symbol semantics rather than strict conditioning on provided rules.
> ⋆ Semantic shift robust rule selection emerges only in a small subset of frontier systems, indicating that this capability is rare and not a generic consequence of scale or reasoning prompts (CoT has no impact).

## 5.3. Long-Horizon Rule-Conditioned Reasoning (H3)

**Motivation**. H1 examined global outcomes, while H2 targeted state-free rule selection. H3 asks whether LLMs can *sustain* rule-conditioned reasoning throughout full program executions. We test this via the **PredTrace** task, which—like **PredState** — targets long-horizon reasoning but at a finer, execution-trace granularity: models must generate complete sequences of semantic rule applications and intermediate states, isolating whether they can repeatedly re-ground their reasoning in explicit operational definitions while maintaining the long-range dependencies induced by loops, branching, and mutable stores.

**Dataset**. The execution trace $[\![\mathcal{P}]\!]^{\tau}_{\Psi^{\square,\diamond}}(\sigma_0)$ of a program $\mathcal{P}$ (Definition 2.2), with an initial program state $\sigma_0$, and under a semantic formalization $\Psi^{\square,\diamond}$ is used as the ground-truth in **PredTrace**. If $\mathcal{D}'$ be the subset of a **PredTrace** dataset $\mathcal{D}$ for which the model's predictions are well formed, we define the accuracy over a dataset $\mathcal{D}$ for this analysis as:

$$\text{Acc}^{\square}_{\diamond} \triangleq \frac{1}{|\mathcal{D}|} \sum_{(\mathcal{P},\sigma_0) \in \mathcal{D}'} \mathbf{1}\left[[\![\mathcal{P}]\!]^{\tau}_{\Psi^{\square,\diamond}}(\sigma_0) = \tau_{\text{pred}}\big|_{\Psi^{\square,\diamond},\mathcal{P},\sigma_0}\right]$$

$\square, \diamond \in \{\mathbb{S}, \mathbb{K}\} \times \{\textbf{std}, \textbf{swap}, \textbf{obf}\}$ and $\textbf{dom}(\sigma_0) = \varnothing$. $\mathcal{D}$=Human-Written and $\tau_{\text{pred}}\big|_{\Psi^{\square,\diamond},\mathcal{P},\sigma_0}$ is the model predicted execution trace for the program $\mathcal{P}$.

**Analysis**. Table 6 shows the accuracy scores (%) for **Pred-**

Trace under *one-shot* prompting.

**Can models sustain rule conditioning over long horizons?** PredTrace sharply exposes the fragility of long-horizon rule conditioning: only four models achieve non-zero accuracy at all, and even these remain far from reliable. Under $\mathbb{K}$, all surviving models exhibit negative $\Delta_{is}$ (e.g., QwQ 32B and o3-mini lose 2–16 points under KeywordSwap), indicating that symbol–meaning conflicts rapidly derail multi-step rule application.

**Is long-horizon robustness a rare capability?** The distribution of $\Delta_{is}$ is highly skewed: most models collapse to zero accuracy before robustness can even be meaningfully measured, while the few remaining systems show sharply divergent behavior. For instance, o3-mini and GPT-5-mini degrade under semantic swaps, whereas GEMINI-2.5-pro improves, indicating that the ability to sustain rule conditioning across dozens of steps is *not* a smooth function of scale or reasoning prompts, but instead appears only in a small subset of frontier models.

---

**Key Findings**

★ Sustaining rule-conditioned reasoning over long execution horizons is extremely brittle: most models fail entirely once state tracking and repeated rule application are required.
★ Only GEMINI-2.5-pro exhibits consistently positive $\Delta_{is}$ under $\mathbb{S}$, indicating that mutation-robust long-horizon rule conditioning is a rare and specialized capability.

---

## 6. Related Work

### 6.1. Code Reasoning and Execution Benchmarks

Recent benchmarks evaluate LLMs' ability to reason about program execution and behavior (CRUXEval (Gu et al., 2024), CRUXEval-X (Xu et al., 2025), Live-CodeBench (Jain et al., 2025), BigCodeBench (Zhuo et al., 2024), REval (Chen et al., 2025), CoCoNUT (Beger & Dutta, 2025), CodeMind (Liu et al., 2024), SURGE (Lyu et al., 2025), and LLMs as code executors (Wang et al., 2024)), trace-trained models (CWM (Team, 2025a)), and code-reasoning generalization studies (Yang et al., 2025). These works evaluate end-to-end inputs/outputs or traces under *fixed* language semantics; PLSEMANTICSBENCH instead supplies formal inference rules and uses execution as a controlled lens for whether models condition step-level reasoning on those rules.

### 6.2. Execution-Aware Training

A growing body of work argues that exposing LLMs to program executions improves downstream performance, including execution-guided synthesis (Chen et al., 2018), NExT (Ni et al., 2024), SemCoder (Ding et al., 2024a), TRACED (Ding et al., 2024b), and CodeI/O (Li et al., 2025). Jin & Rinard (2024) further report that representations of formal trace semantics emerge in transformer hidden states

under next-token training. The implicit hypothesis is that models internalize program semantics from such training. PLSEMANTICSBENCH provides the missing diagnostic by directly supplying formal semantic rules and measuring whether models condition their reasoning on those rules.

### 6.3. Perturbing Programs vs. Perturbing Semantics

EquiBench (Wei et al., 2025a), SeqCoBench (Maveli et al., 2025), SPAT (Yu et al., 2022), CodeARC (Wei et al., 2025b), and Orvalho & Kwiatkowska (2025) mutate *programs* under semantics-preserving transformations to test whether models track underlying behavior across syntactic variants. We invert the setup: programs remain syntactically identical while the externally supplied formal semantics are altered, isolating reliance on pretrained symbol–semantics associations from sensitivity to surface form. K-framework formalizations of C (Ellison & Roşu, 2012; Hathhorn et al., 2015), Java (Bogdanas & Roşu, 2015), and Python (Guth et al., 2020) make this methodology directly extensible to richer languages.

### 6.4. Rule Following and Conflicts with Priors

RuleBreakers (Chan et al., 2025) and Sun et al. (2025) probe whether LLMs follow natural-language inferential rules; in NLP more broadly, CheckList (Ribeiro et al., 2020), Contrast Sets (Gardner et al., 2020), HANS (McCoy et al., 2019), NLI stress tests (Naik et al., 2018), and semantic sensitivity probes (Arakelyan et al., 2024) expose heuristic shortcuts via input perturbations, paralleling texture-bias diagnostics in vision (Geirhos et al., 2019). PLSEMANTICSBENCH transposes this question to *formal* rule following: complete operational semantics are supplied and we test whether reasoning conditions on those rules when they redefine standard operator meaning—a conflict that arises in practice with operator overloading, DSLs, and proof assistants.

## 7. Conclusion

We introduced PLSEMANTICSBENCH, a semantics-driven benchmark for studying whether large language models (LLMs) condition their reasoning on explicit formal rules rather than pretrained syntactic priors. Using program execution as a controlled probe, a programming language with two semantic formalisms and shifts, we isolate four capabilities: global rule composition, state-free rule selection, long-horizon conditioning, and robustness to semantic shift.

Across 11 frontier models, performance drops sharply under semantic shifts and long horizons despite high standard-semantics accuracy; only a small subset shows robustness to novel rules. These results position inferential rule conditioning as a largely unsolved capability axis motivating models to adapt to externally specified formal systems rather than entrenched lexical associations.

## Acknowledgments

We thank Cheng Ding, Ivan Grigorik, Michael Y. Levin, Yan Levin, Tong-Nong Lin, Karl Palmskog, Zijian Yi, Zhiqiang Zang, Linghan Zhong and the anonymous reviewers for helpful feedback and discussions.

Computational resources were provided by the Texas Advanced Computing Center at The University of Texas at Austin[†]. This work was supported in part by the U.S. National Science Foundation (NSF) Nos. CCF-2217696, CCF-2313027, CCF-2403036, CCF-2421782; the NSF–Simons AI Institute for Cosmic Origins[‡] funded by NSF award AST-2421782; the Simons Foundation (MPS-AI-00010515); and a sponsored research award by Cisco Research.

The views expressed are those of the authors and do not necessarily reflect those of sponsors.

## Impact Statement

This paper introduces the first large-scale study of whether large language models can condition their reasoning on explicitly provided formal semantics, using program execution as a canonical setting for investigating this capability. We present new evaluation tasks and datasets that probe models' ability to select and compose inference rules across full executions and under controlled semantic perturbations, establishing a foundation for systematic study of rule-grounded reasoning in programming languages.

Overall, this paper positions inferential rule conditioning as a new capability axis for evaluating learning-based systems for programming languages, with the long-term goal of building models that reason more faithfully about formal specifications.

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

# Appendix

## A. Operator Overloading Conflicts Learned Priors

```
1  class X:
2      def __init__(self, v):
3          self.v = v
4      def __add__(self, other):
5          return X(self.v - other.v)
6  print((X(10) + X(3)).v) # 7
```

*(a)* Python (redefining existing operator).

```
1  object Main:
2    extension (x: Int)
3      def ~>(y: Int): Int = x - y
4
5    @main def run(): Unit =
6      println(10 ~> 3) // 7
```

*(b)* Scala (defining new operator).

*Figure 4.* Operator Overloading

There are real-world situations where operators and other language constructs can have very different semantic meaning relative to that assumed during training/pre-training. Very popular languages such as C++, Haskell, Julia, Python, Scala, Swift, etc., support operator overloading where new semantics can be assigned to existing operators (+, -, *, etc.) or to completely new symbols (Haskell, Julia, Scala), previously unencountered during training.

## B. C⋆ Formalization

Here we describe the syntax and semantics of C⋆ used in all our experiments.

### B.1. C⋆ Syntax Description

```
1  <program>   ::= <stmt_list>
2  <stmt_list> ::= (<stmt> ';')*
3  <stmt>      ::= 'int' <id>
4              | <id> '=' <aexp>
5              | 'if' '(' <bexp> ')' '{' <stmt_list> '}' 'else' '{' <stmt_list> '}'
6              | 'while' '(' <bexp> ')' '{' <stmt_list> '}'
7              | 'loop' '(' <bexp> ')' '{' <stmt_list> '}'
8              | 'halt'
9              | 'continue'
10             | 'break'
11             | 'LE'
12 <aexp>      ::= <id>
13             | <literal>
14             | '(' <aexp>? <mathop> <aexp> ')'
15 <bexp>      ::= '(' <bool> ')'
16             | '(' <aexp> <relop> <aexp> ')'
17             | '(' <lognot> <bexp> ')'
18             | '(' <bexp> <logicalop> <bexp> ')'
19 <bool>      ::= 'true' | 'false'
20 <mathop>    ::= '+' | '-' | '*' | '/' | '%'
21 <relop>     ::= '<' | '<=' | '>' | '>=' | '==' | '!='
22 <lognot>    ::= '!'
23 <logicalop> ::= '&&' | '||'
24 <id>        ::= <letter> (<letter> | <digit>)*
25 <literal>   ::= <digit>+
26 <letter>    ::= 'a' | 'b' | 'c' | 'd' | 'e' | 'f' | 'g' | 'h' | 'i' | 'j'
27             | 'k' | 'l' | 'm' | 'n' | 'o' | 'p' | 'q' | 'r' | 's' | 't'
28             | 'u' | 'v' | 'w' | 'x' | 'y' | 'z'
29             | 'A' | 'B' | 'C' | 'D' | 'E' | 'F' | 'G' | 'H' | 'I' | 'J'
30             | 'K' | 'L' | 'M' | 'N' | 'O' | 'P' | 'Q' | 'R' | 'S' | 'T'
31             | 'U' | 'V' | 'W' | 'X' | 'Y' | 'Z'
32 <digit>     ::= '0' | '1' | '2' | '3' | '4' | '5' | '6' | '7' | '8' | '9'
```

*Figure 5.* Complete syntax of C⋆ used in our experiments in EBNF.

The C⋆ syntax used in all our experiments is given in EBNF in Figure 5. The terminals are shown in red while the non-terminals are shown in blue.

### B.2. Small-step Operational Semantics ($\mathbb{S}$) Rules for C⋆

We formalize C⋆ using a small-step structural operational semantics ($\mathbb{S}$). We use two types of configurations: expression configurations $\langle expr, \sigma, \chi \rangle$, and statement configurations: $\langle stmt, \sigma, \chi \rangle$, where $\sigma : id \mapsto literal$ is the program store mapping identifiers to values, and $\chi$ is a last-in, first-out *control stack* of loop headers that records the dynamic nesting of currently active loops: $\chi ::= \epsilon \mid s :: \chi'$. The top of $\chi$ is the innermost executing loop.

*Table 7.* Metavariables used in the $\mathbb{S}$ formalization of C⋆.

| Meta-var | Sort | Ranges over / Domain |
|---|---|---|
| x | id | Identifiers (program variable names) |
| v | literal | Integer literals |
| q | bool | Boolean literals |
| a | aexp | Integer expressions |
| b | bexp | Boolean expressions |
| s | stmt | Statements of the language |
| SL | stmt_list | Finite statement lists (SL ::= ε \| s :: SL') |

We use standard metavariables $x, v, q, a, b, s, SL$ with their sorts summarized in Table 7. For example, $a$ ranges over arithmetic expressions, so rules mentioning $a1, a2, \ldots$ concern arithmetic evaluation. Auxiliary metafunctions (push, pop, top) for manipulating the control stack $\chi$ and concatenating ($++$) statement lists $SL$ are given in Table 8.

*Table 8.* Metafunctions for control stack and statement-list concatenation.

| Function | Signature | Definition |
|---|---|---|
| push | $\text{stmt} \times \text{Stack} \to \text{Stack}$ | $\text{push}(s, \chi) \triangleq s :: \chi$ |
| pop | $\text{Stack}_{\neq \epsilon} \to \text{Stack}$ | $\text{pop}(s :: \chi) \triangleq \chi$ |
| top | $\text{Stack} \to \text{stmt} \cup \{\epsilon\}$ | $\text{top}(\chi) \triangleq \begin{cases} \epsilon & \text{if } \chi = \epsilon, \\ s & \text{if } \chi = s :: \chi' \end{cases}$ |
| ++ | $\text{stmt\_list} \times \text{stmt\_list} \to \text{stmt\_list}$ | $SL1 ++ SL2 \triangleq \begin{cases} SL2 & \text{if } SL1 = \epsilon, \\ s :: (SL1' ++ SL2) & \text{if } SL1 = s :: SL1'. \end{cases}$ |

Program execution proceeds by repeatedly applying the transition relation $\to$ to expression configurations and $\rightrightarrows$ to statement configurations, starting from $\langle SL, \sigma, \chi \rangle$, where $SL$ is the program's statement list, until a terminal configuration is reached. We treat $\langle \epsilon, \sigma, \chi \rangle$, $\langle \text{halt}, \sigma, \chi \rangle$, and $\langle \text{ERROR}, \sigma, \chi \rangle$ statement configurations as terminal configurations.

The complete set of small-step $\mathbb{S}$ rules defining the semantics of $C^\star$ is given in Table 9.

*Table 9.* Small-step $\mathbb{S}$ rules used to formalize $C^\star$.

| Rule | Formalization | Description |
|---|---|---|
| Rule 1 | $$\frac{\sigma(\mathtt{x}) = \mathtt{v}}{\langle \mathtt{x}, \sigma, \chi \rangle \to \mathtt{v}}$$ | Variable lookup returns value. |
| Rule 2 | $$\frac{\sigma(\mathtt{x}) = \bot}{\langle \mathtt{x}, \sigma, \chi \rangle \to \langle \text{ERROR}, \sigma, \chi \rangle}$$ | Read of undefined variable errors. |
| Rule 3 | $$\overline{\langle \mathtt{int\ x\ ::\ \ SL}, \sigma, \chi \rangle \rightrightarrows \langle \mathtt{SL}, \sigma[\mathtt{x} \mapsto 0], \chi \rangle}$$ | Declared int variable initialized to 0. |
| Rule 4 | $$\frac{\langle \mathtt{a}, \sigma, \chi \rangle \to \langle \mathtt{a'}, \sigma, \chi \rangle}{\langle \mathtt{x\ :=\ a\ ::\ \ SL}, \sigma, \chi \rangle \rightrightarrows \langle \mathtt{x\ :=\ a'\ ::\ \ SL}, \sigma, \chi \rangle}$$ | Assignment expression steps. |
| Rule 5 | $$\frac{\sigma(\mathtt{x}) \neq \bot}{\langle \mathtt{x\ :=\ v\ ::\ \ SL}, \sigma, \chi \rangle \rightrightarrows \langle \mathtt{SL}, \sigma[\mathtt{x} \mapsto \mathtt{v}], \chi \rangle}$$ | Writeback to existing variable. |
| Rule 6 | $$\frac{\sigma(\mathtt{x}) = \bot}{\langle \mathtt{x\ :=\ v\ ::\ \ SL}, \sigma, \chi \rangle \rightrightarrows \langle \text{ERROR}, \sigma, \chi \rangle}$$ | Assign to undefined variable errors. |
| Rule 7 | $$\frac{\langle \mathtt{a1}, \sigma, \chi \rangle \to \langle \mathtt{a1'}, \sigma, \chi \rangle}{\langle \mathtt{a1\ +\ a2}, \sigma, \chi \rangle \to \langle \mathtt{a1'\ +\ a2}, \sigma, \chi \rangle}$$ | Plus - step left operand. |
| Rule 8 | $$\frac{\langle \mathtt{a2}, \sigma, \chi \rangle \to \langle \mathtt{a2'}, \sigma, \chi \rangle}{\langle \mathtt{v1\ +\ a2}, \sigma, \chi \rangle \to \langle \mathtt{v1\ +\ a2'}, \sigma, \chi \rangle}$$ | Plus - step right operand. |
| Rule 9 | $$\frac{\mathtt{v3} = \mathtt{v1} + \mathtt{v2}}{\langle \mathtt{v1\ +\ v2}, \sigma, \chi \rangle \to \mathtt{v3}}$$ | Plus - compute. |
| Rule 10 | $$\frac{\langle \mathtt{a1}, \sigma, \chi \rangle \to \langle \mathtt{a1'}, \sigma, \chi \rangle}{\langle \mathtt{a1\ -\ a2}, \sigma, \chi \rangle \to \langle \mathtt{a1'\ -\ a2}, \sigma, \chi \rangle}$$ | Minus - step left operand. |
| Rule 11 | $$\frac{\langle \mathtt{a2}, \sigma, \chi \rangle \to \langle \mathtt{a2'}, \sigma, \chi \rangle}{\langle \mathtt{v1\ -\ a2}, \sigma, \chi \rangle \to \langle \mathtt{v1\ -\ a2'}, \sigma, \chi \rangle}$$ | Minus - step right operand. |
| Rule 12 | $$\frac{\mathtt{v3} = \mathtt{v1} - \mathtt{v2}}{\langle \mathtt{v1\ -\ v2}, \sigma, \chi \rangle \to \mathtt{v3}}$$ | Minus - compute. |
| Rule 13 | $$\frac{\langle \mathtt{a1}, \sigma, \chi \rangle \to \langle \mathtt{a1'}, \sigma, \chi \rangle}{\langle \mathtt{a1\ *\ a2}, \sigma, \chi \rangle \to \langle \mathtt{a1'\ *\ a2}, \sigma, \chi \rangle}$$ | Times - step left operand. |

| Rule 14 | $$\frac{\langle \texttt{a2}, \sigma, \chi \rangle \to \langle \texttt{a2'}, \sigma, \chi \rangle}{\langle \texttt{v1 * a2}, \sigma, \chi \rangle \to \langle \texttt{v1 * a2'}, \sigma, \chi \rangle}$$ | Times - step right operand. |
|---|---|---|
| Rule 15 | $$\frac{\texttt{v3} = \texttt{v1} * \texttt{v2}}{\langle \texttt{v1 * v2}, \sigma, \chi \rangle \to \texttt{v3}}$$ | Times - compute. |
| Rule 16 | $$\frac{\langle \texttt{a1}, \sigma, \chi \rangle \to \langle \texttt{a1'}, \sigma, \chi \rangle}{\langle \texttt{a1 / a2}, \sigma, \chi \rangle \to \langle \texttt{a1' / a2}, \sigma, \chi \rangle}$$ | Division - step left operand. |
| Rule 17 | $$\frac{\langle \texttt{a2}, \sigma, \chi \rangle \to \langle \texttt{a2'}, \sigma, \chi \rangle}{\langle \texttt{v1 / a2}, \sigma, \chi \rangle \to \langle \texttt{v1 / a2'}, \sigma, \chi \rangle}$$ | Division - step right operand. |
| Rule 18 | $$\frac{\texttt{v2} \neq 0 \quad \texttt{v3} = \texttt{v1}/\texttt{v2}}{\langle \texttt{v1 / v2}, \sigma, \chi \rangle \to \texttt{v3}}$$ | Division - compute (nonzero). |
| Rule 19 | $$\frac{\texttt{v2} = 0}{\langle \texttt{v1 / v2}, \sigma, \chi \rangle \to \langle \texttt{ERROR}, \sigma, \chi \rangle}$$ | Division by zero errors. |
| Rule 20 | $$\frac{\langle \texttt{a1}, \sigma, \chi \rangle \to \langle \texttt{a1'}, \sigma, \chi \rangle}{\langle \texttt{a1 \% a2}, \sigma, \chi \rangle \to \langle \texttt{a1' \% a2}, \sigma, \chi \rangle}$$ | Modulus - step left operand. |
| Rule 21 | $$\frac{\langle \texttt{a2}, \sigma, \chi \rangle \to \langle \texttt{a2'}, \sigma, \chi \rangle}{\langle \texttt{v1 \% a2}, \sigma, \chi \rangle \to \langle \texttt{v1 \% a2'}, \sigma, \chi \rangle}$$ | Modulus - step right operand. |
| Rule 22 | $$\frac{\texttt{v2} \neq 0 \quad \texttt{v3} = \texttt{v1 \% v2}}{\langle \texttt{v1 \% v2}, \sigma, \chi \rangle \to \texttt{v3}}$$ | Modulus - compute (nonzero). |
| Rule 23 | $$\frac{\texttt{v2} = 0}{\langle \texttt{v1 \% v2}, \sigma, \chi \rangle \to \langle \texttt{ERROR}, \sigma, \chi \rangle}$$ | Modulus by zero errors. |
| Rule 24 | $$\frac{\langle \texttt{a}, \sigma, \chi \rangle \to \langle \texttt{a'}, \sigma, \chi \rangle}{\langle \texttt{- a}, \sigma, \chi \rangle \to \langle \texttt{- a'}, \sigma, \chi \rangle}$$ | Unary minus - step. |
| Rule 25 | $$\frac{\texttt{v2} = -\texttt{v1}}{\langle \texttt{- v1}, \sigma, \chi \rangle \to \texttt{v2}}$$ | Unary minus - compute. |
| Rule 26 | $$\frac{\langle \texttt{a}, \sigma, \chi \rangle \to \langle \texttt{a'}, \sigma, \chi \rangle}{\langle \texttt{+ a}, \sigma, \chi \rangle \to \langle \texttt{+ a'}, \sigma, \chi \rangle}$$ | Unary plus - step. |
| Rule 27 | $$\frac{}{\langle \texttt{+ v}, \sigma, \chi \rangle \to \texttt{v}}$$ | Unary plus - no-op. |
| Rule 28 | $$\frac{\langle \texttt{a1}, \sigma, \chi \rangle \to \langle \texttt{a1'}, \sigma, \chi \rangle}{\langle \texttt{a1 < a2}, \sigma, \chi \rangle \to \langle \texttt{a1' < a2}, \sigma, \chi \rangle}$$ | Less-than - step left. |
| Rule 29 | $$\frac{\langle \texttt{a2}, \sigma, \chi \rangle \to \langle \texttt{a2'}, \sigma, \chi \rangle}{\langle \texttt{v1 < a2}, \sigma, \chi \rangle \to \langle \texttt{v1 < a2'}, \sigma, \chi \rangle}$$ | Less-than - step right. |
| Rule 30 | $$\frac{\texttt{v1} < \texttt{v2}}{\langle \texttt{v1 < v2}, \sigma, \chi \rangle \to \texttt{true}}$$ | Less-than true. |
| Rule 31 | $$\frac{\texttt{v1} \geq \texttt{v2}}{\langle \texttt{v1 < v2}, \sigma, \chi \rangle \to \texttt{false}}$$ | Less-than false. |
| Rule 32 | $$\frac{\langle \texttt{a1}, \sigma, \chi \rangle \to \langle \texttt{a1'}, \sigma, \chi \rangle}{\langle \texttt{a1 <= a2}, \sigma, \chi \rangle \to \langle \texttt{a1' <= a2}, \sigma, \chi \rangle}$$ | Less-than-equal - step left. |

| Rule 33 | $$\frac{\langle \texttt{a2}, \sigma, \chi \rangle \rightarrow \langle \texttt{a2'}, \sigma, \chi \rangle}{\langle \texttt{v1 <= a2}, \sigma, \chi \rangle \rightarrow \langle \texttt{v1 <= a2'}, \sigma, \chi \rangle}$$ | Less-than-equal - step right. |
|---|---|---|
| Rule 34 | $$\frac{\texttt{v1} \leq \texttt{v2}}{\langle \texttt{v1 <= v2}, \sigma, \chi \rangle \rightarrow \texttt{true}}$$ | Less-than-equal true. |
| Rule 35 | $$\frac{\texttt{v1} > \texttt{v2}}{\langle \texttt{v1 <= v2}, \sigma, \chi \rangle \rightarrow \texttt{false}}$$ | Less-than-equal false. |
| Rule 36 | $$\frac{\langle \texttt{a1}, \sigma, \chi \rangle \rightarrow \langle \texttt{a1'}, \sigma, \chi \rangle}{\langle \texttt{a1 > a2}, \sigma, \chi \rangle \rightarrow \langle \texttt{a1' > a2}, \sigma, \chi \rangle}$$ | Greater-than - step left. |
| Rule 37 | $$\frac{\langle \texttt{a2}, \sigma, \chi \rangle \rightarrow \langle \texttt{a2'}, \sigma, \chi \rangle}{\langle \texttt{v1 > a2}, \sigma, \chi \rangle \rightarrow \langle \texttt{v1 > a2'}, \sigma, \chi \rangle}$$ | Greater-than - step right. |
| Rule 38 | $$\frac{\texttt{v1} > \texttt{v2}}{\langle \texttt{v1 > v2}, \sigma, \chi \rangle \rightarrow \texttt{true}}$$ | Greater-than true. |
| Rule 39 | $$\frac{\texttt{v1} \leq \texttt{v2}}{\langle \texttt{v1 > v2}, \sigma, \chi \rangle \rightarrow \texttt{false}}$$ | Greater-than false. |
| Rule 40 | $$\frac{\langle \texttt{a1}, \sigma, \chi \rangle \rightarrow \langle \texttt{a1'}, \sigma, \chi \rangle}{\langle \texttt{a1 >= a2}, \sigma, \chi \rangle \rightarrow \langle \texttt{a1' >= a2}, \sigma, \chi \rangle}$$ | Greater-than-equal - step left. |
| Rule 41 | $$\frac{\langle \texttt{a2}, \sigma, \chi \rangle \rightarrow \langle \texttt{a2'}, \sigma, \chi \rangle}{\langle \texttt{v1 >= a2}, \sigma, \chi \rangle \rightarrow \langle \texttt{v1 >= a2'}, \sigma, \chi \rangle}$$ | Greater-than-equal - step right. |
| Rule 42 | $$\frac{\texttt{v1} \geq \texttt{v2}}{\langle \texttt{v1 >= v2}, \sigma, \chi \rangle \rightarrow \texttt{true}}$$ | Greater-than-equal true. |
| Rule 43 | $$\frac{\texttt{v1} < \texttt{v2}}{\langle \texttt{v1 >= v2}, \sigma, \chi \rangle \rightarrow \texttt{false}}$$ | Greater-than-equal false. |
| Rule 44 | $$\frac{\langle \texttt{a1}, \sigma, \chi \rangle \rightarrow \langle \texttt{a1'}, \sigma, \chi \rangle}{\langle \texttt{a1 == a2}, \sigma, \chi \rangle \rightarrow \langle \texttt{a1' == a2}, \sigma, \chi \rangle}$$ | Equality - step left. |
| Rule 45 | $$\frac{\langle \texttt{a2}, \sigma, \chi \rangle \rightarrow \langle \texttt{a2'}, \sigma, \chi \rangle}{\langle \texttt{v1 == a2}, \sigma, \chi \rangle \rightarrow \langle \texttt{v1 == a2'}, \sigma, \chi \rangle}$$ | Equality - step right. |
| Rule 46 | $$\frac{\texttt{v1} = \texttt{v2}}{\langle \texttt{v1 == v2}, \sigma, \chi \rangle \rightarrow \texttt{true}}$$ | Equality true. |
| Rule 47 | $$\frac{\texttt{v1} \neq \texttt{v2}}{\langle \texttt{v1 == v2}, \sigma, \chi \rangle \rightarrow \texttt{false}}$$ | Equality false. |
| Rule 48 | $$\frac{\langle \texttt{a1}, \sigma, \chi \rangle \rightarrow \langle \texttt{a1'}, \sigma, \chi \rangle}{\langle \texttt{a1 != a2}, \sigma, \chi \rangle \rightarrow \langle \texttt{a1' != a2}, \sigma, \chi \rangle}$$ | Not-equal - step left. |
| Rule 49 | $$\frac{\langle \texttt{a2}, \sigma, \chi \rangle \rightarrow \langle \texttt{a2'}, \sigma, \chi \rangle}{\langle \texttt{v1 != a2}, \sigma, \chi \rangle \rightarrow \langle \texttt{v1 != a2'}, \sigma, \chi \rangle}$$ | Not-equal - step right. |
| Rule 50 | $$\frac{\texttt{v1} \neq \texttt{v2}}{\langle \texttt{v1 != v2}, \sigma, \chi \rangle \rightarrow \texttt{true}}$$ | Not-equal true. |
| Rule 51 | $$\frac{\texttt{v1} = \texttt{v2}}{\langle \texttt{v1 != v2}, \sigma, \chi \rangle \rightarrow \texttt{false}}$$ | Not-equal false. |

| Rule 52 | $$\dfrac{\langle \text{b1}, \sigma, \chi \rangle \rightarrow \langle \text{b1'}, \sigma, \chi \rangle}{\langle \text{b1 \&\& b2}, \sigma, \chi \rangle \rightarrow \langle \text{b1' \&\& b2}, \sigma, \chi \rangle}$$ | AND - step left. |
|---|---|---|
| Rule 53 | $$\dfrac{\langle \text{b2}, \sigma, \chi \rangle \rightarrow \langle \text{b2'}, \sigma, \chi \rangle}{\langle \text{q1 \&\& b2}, \sigma, \chi \rangle \rightarrow \langle \text{q1 \&\& b2'}, \sigma, \chi \rangle}$$ | AND - step right. |
| Rule 54 | $$\dfrac{\text{q1} = \texttt{true} \wedge \text{q2} = \texttt{true}}{\langle \text{q1 \&\& q2}, \sigma, \chi \rangle \rightarrow \texttt{true}}$$ | AND true. |
| Rule 55 | $$\dfrac{\text{q1} = \texttt{false} \vee \text{q2} = \texttt{false}}{\langle \text{q1 \&\& q2}, \sigma, \chi \rangle \rightarrow \texttt{false}}$$ | AND false. |
| Rule 56 | $$\dfrac{\langle \text{b1}, \sigma, \chi \rangle \rightarrow \langle \text{b1'}, \sigma, \chi \rangle}{\langle \text{b1 || b2}, \sigma, \chi \rangle \rightarrow \langle \text{b1' || b2}, \sigma, \chi \rangle}$$ | OR - step left. |
| Rule 57 | $$\dfrac{\langle \text{b2}, \sigma, \chi \rangle \rightarrow \langle \text{b2'}, \sigma, \chi \rangle}{\langle \text{q1 || b2}, \sigma, \chi \rangle \rightarrow \langle \text{q1 || b2'}, \sigma, \chi \rangle}$$ | OR - step right. |
| Rule 58 | $$\dfrac{\text{q1} = \texttt{true} \vee \text{q2} = \texttt{true}}{\langle \text{q1 || q2}, \sigma, \chi \rangle \rightarrow \texttt{true}}$$ | OR true. |
| Rule 59 | $$\dfrac{\text{q1} = \texttt{false} \wedge \text{q2} = \texttt{false}}{\langle \text{q1 || q2}, \sigma, \chi \rangle \rightarrow \texttt{false}}$$ | OR false. |
| Rule 60 | $$\dfrac{\langle \text{b}, \sigma, \chi \rangle \rightarrow \langle \text{b'}, \sigma, \chi \rangle}{\langle \text{!b}, \sigma, \chi \rangle \rightarrow \langle \text{!b'}, \sigma, \chi \rangle}$$ | NOT - step. |
| Rule 61 | $$\dfrac{\text{q} = \texttt{false}}{\langle \text{!q}, \sigma, \chi \rangle \rightarrow \texttt{true}}$$ | NOT of false is true. |
| Rule 62 | $$\dfrac{\text{q} = \texttt{true}}{\langle \text{!q}, \sigma, \chi \rangle \rightarrow \texttt{false}}$$ | NOT of true is false. |
| Rule 63 | $$\dfrac{\langle \text{s}, \sigma, \chi \rangle \Rightarrow \langle \text{s'}, \sigma', \chi' \rangle}{\langle \text{s :: SL}, \sigma, \chi \rangle \Rightarrow \langle \text{s' :: SL}, \sigma', \chi' \rangle}$$ | Sequence head steps. |
| Rule 64 | $$\dfrac{\langle \text{b}, \sigma, \chi \rangle \rightarrow \langle \text{b'}, \sigma, \chi \rangle}{\langle \text{if(b) \{SL1\} else \{SL2\} :: SL3}, \sigma, \chi \rangle \Rightarrow \langle \text{if(b') \{SL1\} else \{SL2\} :: SL3}, \sigma, \chi \rangle}$$ | If-else predicate steps. |
| Rule 65 | $$\dfrac{\text{q} = \texttt{true}}{\langle \text{if(q) \{SL1\} else \{SL2\} :: SL3}, \sigma, \chi \rangle \Rightarrow \langle \text{SL1 ++ SL3}, \sigma, \chi \rangle}$$ | If-else takes then-branch. |
| Rule 66 | $$\dfrac{\text{q} = \texttt{false}}{\langle \text{if(q) \{SL1\} else \{SL2\} :: SL3}, \sigma, \chi \rangle \Rightarrow \langle \text{SL2 ++ SL3}, \sigma, \chi \rangle}$$ | If-else takes else-branch. |
| Rule 67 | $$\dfrac{}{\langle \text{while(b) \{SL\} :: SL1}, \sigma, \chi \rangle \Rightarrow \langle \text{loop(b) \{SL\} :: SL1}, \sigma, \text{push}(\text{while(b) \{SL\}}, \chi) \rangle}$$ | While creates loop frame. |
| Rule 68 | $$\dfrac{\langle \text{b}, \sigma, \chi \rangle \rightarrow \langle \text{b'}, \sigma, \chi \rangle}{\langle \text{loop(b) \{SL\} :: SL1}, \sigma, \chi \rangle \Rightarrow \langle \text{loop(b') \{SL\} :: SL1}, \sigma, \chi \rangle}$$ | Loop predicate steps. |
| Rule 69 | $$\dfrac{\text{q} = \texttt{false}}{\langle \text{loop(q) \{SL\} :: SL1}, \sigma, \chi \rangle \Rightarrow \langle \text{SL1}, \sigma, \text{pop}(\chi) \rangle}$$ | Loop exits on false. |
| Rule 70 | $$\dfrac{\text{q} = \texttt{true}}{\langle \text{loop(q) \{SL\} :: SL1}, \sigma, \chi \rangle \Rightarrow \langle \text{SL ++ (LE :: SL1)}, \sigma, \chi \rangle}$$ | Insert loop-body into statement list while adding a loop-end (LE) marker in between. |

| Rule 71 | $$\frac{\chi \neq \epsilon \ \wedge \ \mathtt{s} \neq \mathtt{LE}}{\langle \mathtt{break} \ :: \ \mathtt{s} \ :: \ \mathtt{SL}, \sigma, \chi \rangle \rightrightarrows \langle \mathtt{break} \ :: \ \mathtt{SL}, \sigma, \chi \rangle}$$ | break propagates to LE inside loop. |
|---|---|---|
| Rule 72 | $$\frac{\chi \neq \epsilon \ \wedge \ \mathtt{s} = \mathtt{LE}}{\langle \mathtt{break} \ :: \ \mathtt{s} \ :: \ \mathtt{SL}, \sigma, \chi \rangle \rightrightarrows \langle \mathtt{SL}, \sigma, \mathrm{pop}(\chi) \rangle}$$ | break at LE pops $\chi$ and terminates loop. |
| Rule 73 | $$\frac{\chi = \epsilon}{\langle \mathtt{break} \ :: \ \mathtt{SL}, \sigma, \chi \rangle \rightrightarrows \langle \mathtt{ERROR}, \sigma, \chi \rangle}$$ | break outside loop errors. |
| Rule 74 | $$\frac{\chi \neq \epsilon \ \wedge \ \mathtt{s} \neq \mathtt{LE}}{\langle \mathtt{continue} \ :: \ \mathtt{s} \ :: \ \mathtt{SL}, \sigma, \chi \rangle \rightrightarrows \langle \mathtt{continue} \ :: \ \mathtt{SL}, \sigma, \chi \rangle}$$ | continue propagates to LE inside loop. |
| Rule 75 | $$\frac{\chi \neq \epsilon \ \wedge \ \mathtt{s} = \mathtt{LE} \quad \mathtt{s1} = \mathrm{top}(\chi)}{\langle \mathtt{continue} \ :: \ \mathtt{s} \ :: \ \mathtt{SL}, \sigma, \chi \rangle \rightrightarrows \langle \mathtt{s1} \ :: \ \mathtt{SL}, \sigma, \mathrm{pop}(\chi) \rangle}$$ | continue at LE pops $\chi$ and restarts loop. |
| Rule 76 | $$\frac{\chi = \epsilon}{\langle \mathtt{continue} \ :: \ \mathtt{SL}, \sigma, \chi \rangle \rightrightarrows \langle \mathtt{ERROR}, \sigma, \chi \rangle}$$ | continue outside loop errors. |
| Rule 77 | $$\frac{\mathtt{s} = \mathrm{top}(\chi)}{\langle \mathtt{LE} \ :: \ \mathtt{SL}, \sigma, \chi \rangle \rightrightarrows \langle \mathtt{s} \ :: \ \mathtt{SL}, \sigma, \mathrm{pop}(\chi) \rangle}$$ | LE pops $\chi$ and restarts loop. |
| Rule 78 | $$\overline{\langle \mathtt{halt} \ :: \ \mathtt{SL}, \sigma, \chi \rangle \rightrightarrows \langle \mathtt{halt}, \sigma, \chi \rangle}$$ | Halt statement terminates program execution. |

# C. C⋆ Program Example

In this section, we describe the collection of C⋆ programs for: (1) the Human-Written, and (2) the Fuzzer-Generated datasets and provide examples.

## C.1. Human-Written Dataset

```
1   int sumEven(int l, int r)
2   {
3       int sum = 0;
4       for (int i = l; i <= r; i++)
5       {
6           if (i % 2 == 0)
7           {
8               sum += i;
9           }
10      }
11      return sum;
12  }
```

*(a)* The C++ solution to the problem "MBCPP/962" in Babel-Code MBPP and one public test case. The public test we use is sumEven(3, 8)==18.

```
1   int sum;
2   int i;
3   int l;
4   int r;
5   l = 3;
6   r = 8;
7   i = l;
8   while(i <= r)
9   {
10      if((i % 2) == 0)
11      {
12          sum = (sum + i);
13      }
14      else
15      {
16
17      };
18      i = (i + 1);
19  };
```

*(b)* The C⋆ program re-written from the C++ solution.

*Figure 6.* An example of re-writing a C++ program into an C⋆ program in the Human-Written dataset.

In Figure 6, we show an example C++ solution to a problem from the BabelCode MBPP benchmark (Figure 6a) and its corresponding C⋆ program re-written by us (Figure 6b). To convert the C++ program into an C⋆ program, we remove the function definitions (e.g., sumEven), while keeping the body of the function. Unsupported syntactic constructs are either re-written (e.g., replacing the for loop with a while loop) or removed (e.g., removing the return statement). One public test case is adopted as the program input, and its output is used to verify correctness. In this example, l is assigned to 3 and r is assigned to 8, the test oracle 18 is used to verify the final-state of sum after program execution.

The code-complexity profile of the C⋆ program in Figure 6b is: control-flow complexity ($\Omega_{CC} = 3$, $\Omega_{If} = 1$, $\Omega_{Loop} = 1$, $\hat{\Omega}_{If} = 1$, $\hat{\Omega}_{Loop} = 1$), data-flow complexity ($\Omega_{DD} = 12$, $\hat{\Omega}_{Assign} = 12$), and program-size complexity ($\Omega_{Loc} = 19$, $\Omega_{Vol} = 294$, $\Omega_{Voc} = 23$, $\hat{\Omega}_{Trace} = 29$).

## C.2. Fuzzer-Generated Dataset

The Fuzzer-Generated dataset is constructed using a semantic aware grammar based fuzzer with knobs for: (1) the generation probabilities of different statements, (2) the maximum nesting depth of the program (nested loops and conditionals), (3) the maximum and the minimum number of statements to generate per block, (4) the maximum number of terms and variable terms in arithmetic expressions, (5) the maximum number of terms in boolean expressions (relational and logical), and (6) the maximum and the minimum number of variable declarations in a program. We use the settings as shown in Table 10.

The fuzzer starts by randomly sampling an integer from the range defined by the minimum and maximum number of variable declarations. This integer specifies the number of variables to be declared and used for the C⋆ program being generated. The fuzzer next samples alphabets from the set {a-z} and {A-Z} until the required number of unique alphabets to use as variables is obtained. Declaration statments are then generated to declare these variables.

Following this, one assignment statement is generated per declared variable to assign it with a randomly generated arithmetic expression. The arithmetic expression itself is generated using the pool of declared variables and integer constants (sampled from the set {0-9}).

The fuzzer next generates statements from the set {Assignment, While, If, Break, Continue, Halt} in accordance with the statement probabilities given in Table 10. No more than three statements are generated per block. These probabilities are used until the generation block depth reaches the specified minimum block depth (5). Beyond this, the statement probabilities are cosine-tapered to decrease the probabilities of generating while and if-else statements. For generation processes where the block depth reaches the maximum specified block depth (10), the probabilities of further generating while and if-else is reduced to zero.

To ensure high probability in termination of loops, the fuzzer generates one new variable (prefixed with `ble`) per loop. A monotone update type (incrementing or decrementing) is chosen for this variable each with a 50% probability of being chosen. The bounds, initial (before iteration) and expected final (after loop termination) values are then chosen from the range [-20,20] and the size of the update per iteration from the range [1 step, (final / 3) step]. The variable monotone update statement is inserted towards the end of the loop body and the bound is conjoined with the loop predicate. This prevents infinite loops. The declaration and assignment statements for these new generated variables is inserted right after the assignment statements for the intially chosen variables.

The fuzzer can be used to generate extremely complex $C^\star$ programs (as measured by the code-complexity metrics introduced earlier) with high probability of normal program termination. Figure 11 shows an example $C^\star$ program from the Fuzzer-Generated dataset that was generated using our fuzzer. Its code-complexity metric profile is: control-flow complexity ($\Omega_{CC} = 62$, $\Omega_{If} = 5$, $\Omega_{Loop} = 6$, $\hat{\Omega}_{If} = 3$, $\hat{\Omega}_{Loop} = 5$), data-flow complexity ($\Omega_{DD} = 2603$, $\hat{\Omega}_{Assign} = 86$), and program-size complexity ($\Omega_{Loc} = 492$, $\Omega_{Vol} = 37140$, $\Omega_{Voc} = 91$, $\hat{\Omega}_{Trace} = 249$). This shows that out of the maximum loop nesting depth six ($\Omega_{Loop}$) present in the program, the execution reaches a maximum loop nesting depth of five ($\hat{\Omega}_{Loop}$) implying that the execution reached a loop contining four outer loops.

Figure 11 shows one of the programs from the Fuzzer-Generated dataset that the GEMINI-2.5-pro model was successful on in the **PredState** task.

**Table 10.** Settings for the fuzzer knobs used to generate $C^\star$ programs for the Fuzzer-Generated dataset.

| Knob | Value |
|---|---|
| **Structural limits** | |
| Minimum number of statements per block | 1 |
| Maximum number of statements per block | 3 |
| Minimum block depth | 5 |
| Maximum block depth | 10 |
| Minimum number of variables | 5 |
| Maximum number of variables | 10 |
| **Statement generation probabilities** | |
| Assignment | 0.4 |
| While | 0.3 |
| If | 0.2 |
| Break | 0.09 |
| Continue | 0.005 |
| Halt | 0.005 |
| **Expression limits** | |
| Maximum number of terms in arithmetic expr | 6 |
| Maximum number of variable terms in arithmetic expr | 3 |
| Maximum number of terms in boolean expr | 4 |

## D. Code-Complexity Distributions

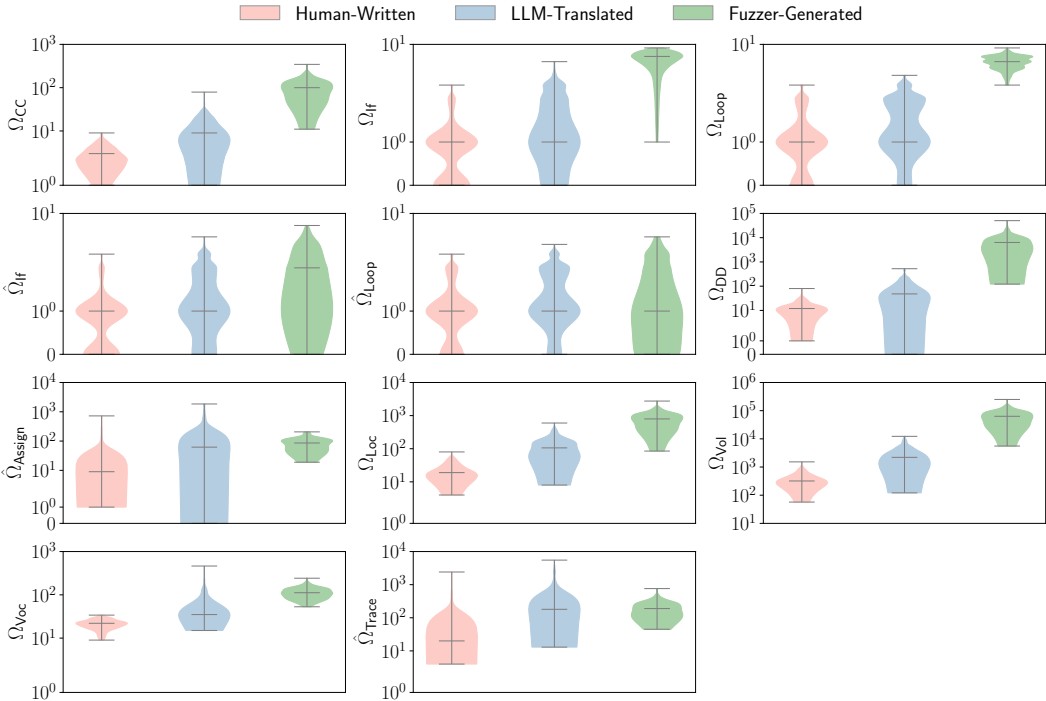

*Figure 7.* Distributions of the code-complexity metrics extended cyclomatic complexity ($\Omega_{CC}$), maximum nested if–else ($\Omega_{If}$) and nested loop ($\Omega_{Loop}$) depths , maximum taken nested if–else ($\hat{\Omega}_{If}$), and taken nested loop ($\hat{\Omega}_{Loop}$) depths, the program data-flow complexity metrics DepDegree ($\Omega_{DD}$) and the total number of assignments to variables in execution traces ($\hat{\Omega}_{Assign}$), and finally the program size complexity metrics, lines of code ($\Omega_{Loc}$), Halstead metrics Volume ($\Omega_{Vol}$) and Vocabulary($\Omega_{Voc}$), and execution trace length ($\hat{\Omega}_{Trace}$).

The distributions of the code-complexity metrics used to characterize the control-flow, data-flow, and the program size complexity are given in Figure 7. We mark the median and the extremas for each distribution. We see that the median $\Omega_{\text{If}}$ and $\Omega_{\text{Loop}}$ is similar for the Human-Written and the LLM-Translated datasets, whereas for every other metric, the LLM-Translated has slightly higher median values than Human-Written and thus more complex programs. The Fuzzer-Generated dataset on the other hand has median values significantly higher for every metric except $\hat{\Omega}_{\text{Trace}}$ and $\hat{\Omega}_{\text{Assign}}$, than the other two datasets. This implies that programs in the Fuzzer-Generated and the LLM-Translated datasets run for roughly the same number of execution steps (measured as per the $\mathbb{S}$ semantics) but the programs in the former are significantly more complex than those in the latter.

# E. Experiments Details

## E.1. Parameters

We use the default temperature settings for reasoning models by not specifying a specific temperature. For other non-reasoning models, we set the temperature to zero. All models are evaluated under one-shot setting.

## E.2. Compute Resources

The experiments on open-weight models with fewer than 70 billion parameters are conducted on a single compute node equipped with one NVIDIA H200 GPU (96 GB memory), an NVIDIA Grace CPU @ 3.1 GHz with 72 cores, and 116 GB LPDDR5 memory. For experiments involving 70B-parameter models, we use four compute nodes.

## E.3. Prompts

### E.3.1 Prompt for **PredState** task.

**No-semantics**:
```
You are an interpreter for my language called {language}.

Here is the {language} program
      {program}
```

$\mathbb{S}$:
```
You are an interpreter for a language called {language}.  I will describe the syntax for {language} in EBNF and
its semantics using small-step operational semantics.  You will use this to execute a {language} program.  You
will only use the rules described in the semantics I provide.  Assume all the rules in the semantics I give
are correct.  A program has finished execution when one of the terminal configurations ⟨ε, σ, χ⟩, ⟨{HALT}, σ, χ⟩,
⟨{ERROR}, σ, χ⟩ is reached.

Here is the syntax of {language} in EBNF
      {syntax}

Here is the small-step operational semantics of {language}
      {semantics}

Here is the {language} program
      {program}
```

$\mathbb{K}$-**semantics**:
```
You are an interpreter for a language called {language}.  I will describe the syntax and the semantics of the
language using the K-framework.  You will use this to execute a {language} program.  You will only use the rules
described in the semantics I provide.  Assume all the rules in the semantics I give are correct.
Here is the K-framework formalization of {language}
      {semantics}

Here is the {language} program
      {program}

## TASK: predict the values of all the declared variables after executing the above program.
- If you think the program will never terminate, answer with the special word '##timeout##':

      <answer>##timeout##</answer>

- If you believe the program has an error or has undefined behavior, answer with the special word '##error##':

      <answer>##error##</answer>

- Otherwise, provide the predicted values of all the declared variables in the following format:
```

```
        <answer>[Your answer]</answer>

Here is one example:

** Program **
int a;
int b;
int ans;
int c;
a {ASSIGN_OP} 10;
b {ASSIGN_OP} 23;
c {ASSIGN_OP} 12;
ans {ASSIGN_OP} a {ADD_OP} b;

The final expected output is:
<answer>
  <a>10</a>
  23
  <c>12</c>
  <ans>33</ans>
</answer>

Non-CoT: Only write the answer.  You **MUST** wrap your prediction with '<answer>' tags.
CoT:  Explain your reasoning step-by-step **before** answering.  Wrap your reasoning in '<reason>' tags.  Note
that you **MUST** wrap your reasoning steps with '<reason>' tags and the prediction with '<answer>' tags.
```

## E.3.2 Prompt for **PredRule** task.

$\mathbb{S}$:
You are an interpreter for a language called {language}.  I will describe the syntax for {language} in EBNF and
its semantics using small-step operational semantics.  You will use this to execute a {language} program.  You
will only use the rules described in the semantics I provide.  Assume all the rules in the semantics I give
are correct.  A program has finished execution when one of the terminal configurations $\langle \epsilon, \sigma, \chi \rangle, \langle \{HALT\}, \sigma, \chi \rangle,$
$\langle \{ERROR\}, \sigma, \chi \rangle$ is reached.

Here is the syntax of {language} in EBNF
        {syntax}

Here is the small-step operational semantics of {language}
        {semantics}

Here is the {language} program
        {program}

## TASK:
For each question below, you'll be given:
1.  A program
2.  The program state ($\sigma$) (variable values) before executing the program
3.  The control stack ($\chi$) before executing the program

Assume that all necessary variables have been declared and have the values as indicated in the provided program
state.
You must:
- Correctly identify and apply the small-step operational semantic rules required to evaluate the program to
completion
- List them in the correct order of application

A program is executed completely when its evaluation reaches one of the terminal configurations
$\langle \epsilon, \sigma, \chi \rangle, \langle \{HALT\}, \sigma, \chi \rangle, \langle \{ERROR\}, \sigma, \chi \rangle$.

Here is one example:
** Program:**
{WHILE} (n {LTEQ_OP} 0)
{{
      {HALT};
}};

**Program state($\sigma$) before execution:**
{{'n': 100, 'sum': 0}}

**Control stack($\chi$) before execution:**
$\epsilon$

This is the sequence of steps:

```
1.  First, we transform the {WHILE} into {LOOP} using **Rule 67**.
2.  Reduce the loop predicate using **Rule 68**.
3.  The loop predicate is a {LTEQ_OP} operator which triggers **Rule 32** to first reduce the left-hand side 'n'
to a literal using **Rule 1**.
4.  The right-hand side is already a literal and since '100' is not less-than or equal to '0'.  We use **Rule
35** to evaluate this operation to 'false'.
5.  Since the loop predicate is 'false', we use **Rule 69** to terminate the loop.
6.  Since there are no more statements left, we have reached the terminal configuration ⟨ε, σ, χ⟩ and the program
evaluation terminates.

Therefore, the final answer is:
<ans>
   <answer id="1">
     <rule>67</rule>
     <rule>68</rule>
     <rule>32</rule>
     <rule>1</rule>
     <rule>35</rule>
     <rule>69</rule>
   </answer>
</ans>

## Questions:
{questions}

## Response Format:
Respond with an XML block structured as follows:

<ans>
   <answer id="1">
     <rule>1</rule>
     <rule>2</rule>
   ...
   </answer>
   <answer id="2">
     <rule>1</rule>
     <rule>2</rule>
   ...
   </answer>
   ...
</ans>

### Notes:
- Each <answer id="N"> element corresponds to the N-th question.
- Inside each <answer> block, list each semantic rule in the correct order using <rule> tags.

## Important Notes:
- The **order** of rules matters and should reflect the evaluation sequence.
- A single rule may be needed to be applied multiple times during evaluation.
- You must include **all** semantic rules required for complete execution.
- Base your analysis solely on the provided semantics, not on general programming knowledge.

𝕂-semantics:
You are an interpreter for a language called {language}.  I will describe the syntax and the semantics of the
language using the K-framework.  You will use this to execute a {language} program.  You will only use the rules
described in the semantics I provide.  Assume all the rules in the semantics I give are correct.

Here is the K-framework formalization of {language}
       {semantics}

Here is the {language} program
       {program}

## TASK:
For each question below, you'll be given:
1.  A program
2.  The program state (σ) (variable values) before executing the program
3.  The control stack (χ) before executing the program

Assume that all necessary variables have been declared and have the values as
indicated in the provided program state.

You must:
- Correctly identify and apply the K-semantic rules required to evaluate the program to completion
- List them in the correct order of application
```

```
Here is one example:
** Program:**
{WHILE} (n {LTEQ_OP} 0)
{{
      {HALT};
}};

**Program state(σ) before execution:**
{{'n':  100, 'sum':  0}}

**Control stack(χ) before execution:**
ε

This is the sequence of steps:
1.  First, we transform the '{WHILE}' into '{WHILE}1' while also inserting a 'breakMarker' after '{WHILE}1' using
**Rule 24**.
2.  Next we transform the '{WHILE}1' into an '{IF}-{ELSE}' with the '{WHILE}1' as the body of the '{IF}' using
**Rule 25**.
3.  We then reduce the loop predicate to a boolean by first reducing left-hand-side which is a variable using
**Rule 1** and then applying the '{LTEQ_OP}' using **Rule 13**.
4.  Since the loop predicate evaluates to 'false', we apply the '{IF}' not taken rule **Rule 23** to take the
'{ELSE}' branch which is empty.
5.  Finally, we evaluate the 'breakMarker' statement using **Rule 27** to conclude the program execution.

Therefore, the final answer is:
<ans>
  <answer id="1">
    <rule>24</rule>
    <rule>25</rule>
    <rule>1</rule>
    <rule>13</rule>
    <rule>23</rule>
    <rule>27</rule>
  </answer>
</ans>

## Questions:
{questions}

## Response Format:
Respond with an XML block structured as follows:

<ans>
  <answer id="1">
    <rule>1</rule>
    <rule>2</rule>
  ...
  </answer>
  <answer id="2">
    <rule>1</rule>
    <rule>2</rule>
  ...
  </answer>
  ...
</ans>

### Notes:
- Each '<answer id="N">' element corresponds to the N-th question.
- Inside each '<answer>' block, list each semantic rule in the correct order using '<rule>' tags.

## Important Notes:
- The **order** of rules matters and should reflect the evaluation sequence.
- Only rules that have names indicated in '[]' adjacent to it must be reported in the answer.
- A single rule may be needed to be applied multiple times during evaluation.
- You must include **all** semantic rules required for complete execution.
- Base your analysis solely on the provided semantics, not on general programming knowledge.

Non-CoT: Only output the '<ans>' XML block.  Do not include any other content.
CoT:  Explain your reasoning step-by-step **before** answering.  Wrap your reasoning in '<reason>' tags.
```

$\mathbb{S}$:
You are an interpreter for a language called {language}. I will describe the syntax for {language} in EBNF and
its semantics using small-step operational semantics. You will use this to execute a {language} program. You
will only use the rules described in the semantics I provide. Assume all the rules in the semantics I give
are correct. A program has finished execution when one of the terminal configurations $\langle \epsilon, \sigma, \chi \rangle, \langle \{\text{HALT}\}, \sigma, \chi \rangle,$
$\langle \{\text{ERROR}\}, \sigma, \chi \rangle$ is reached.

Here is the syntax of {language} in EBNF
        {syntax}

Here is the small-step operational semantics of {language}
        {semantics}

Here is the {language} program
        {program}

## TASK:
Given a program and its semantics, predict the execution trace. Your goal is to simulate execution, step by step
of executing the program using the given small-step operational semantics rules. Do not skip any rules that is
needed to evaluate the program. You will output your answer in the following format.

## Response Format:
Respond with an XML block structured as follows:

```
<answer>
  <step>
    <rule>1</rule>
    <program_state>
      <n>0</n>
      <sum>0</sum>
    </program_state>
  </step>
  <step>
    <rule>2</rule>
    <program_state>
      <n>100</n>
      <sum>0</sum>
    </program_state>
  </step>
...
</answer>
```

## Here is an example:

Here is the {language} program:
```
int i;
int j;
i {ASSIGN_OP} 0;
{WHILE} (i {LT_OP} 2)
{{
      {HALT};
}};
```

## Expected output:
```
<answer>
  <step>
    <rule>3</rule>
    <program_state>
      0
    </program_state>
  </step
  <step>
    <rule>3</rule>
    <program_state>
      0
      <j>0</j>
    </program_state>
  </step>
  <step>
    <rule>5</rule>
    <program_state>
      0
      <j>0</j>
    </program_state>
  </step>
  <step>
```

```
      <rule>67</rule>
      <program_state>
        0
        <j>0</j>
      </program_state>
    </step>
    <step>
      <rule>68</rule>
      <program_state>
        0
        <j>0</j>
      </program_state>
    </step>
    <step>
      <rule>28</rule>
      <program_state>
        0
        <j>0</j>
      </program_state>
    </step>
    <step>
      <rule>1</rule>
      <program_state>
        0
        <j>0</j>
      </program_state>
    </step>
    <step>
      <rule>30</rule>
      <program_state>
        0
        <j>0</j>
      </program_state>
    </step>
    <step>
      <rule>70</rule>
      <program_state>
        0
        <j>0</j>
      </program_state>
    </step>
    <step>
      <rule>78</rule>
      <program_state>
        0
        <j>0</j>
      </program_state>
    </step>
</answer>

## Notes:
- Each '<step>' must correspond to **exactly one small-step operational semantics rule** that is needed to
evaluate a statement in the given program.
- The '<rule>' must indicate a rule used in the evaluation of a statement.
- The '<program_state>' must represent the **entire program state immediately after** the execution of that rule.
- The program state must list **all variables currently in scope**, using the variable names as XML tags and
their current values as tag content.
- Include variables even if they did not change.
- Do not skip any step or merge multiple steps into one.
- Do not skip any rules (including those used to reduce expressions and variables) that are needed to evaluate
the program.
- The program execution is complete when one of the terminal configurations $\langle \epsilon, \sigma, \chi \rangle, \langle \{\text{HALT}\}, \sigma, \chi \rangle, \langle \{\text{ERROR}\}, \sigma, \chi \rangle$ is
reached
```

**$\mathbb{K}$-semantics**:
You are an interpreter for a language called {language}.  I will describe the syntax and the semantics of the
language using the K-framework.  You will use this to execute a {language} program.  You will only use the rules
described in the semantics I provide.  Assume all the rules in the semantics I give are correct.

Here is the K-framework formalization of {language}
        {semantics}

Here is the {language} program
        {program}

## TASK:
Given a program and its semantics, predict the execution trace.  Your goal is to simulate execution, step by step

```
of executing the program using the given K-framework semantics rules.  Do not skip any rules that is needed to
evaluate the program.  You will output your answer in the following format.

## Response Format:
Respond with an XML block structured as follows:

<answer>
  <step>
    <rule>1</rule>
    <program_state>
      <n>0</n>
      <sum>0</sum>
    </program_state>
  </step>
  <step>
    <rule>2</rule>
    <program_state>
      <n>100</n>
      <sum>0</sum>
    </program_state>
  </step>
...
</answer>

## Here is an example:

Here is the {language} program:
int i;
int j;
i {ASSIGN_OP} 0;
{WHILE} (i {LT_OP} 2)
{{
      {HALT};
}};

## Expected output:

<answer>
  <step>
    <rule>36</rule>
    <program_state>
      0
    </program_state>
  </step>
  <step>
    <rule>36</rule>
    <program_state>
      0
      <j>0</j>
    </program_state>
  </step>
  <step>
    <rule>21</rule>
    <program_state>
      0
      <j>0</j>
    </program_state>
  </step>
  <step>
    <rule>24</rule>
    <program_state>
      0
      <j>0</j>
    </program_state>
  </step>
  <step>
    <rule>25</rule>
    <program_state>
      0
      <j>0</j>
    </program_state>
  </step>
  <step>
    <rule>1</rule>
    <program_state>
      0
      <j>0</j>
    </program_state>
  </step>
```

```
   <step>
     <rule>12</rule>
     <program_state>
       0
       <j>0</j>
     </program_state>
   </step>
   <step>
     <rule>22</rule>
     <program_state>
       0
       <j>0</j>
     </program_state>
   </step>
   <step>
     <rule>26</rule>
     <program_state>
       0
       <j>0</j>
     </program_state>
   </step>
</answer>
```

```
## Notes:
- Each '<step>' must correspond to **exactly one K-semantics re-write rule** that is needed to evaluate a
statement in the given program.
- Only rules that have names indicated in '[]' adjacent to it must be reported in the answer.
- The '<rule>' must indicate a rule used in the evaluation of a statement.
- The '<program_state>' must represent the **entire program state immediately after** the execution of that rule.
- The program state must list **all variables currently in scope**, using the variable names as XML tags and
their current values as tag content.
- Include variables even if they did not change.
- Do not skip any step or merge multiple steps into one.
- Do not skip any rules (including those used to reduce expressions and variables) that are needed to evaluate
the program.

Non-CoT: Only output the '<answer>' XML block.  Do not include explanations, comments, or any other text.
CoT: Explain your reasoning step-by-step **before** answering.  Wrap your reasoning in '<reason>' tags.  Note
that you **MUST** wrap your reasoning steps with '<reason>' tags, the prediction with '<answer>' tags.
```

# F. Is Keyword Obfuscation a Tokenization Artifact?

A potential concern with the KeywordObf semantic shift is that it introduces rare Unicode characters (e.g., 𐕄 from Caucasian-Albanian script) that may be split into many subword tokens, artificially inflating prompt length and degrading model performance. If so, the observed failures would primarily reflect tokenizer limitations rather than deficiencies in rule-conditioned reasoning.

To isolate this factor, we perform a controlled ablation that preserves the semantic transformation of KeywordObf while removing tokenization effects.

## F.1. Tokenizer-Controlled Symbol Substitution

We construct a variant, 1Tok, in which every keyword and operator is replaced with a symbol verified to be encoded as a single token by GPT-4o-mini tokenizer.

*Table 11.* **PredState** accuracy under KeywordObf with different symbol inventories. We compare the original Caucasian-Albanian symbol substitution against a tokenization-controlled variant (1Tok) in which every keyword is replaced by a symbol verified to occupy a single GPT-4o-mini token. Accuracy remains low in both settings, indicating that failures are not driven by token inflation.

| Semantics | Variant | Model | Accuracy(%) |
|---|---|---|---|
| $\mathbb{S}$ | Caucasian-Albanian | GPT-4o-mini | 8.4 |
| | | GPT-4o-mini-CoT | 21.8 |
| | 1Tok | GPT-4o-mini | 7.0 |
| | | GPT-4o-mini-CoT | 19.3 |
| $\mathbb{K}$ | Caucasian-Albanian | GPT-4o-mini | 8.2 |
| | | GPT-4o-mini-CoT | 18.1 |
| | 1Tok | GPT-4o-mini | 6.2 |
| | | GPT-4o-mini-CoT | 16.9 |

This shift maintains the same distribution shift—models must map novel surface forms to formal rules—while preventing input-length inflation from multi-byte Unicode characters.

## F.2. Experimental Setup

We rerun the **PredState** task on the Human-Written split under KeywordObf semantics for both $\mathbb{S}$ and $\mathbb{K}$ based formalizations, comparing the original Caucasian-Albanian symbol substitution to the new 1-token variant. All prompts, rules, and evaluation procedures match those in §5.1; only the symbol inventory changes.

## F.3. Results

Table 11 reports the results. Replacing multi-token Unicode symbols with guaranteed single-token alternatives yields only modest changes in accuracy.

Under $\mathbb{S}$, GPT-4o-mini decreases slightly from 8.4% to 7.0%, and GPT-4o-mini-CoT from 21.8% to 19.3%. A similar pattern holds for $\mathbb{K}$, where GPT-4o-mini drops from 8.2% to 6.2% and GPT-4o-mini-CoT from 18.1% to 16.9%.

Crucially, in all cases performance remains far below that observed under standard semantic formalization, and the qualitative failure pattern under KeywordObf is unchanged.

## F.4. Implications

These results rule out tokenization inefficiency as the primary driver of degraded performance under KeywordObf. Even when all obfuscated symbols are atomic tokens, models still fail to reliably apply the correct operational rules and compose them over execution.

We therefore conclude that KeywordObf probes limitations in rule-conditioned reasoning over unfamiliar formal systems, rather than lexical or tokenizer artifacts.

# G. Task Extended Analysis

## G.1. Formal Semantics Notation Comprehension

*Table 12.* Hierarchical organization of $\mathbb{S}$ semantics rules into families and constructs, with associated categories and semantic roles, used to sample near-miss distractors for the formal-semantics notation comprehension tasks.

| Family | Construct | Rules | Category | Semantic Role |
|---|---|---|---|---|
| **A. Variable & State Access** | | | | |
| F1 | Variable lookup | 1–2 | AEXP | Read $\sigma(x)$; distinguishes value vs unbound-variable error |
| **B. Declarations & Assignment** | | | | |
| F2 | Declaration | 3 | SL | Initialize state via $\sigma[x \mapsto 0]$ |
| F3 | Assignment | 4–6 | SL | Step RHS vs commit update; uninitialized-assignment error |
| **C. Arithmetic Expressions (Binary)** | | | | |
| F4 | Addition | 7–9 | AEXP | Left-step / right-step / compute for binary $+$ |
| F5 | Subtraction | 10–12 | AEXP | Left-step / right-step / compute for binary $-$ |
| F6 | Multiplication | 13–15 | AEXP | Left-step / right-step / compute for binary $\times$ |
| F7 | Division | 16–19 | AEXP | Nonzero vs div-by-zero error case |
| F8 | Modulo | 20–23 | AEXP | Nonzero vs mod-by-zero error case |
| **D. Arithmetic Expressions (Unary)** | | | | |
| F9 | Unary minus | 24–25 | AEXP | Step argument vs compute negation |
| F10 | Unary plus | 26–27 | AEXP | Step argument vs compute identity |
| **E. Relational Comparisons** | | | | |
| F11 | Less-than | 28–31 | BEXP | Step operands; compute boolean (true/false) for $<$ |
| F12 | Less-or-equal | 32–35 | BEXP | Step operands; compute boolean (true/false) for $\leq$ |
| F13 | Greater-than | 36–39 | BEXP | Step operands; compute boolean (true/false) for $>$ |
| F14 | Greater-or-equal | 40–43 | BEXP | Step operands; compute boolean (true/false) for $\geq$ |
| F15 | Equality | 44–47 | BEXP | Step operands; compute boolean (true/false) for $=$ |
| F16 | Inequality | 48–51 | BEXP | Step operands; compute boolean (true/false) for $\neq$ |
| **F. Boolean Connectives** | | | | |
| F17 | Boolean AND | 52–55 | BEXP | Step operands; compute conjunction |
| F18 | Boolean OR | 56–59 | BEXP | Step operands; compute disjunction |
| F19 | Boolean NOT | 60–62 | BEXP | Step operand; compute negation |
| **G. Sequencing & Statement Plumbing** | | | | |
| F20 | Sequencing / head stepping | 63 | SL | Lift a head-statement step into the statement list |
| **H. Conditional Control Flow** | | | | |
| F21 | If–then–else | 64–66 | CTRL | Step condition; branch via list splicing |
| **I. Loops & Loop Context** | | | | |
| F22 | While entry | 67 | CTRL | Desugar WHILE; push loop frame onto $\chi$ |
| F23 | Loop execution | 68–70 | CTRL | Exit on false (pop $\chi$); iterate on true (insert LE) |
| F24 | Loop-exit marker (LE) | 77 | CTRL | Restore continuation via $\mathrm{top}(\chi)$ |
| **J. Non-local Control Flow** | | | | |
| F25 | Break | 71–73 | CTRL | Propagate to LE; handle at LE; error if $\chi = \epsilon$ |
| F26 | Continue | 74–76 | CTRL | Propagate to LE; resume loop; error if $\chi = \epsilon$ |
| **K. Termination** | | | | |
| F27 | Halt | 78 | CTRL | Terminal configuration; stops execution |

This section provides the full details of the hierarchical distractor sampling strategy and the rule-family coverage distributions referenced in § 4.

**Hierarchical rule organization**. Tables 12 and 13 list every semantic rule used in our NL→Rule and Rule→NL evaluation tasks for $\mathbb{S}$ and $\mathbb{K}$ respectively. Rules are organized into 27 *families* (F1–F27), grouped under 11 top-level categories (A–K). Each family corresponds to a single language construct (e.g., addition, while-entry) and may contain multiple rules that differ in their *semantic role* (e.g., left-step vs. right-step vs. compute for a binary arithmetic operator under $\mathbb{S}$). $\mathbb{K}$ rules are at a coarser granularity: because $\mathbb{K}$ is a big-step semantics, many of the intermediate reduction steps present under $\mathbb{S}$ (e.g., left-step and right-step rules for binary operators) are collapsed into a single rule, yielding fewer rules per family.

**Distractor sampling strategy**. When constructing each multiple-choice sample (five choices), we draw distractors in a hierarchical order designed to maximize semantic proximity to the correct answer:

1. **Same family, different semantic role.** We first attempt to sample distractors from the same family as the correct rule. These share the same language construct but differ in semantic role (e.g., the compute rule vs. a step rule for addition), making them the hardest distractors.

*Table 13.* Hierarchical organization of $\mathbb{K}$ semantics rules into families and constructs, with associated categories and semantic roles, used to sample near-miss distractors for the formal-semantics notation comprehension tasks.

| Family | Construct | Rules | Category | Semantic Role |
|---|---|---|---|---|
| **A. Variable & State Access** | | | | |
| F1 | Variable lookup | 1–2 | AEXP | Read state binding; distinguishes value retrieval vs unbound-variable error (halts). |
| **B. Declarations & Assignment** | | | | |
| F2 | Declaration | 36 | SL | Initialize state (introduce variable with default value 0). |
| F3 | Assignment | 21 | SL | Commit update to state once RHS is a value (update existing variable). |
| **C. Arithmetic Expressions (Binary)** | | | | |
| F4 | Addition | 3 | AEXP | Compute binary $+$ when both operands are values. |
| F5 | Subtraction | 4 | AEXP | Compute binary $-$ when both operands are values. |
| F6 | Multiplication | 5 | AEXP | Compute binary $\times$ when both operands are values. |
| F7 | Division | 6–7 | AEXP | Case split: nonzero divisor computes division; zero divisor raises error (halts). |
| F8 | Modulo | 8–9 | AEXP | Case split: nonzero divisor computes modulus; zero divisor raises error (halts). |
| **D. Arithmetic Expressions (Unary)** | | | | |
| F9 | Unary minus | 11 | AEXP | Compute unary negation once operand is a value. |
| F10 | Unary plus | 10 | AEXP | Compute unary identity (operationalized as adding 0) once operand is a value. |
| **E. Relational Comparisons** | | | | |
| F11 | Less-than | 12 | BEXP | Compute boolean result for < on value operands. |
| F12 | Less-or-equal | 13 | BEXP | Compute boolean result for <= on value operands. |
| F13 | Greater-than | 14 | BEXP | Compute boolean result for > on value operands. |
| F14 | Greater-or-equal | 15 | BEXP | Compute boolean result for >= on value operands. |
| F15 | Equality | 16 | BEXP | Compute boolean result for == on value operands. |
| F16 | Inequality | 17 | BEXP | Compute boolean result for != on value operands. |
| **F. Boolean Connectives** | | | | |
| F17 | Boolean AND | 19 | BEXP | Compute conjunction on boolean operands. |
| F18 | Boolean OR | 20 | BEXP | Compute disjunction on boolean operands. |
| F19 | Boolean NOT | 18 | BEXP | Compute negation on boolean operand. |
| **H. Conditional Control Flow** | | | | |
| F21 | If–then–else | 22–23 | CTRL | Branch on boolean predicate. |
| **I. Loops & Loop Context** | | | | |
| F22 | While entry | 24 | CTRL | Push loop statement onto stack; rewrite to internal form; insert markers. |
| F23 | Loop execution | 25 | CTRL | Desugar internal while into if–else: iterate on true, exit on false. |
| F24 | Loop markers | 27–28 | CTRL | Marker plumbing: remove loop markers when encountered to resume execution. |
| **J. Non-local Control Flow** | | | | |
| F25 | Break | 33–35 | CTRL | Handle `break` statements |
| F26 | Continue | 29–32 | CTRL | Handle `continue` statements |
| **K. Termination** | | | | |
| F27 | Halt | 26 | CTRL | Terminal configuration: stops execution immediately; removes remaining computation. |

2. **Same category, different construct.** If the family does not contain enough candidate rules, we sample from other families within the same top-level category (e.g., another binary arithmetic operator from category C).

3. **Different category.** As a last resort, distractors are drawn from a different category entirely (e.g., a control-flow rule used as a distractor for an arithmetic-expression question).

This ordering ensures that each question is discriminative: models must distinguish among rules that govern closely related constructs rather than exploit superficial differences in operator type or syntactic category.

### G.2. Final-State Prediction (PredState)

This section analyzes (1) the impact of code-complexity metrics on LLM performance in the PredState task, and (2) the average percentage of variables per program whose final states are predicted correctly.

G.2.1. IMPACT OF CODE-COMPLEXITY METRICS

Figure 8a illustrates the workflow of the PredState task. An C$^\star$ program, together with optional semantics ($\mathbb{K}$-semantics or $\mathbb{S}$) and syntax, is used both to construct prompts for the LLMs and to obtain gold final states by executing the program in the $\mathbb{K}$-framework. The LLM's predicted final states are then compared with the gold states for each declared variable. A match is recorded as 1 (pass), and a mismatch as 0 (fail).

Different LLMs naturally excel on different C$^\star$ programs. To understand why an LLM may predict all final states correctly for one program but fail on another, we cast this task as a classification problem as shown in Figure 8b. Each C$^\star$ program is mapped to a predictor vector that characterizes its complexity, using the code-complexity metrics introduced earlier. Each predictor is then normalized using z-score normalization to ensure fair contribution from all the variables. The resulting

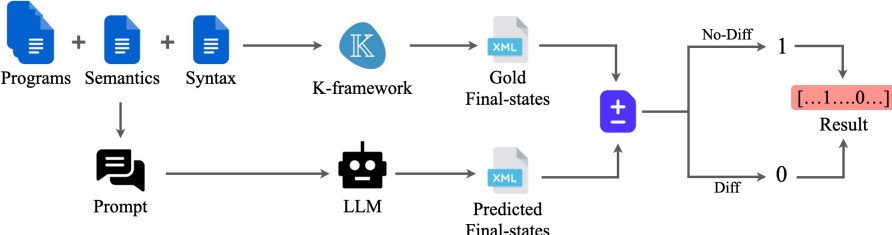

*(a)* Workflow of the **PredState** task. C$^\star$ programs, along with optional semantics (K-semantics or $\mathbb{S}$) and syntax, are: (1) executed in the $\mathbb{K}$-framework to obtain the gold final states of all declared variables, and (2) used to construct a prompt for the LLMs to predict those final states. The gold and predicted states are then compared, scored as `1` for a match and `0` otherwise, and accumulated into a result vector.

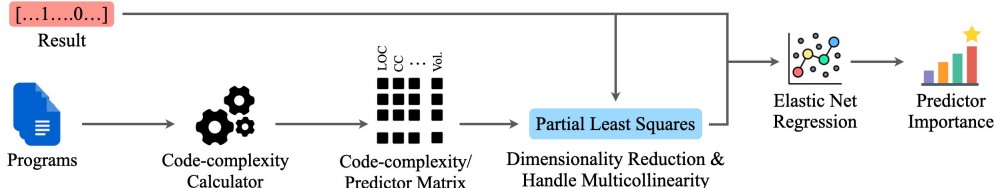

*(b)* Modeling LLM performance on C$^\star$ programs. We treat each LLM as a black box and apply **Elastic Net regression** using code-complexity metrics as predictors. **Partial Least Squares** (PLS) is employed for dimensionality reduction and to address multicollinearity. The magnitude and sign of the regression weights provide insight into the potential impact of each metric on the classifier's performance and hence to an extent the LLM's performance.

*Figure 8.* Analyzing the impact of different code-complexity metrics on LLM performance in the **PredState** task.

predictor matrix, together with the LLM's binary result vector of passes and fails, is then used to train a classifier.

*Table 14.* Odds ratio per interquartile range ($\Theta(\Delta)$) for each code-complexity metric for the **PredState** task **without semantics**. $\Theta(\Delta)$ for a metric is the odds ratio for a correct final-state prediction when that metric increases from its $25^{\text{th}}$ to its $75^{\text{th}}$ percentile, holding other metrics fixed. Reported only for models with <90% accuracy on the **PredState** task (Table 4, left panel) to mitigate class imbalance. The largest absolute values in each row are shown in boldface font.

| Models | Control-flow | | | Data-flow | | Size | | | |
|---|---|---|---|---|---|---|---|---|---|
| | $\Omega_{\text{CC}}$ | $\hat{\Omega}_{\text{If}}$ | $\hat{\Omega}_{\text{Loop}}$ | $\Omega_{\text{DD}}$ | $\hat{\Omega}_{\text{Assign}}$ | $\Omega_{\text{Loc}}$ | $\Omega_{\text{Vol}}$ | $\Omega_{\text{Voc}}$ | $\hat{\Omega}_{\text{Trace}}$ |
| **Human-Written** | | | | | | | | | |
| LLAMA-3.3 70B | -19 | -5 | **-29** | -17 | -2 | -16 | -22 | -25 | -1 |
| LLAMA-3.3 70B-CoT | -21 | -14 | **-28** | -16 | -2 | -17 | -19 | -20 | -1 |
| QWEN2.5-INST 14B | -17 | -5 | **-27** | -16 | -2 | -14 | -20 | -25 | -1 |
| QWEN2.5-INST 14B-CoT | -25 | -18 | **-27** | -15 | -3 | -20 | -21 | -20 | -2 |
| QWEN2.5-INST 32B | -12 | -11 | -12 | -9 | -1 | -12 | -14 | **-17** | -1 |
| QWEN2.5-INST 32B-CoT | -23 | -7 | **-33** | -17 | -4 | -19 | -21 | -20 | -2 |
| GPT-4o-mini | -18 | -7 | **-30** | -16 | -2 | -13 | -18 | -22 | -1 |
| GPT-4o-mini-CoT | -15 | -2 | **-28** | -14 | -2 | -11 | -15 | -16 | -1 |
| DS-QWEN 14B | -13 | -10 | **-16** | -9 | -2 | -11 | -13 | -10 | -1 |
| DS-LLAMA 70B | -14 | -5 | **-22** | -12 | -3 | -11 | -14 | -10 | -2 |
| **LLM-Translated** | | | | | | | | | |
| QWQ 32B | -1 | -5 | 5 | **-20** | -4 | -13 | -20 | -7 | -4 |
| **Fuzzer-Generated** | | | | | | | | | |
| QWQ 32B | -25 | -25 | -25 | -14 | **-33** | -25 | -24 | -28 | -31 |
| GPT-5-mini | -21 | -14 | -19 | -12 | **-27** | -20 | -20 | -21 | -27 |
| GEMINI-2.5-pro | -6 | -5 | -8 | -5 | **-12** | -6 | -6 | -5 | -12 |

Because these complexity metrics are often highly correlated (multicollinearity), we apply Partial Least Squares (PLS) (Wold et al., 2001) for dimensionality reduction. Unlike the unsupervised Principal Component Analysis (PCA) (Wold et al., 1987), which identifies linear combinations of predictors that maximize variance, PLS is supervised: it reduces dimensionality by finding components that maximize the covariance between predictors and the response variables (the result vector). This makes PLS more suitable in our setting, as it better mitigates multicollinearity while preserving predictive power.

We next apply Elastic Net regression (Zou & Hastie, 2005) on the PLS-transformed predictors and the result vector to train a classifier. In regression, each predictor is assigned a coefficient whose magnitude reflects its relative importance and whose sign indicates whether it contributes positively or negatively to prediction accuracy. Elastic Net is chosen because

*Table 15.* Odds ratio per interquartile range ($\Theta(\Delta)$) for each code-complexity metric for the **PredState** task **with the standard C⋆ semantics** ($\mathbb{K}$**-semantics and** $\mathbb{S}$). $\Theta(\Delta)$ for a metric is the odds ratio for a correct final-state prediction when that metric increases from its 25th to its 75th percentile, holding other metrics fixed. Reported only for models with <90% accuracy on the **PredState** task (Table 4, left panel) to mitigate class imbalance. The largest absolute values in each row are shown in boldface font.

| | Models | Control-flow | | | Data-flow | | Size | | | |
|---|---|---|---|---|---|---|---|---|---|---|
| | | $\Omega_{\text{CC}}$ | $\hat{\Omega}_{\text{If}}$ | $\hat{\Omega}_{\text{Loop}}$ | $\Omega_{\text{DD}}$ | $\hat{\Omega}_{\text{Assign}}$ | $\Omega_{\text{Loc}}$ | $\Omega_{\text{Vol}}$ | $\Omega_{\text{Voc}}$ | $\hat{\Omega}_{\text{Trace}}$ |
| | | | | | **Human-Written** | | | | | |
| $\mathbb{K}$ | LLAMA-3.3 70B | -25 | -4 | **-35** | -19 | -2 | -20 | -25 | -29 | -1 |
| | LLAMA-3.3 70B-CoT | -27 | -10 | **-33** | -20 | -3 | -24 | -26 | -25 | -2 |
| | QWEN2.5-INST 14B | -24 | 0 | **-39** | -22 | -2 | -19 | -26 | -28 | -1 |
| | QWEN2.5-INST 14B-CoT | -25 | -8 | **-35** | -16 | -3 | -20 | -22 | -22 | -2 |
| | QWEN2.5-INST 32B | -23 | -7 | **-35** | -19 | -2 | -19 | -25 | -30 | -1 |
| | QWEN2.5-INST 32B-CoT | -21 | -15 | **-27** | -12 | -3 | -16 | -17 | -16 | -2 |
| | GPT-4o-mini | -24 | -14 | **-32** | -21 | -2 | -22 | -27 | -30 | -1 |
| | GPT-4o-mini-CoT | -21 | -14 | **-26** | -15 | -3 | -18 | -20 | -19 | -2 |
| | DS-QWEN 14B | **-29** | -21 | -27 | -20 | -3 | -26 | -27 | -23 | -2 |
| | DS-LLAMA 70B | -26 | -14 | **-33** | -14 | -5 | -19 | -19 | -15 | -3 |
| $\mathbb{S}$ | LLAMA-3.3 70B | -24 | 0 | **-40** | -21 | -2 | -18 | -26 | -32 | -1 |
| | LLAMA-3.3 70B-CoT | -21 | -11 | **-32** | -16 | -4 | -18 | -20 | -21 | -2 |
| | QWEN2.5-INST 14B | -19 | 2 | **-39** | -19 | -2 | -13 | -21 | -25 | -1 |
| | QWEN2.5-INST 14B-CoT | **-26** | -17 | -25 | -16 | -3 | -22 | -22 | -20 | -2 |
| | QWEN2.5-INST 32B | -19 | -9 | **-28** | -16 | -1 | -17 | -21 | -27 | -1 |
| | QWEN2.5-INST 32B-CoT | -19 | -14 | **-22** | -12 | -2 | -17 | -17 | -18 | -1 |
| | GPT-4o-mini | -19 | 4 | **-37** | -19 | -2 | -16 | -23 | -29 | -1 |
| | GPT-4o-mini-CoT | **-15** | -9 | -14 | -7 | -2 | -12 | -12 | -12 | -1 |
| | DS-QWEN 14B | -11 | -4 | **-14** | -9 | -1 | -10 | -12 | -9 | -1 |
| | DS-LLAMA 70B | -23 | -12 | **-32** | -14 | -5 | -18 | -21 | -18 | -3 |
| | | | | | **LLM-Translated** | | | | | |
| $\mathbb{K}$ | QwQ 32B | -11 | -6 | 7 | -22 | 0 | -24 | **-27** | -8 | 0 |
| $\mathbb{S}$ | QwQ 32B | -14 | -9 | -6 | **-28** | -4 | -18 | -20 | -1 | -4 |
| | | | | | **Fuzzer-Generated** | | | | | |
| $\mathbb{K}$ | QwQ 32B | -21 | -27 | -25 | -10 | **-30** | -20 | -20 | -23 | -29 |
| | GPT-5-mini | -23 | -20 | -21 | -13 | **-31** | -22 | -22 | -23 | -30 |
| | GEMINI-2.5-pro | -14 | -10 | -14 | -8 | **-21** | -13 | -13 | -14 | -21 |
| $\mathbb{S}$ | QwQ 32B | -22 | -23 | -24 | -11 | **-31** | -22 | -21 | -25 | -30 |
| | GPT-5-mini | -22 | -19 | -21 | -11 | **-29** | -21 | -21 | -22 | -28 |
| | GEMINI-2.5-pro | -7 | -20 | -24 | -3 | -25 | -7 | -7 | -7 | **-26** |

it combines Lasso (Tibshirani, 1996) and Ridge (Hoerl & Kennard, 1970) regularization: the Lasso component drives irrelevant coefficients to zero, enabling feature selection, while the Ridge component shrinks correlated coefficients, thereby mitigating multicollinearity.

We now briefly describe the Elastic Net regression process to explain how we use the regression coefficients to determine the impact of different metrics. Let $n$, $p$, $\boldsymbol{y}$, and $\boldsymbol{X}$ be the total number of samples, the total number of predictors, the response vector, and the predictor matrix (we will use boldface font to denote vectors and matrices) respectively. Then,

$$\boldsymbol{y} \in \mathbb{R}^n, \quad y_i \in \{0, 1\}, \quad \boldsymbol{x_i} \in \mathbb{R}^p, \quad p_i(y_i = 1 | \boldsymbol{x_i}) = \frac{1}{1 + e^{-(\beta_0 + \boldsymbol{x_i}^\top \boldsymbol{\beta})}}$$

Where $p_i(y_i = 1 | \boldsymbol{x_i})$ along with $p_i(y_i = 0 | \boldsymbol{x_i}) = (1 - p_i(y_i = 1 | \boldsymbol{x_i}))$ represent the class-conditional probabilities and $\boldsymbol{\beta}$ is the vector of coefficients. The Elastic Net objective function for a Negative Log-Likelihood loss is given as (Friedman et al., 2010):

$$\arg\min_{\beta_0, \boldsymbol{\beta}} \left[ \frac{1}{n} \sum_{i=1}^{n} \left[ -y_i \log p_i - (1 - y_i) \log(1 - p_i) \right] + \lambda \underbrace{\sum_{j=1}^{p} \left[ \frac{1 - \alpha}{2} {\beta_j}^2 + \alpha |\beta_j| \right]}_{\text{Ridge and Lasso penalties}} \right]$$

Let $\hat{\boldsymbol{\beta}}$ be the coefficient vector that minimizes this objective function. Then the percentage odds ratio (Agresti, 2013; Cornfield, 1951; Harrell, 2015) $\Theta$ for the inter-quartile-range $\Delta_j$ of the $j$th predictor can be computed as:

$$\Theta(\Delta_j) = 100 \times \big( \exp\!\big(\hat{\beta}_j \, \Delta_j\big) - 1\big).$$

The percentage odds ratio per inter-quartile-range $\Theta(\Delta)$ gives the percentage change in the odds of the classifier's positive outcome (predicting a `1`) for the predictor ranging from its typical low value (25[th] percentile) to its typical high value (75[th] percentile) in the dataset when all other predictors are held constant. Thus if $\Theta(\Delta_j)$ for the $j$[th] predictor is -37%, this implies that one quartile increase in the $j$[th] predictor lowers the odds of the classifier's positive outcome by 37%.

To quantify each metric's effect on accuracy, we report the odds-ratio per interquartile range, $\Theta(\Delta)$, in Tables 14-15 for all LLMs without and with ($\mathbb{K}$-semantics, $\mathbb{S}$) semantics. Overall patterns are similar across settings. On the Human-Written dataset, $\Omega_{\text{Loop}}$ —the maximum executed loop-nesting depth—is the most influential predictor: larger $\Omega_{\text{Loop}}$ is associated with lower odds of a correct final-state prediction. On the LLM-Translated dataset, $\Omega_{\text{DD}}$ (data-flow complexity) and $\Omega_{\text{Vol}}$ (size) dominate without semantics; with semantics, $\Omega_{\text{DD}}$ remains dominant under $\mathbb{S}$, whereas $\Omega_{\text{Vol}}$ dominates under $\mathbb{K}$-semantics. On the Fuzzer-Generated split, $\hat{\Omega}_{\text{Assign}}$ (total variable assignments) is the strongest predictor both without and with semantics, with one exception: for GEMINI-2.5-pro under $\mathbb{S}$, $\hat{\Omega}_{\text{Trace}}$ (execution-trace length) is most predictive. Collectively, these $\Theta(\Delta)$ trends suggest that increasing control-flow depth harms models on human code, whereas data-flow/size factors are more limiting on translated or fuzzer generated code.

### G.2.2. COMPLEXITY-METRIC IMPACT PATTERNS

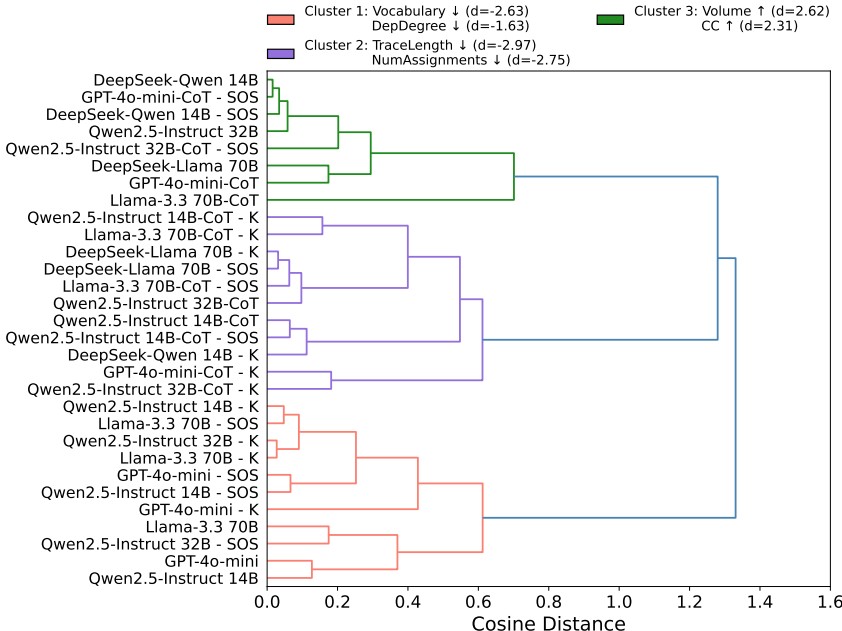

*Figure 9.* Dendrogram of models for the **PredState** task on the Human-Written dataset under no-semantics and standard semantics ($\mathbb{K}$-semantics and $\mathbb{S}$). We show the top two most distinguishable metrics per cluster, identified using the Cohen's d one-vs-rest test. The silhouette score is 0.58 thus indicating a good clustering structure.

To identify if there is a pattern to how models perform on increasing different code-complexity metrics, we perform hierarchical clustering on the standardized regression coefficients ($\hat{\beta}_{\text{SD}}$) of the metrics for the models on the Human-Written dataset. We perform this for the no-semantics and with standard semantics ($\mathbb{K}$-semantics and $\mathbb{S}$) cases. We use the cosine-distance as the pair-wise distance metric and the Cohen's d one-vs-rest test to identify the most distinguishing metric of each cluster. Figure 9 shows the dendrogram of the clustering process.

We see that there are three clusters. All the non-reasoning models without CoT prompting are in *Cluster 1* with the exception of QWEN2.5-INST 32B (under no-semantics case). Cluster 1 responds more negatively to increases in the complexity metrics Vocabulary ($\Omega_{\text{Voc}}$) and DepDegree ($\Omega_{\text{DD}}$) relative to the other two clusters. *Cluster 2* contains only the reasoning models and the non-reasoning models with CoT prompting. It predominantly contains models under the $\mathbb{K}$-semantics and responds more negatively to the dynamically computed metrics, TraceLength ($\hat{\Omega}_{\text{Trace}}$) and NumAssignments ($\hat{\Omega}_{\text{Assign}}$) relative to the rest of the clusters. The last cluster, *Cluster 3* also only contains reasoning models and non-reasoning models with CoT prompting (QWEN2.5-INST 32B is an exception). It predominantly contains models under $\mathbb{S}$ semantics and responds positively to

increases in the metrics, Volume ($\Omega_{\text{Vol}}$) and cyclomatic-code complexity ($\Omega_{\text{CC}}$) relative to the rest.

### G.2.3. AVERAGE PERCENTAGE OF VARIABLES PREDICTED CORRECTLY

*Table 16.* Average percentage of variables predicted correctly per program on the **PredState** task.

| | Models | $\text{Acc}_{\text{na}}$ | $\mathbb{K}$-Formalization | | | $\mathbb{S}$-Formalization | | |
|---|---|---|---|---|---|---|---|---|
| | | | $\text{Acc}_{\text{std}}$ | $\text{Acc}_{\text{swap}}$ | $\text{Acc}_{\text{obf}}$ | $\text{Acc}_{\text{std}}$ | $\text{Acc}_{\text{swap}}$ | $\text{Acc}_{\text{obf}}$ |
| | **Human-Written** | | | | | | | |
| Non-reasoning | QWEN2.5-INST 14B | 70 | 67 | 37 | 53 | 67 | 33 | 50 |
| | QWEN2.5-INST 14B-CoT | 85 | 83 | 36 | 75 | 82 | 35 | 63 |
| | QWEN2.5-INST 32B | 77 | 69 | 32 | 53 | 71 | 32 | 55 |
| | QWEN2.5-INST 32B-CoT | 90 | 89 | 39 | 78 | 84 | 33 | 65 |
| | LLAMA-3.3 70B | 70 | 66 | 38 | 52 | 64 | 34 | 52 |
| | LLAMA-3.3 70B-CoT | 87 | 86 | 33 | 78 | 86 | 28 | 66 |
| | GPT-4o-mini | 67 | 64 | 38 | 47 | 61 | 38 | 41 |
| | GPT-4o-mini-CoT | 75 | 89 | 30 | 62 | 82 | 31 | 54 |
| Reasoning | DS-QWEN 14B | 66 | 83 | 27 | 53 | 60 | 20 | 43 |
| | DS-QWEN 32B | 85 | 97 | 45 | 85 | 98 | 36 | 88 |
| | DS-LLAMA 70B | 81 | 92 | 33 | 73 | 90 | 34 | 65 |
| | QwQ 32B | 94 | 99 | 82 | 91 | 100 | 38 | 92 |
| | o3-mini | 95 | **100** | 59 | 92 | **100** | 74 | 98 |
| | GPT-5-mini | **100** | 100 | 86 | **97** | 100 | 85 | 99 |
| | GEMINI-2.5-pro | 93 | 100 | **98** | 97 | 100 | **99** | **100** |
| | **LLM-Translated** | | | | | | | |
| | QwQ 32B | 90 | 96 | 66 | 86 | 95 | 45 | 87 |
| | GPT-5-mini | **98** | 98 | 88 | 96 | **98** | 81 | 97 |
| | GEMINI-2.5-pro | 96 | **98** | **95** | 96 | 98 | **96** | 97 |
| | **Fuzzer-Generated** | | | | | | | |
| | QwQ 32B | 65 | 70 | 7 | 22 | 69 | 0 | 17 |
| | GPT-5-mini | 91 | 82 | 22 | 33 | 84 | 33 | 34 |
| | GEMINI-2.5-pro | **96** | **94** | **53** | **85** | **95** | **71** | **82** |

We also computed on average (over the total number of declared variables per $C^\star$ program followed by over the total number of $C^\star$ programs) how many of the final-states of the declared variables per $C^\star$ program that are assigned to at least once are being predicted correctly by the models. The results are shown in Table 16. We see that the trend in terms of models performing better without semantics than with semantics is similar to what is observed in the **PredState** task (Table 4, left panel). We also see that although models perform very poorly on the increasingly complex datasets such as the Fuzzer-Generated dataset on the **PredState** task, the average percentage of the final-states of the variables predicted correctly per program is quite high.

### G.2.4. STANDARD DEVIATION OF TASK ACCURACY

*Table 17.* Standard deviation of the accuracy metrics for the **PredState** task.

| | $\text{Acc}_{\text{na}}$ | $\mathbb{K}$-Formalization | | | $\mathbb{S}$-Formalization | | |
|---|---|---|---|---|---|---|---|
| | | $\text{Acc}_{\text{std}}$ | $\text{Acc}_{\text{swap}}$ | $\text{Acc}_{\text{obf}}$ | $\text{Acc}_{\text{std}}$ | $\text{Acc}_{\text{swap}}$ | $\text{Acc}_{\text{obf}}$ |
| *Human-Written* | | | | | | | |
| DS-QWEN 14B | 2.3 | 2.0 | 0.3 | 1.7 | 1.8 | 0.0 | 2.5 |
| DS-QWEN 32B | 1.2 | 2.3 | 5.2 | 0.5 | 1.2 | 0.3 | 1.5 |
| DS-LLAMA 70B | 0.0 | 0.0 | 0.2 | 0.0 | 0.6 | 0.0 | 1.3 |
| QwQ 32B | 0.2 | 0.3 | 0.0 | 0.8 | 0.3 | 1.5 | 2.0 |
| o3-mini | 1.2 | 0.0 | 1.4 | 0.6 | 0.0 | 1.8 | 0.3 |
| GPT-5-mini | 0.0 | 0.5 | 2.6 | 0.3 | 0.3 | 0.8 | 0.0 |
| GEMINI-2.5-pro | 2.0 | 0.3 | 0.0 | 0.9 | 0.5 | 1.0 | 0.3 |
| *LLM-Translated* | | | | | | | |
| QwQ 32B | 1.8 | 1.5 | 1.7 | 1.7 | 1.5 | 0.3 | 3.6 |
| GPT-5-mini | 1.0 | 1.3 | 2.0 | 0.8 | 0.3 | 0.8 | 0.3 |
| GEMINI-2.5-pro | 1.5 | 0.3 | 1.6 | 0.3 | 1.0 | 2.0 | 1.5 |
| *Fuzzer-Generated* | | | | | | | |
| QwQ 32B | 1.5 | 0.5 | 0.3 | 0.3 | 0.5 | 0.0 | 0.8 |
| GPT-5-mini | 0.9 | 1.0 | 1.1 | 1.2 | 1.0 | 1.7 | 0.8 |
| GEMINI-2.5-pro | 0.6 | 2.0 | 2.5 | 1.5 | 2.0 | 2.5 | 1.5 |

We average results over three independent runs for reasoning models and report the standard deviation of accuracy in Table 17. Across all model-dataset-semantic-variant combinations, the standard deviation never exceeds 5.2 percentage points and is typically below 2.0, confirming that the accuracy differences we report under semantic shifts and across code-complexity splits are well above run-to-run variability.

## G.3. Semantic-Rule Prediction (PredRule)

In this section, we discuss: (1) how the statements sampled from $C^\star$ programs are processed for the **PredRule** task, and (2) identify the most mispredicted rule (first-point-of-mismatch) categories in the **PredRule** task.

### G.3.1. PROCESSING $C^\star$ STATEMENTS FOR PREDRULE

*Table 18.* Processing of statements sampled from $C^\star$ programs for the **PredRule** task. The pair <Statement, State> is transformed into the pair <PredRule Program, PredRule State>. The transformed pair is used in constructing the prompt for the **PredRule** task.

| Type | Statement | State | PredRule Program | PredRule State |
|---|---|---|---|---|
| Declaration | `int ;` | $\sigma$ | `int ;` | $\sigma$ |
| Assignment | ` = <EXP>;` | $\sigma$ | ` = <EXP>;` | $\sigma$ |
| While | ```while(<PREDICATE>)`
`{`
`    <BODY>`
`};``` | $\sigma$ | ```while(<PREDICATE>)`
`{`
`-    <BODY>`
`+    halt;`
`};``` | $\sigma$ |
| If-else | ```if(<PREDICATE>)`
`{`
`    <BODY>`
`}`
`else`
`{`
`    <BODY>`
`};``` | $\sigma$ | ```if(<PREDICATE>)`
`{`
`-    <BODY>`
`+    halt;`
`}`
`else`
`{`
`-    <BODY>`
`+    halt;`
`};``` | $\sigma$ |
| Halt | `halt;` | $\sigma$ | `halt;` | $\sigma$ |
| Break | ```while(<PREDICATE>)`
`{`
`    ...`
`    break;`
`    ...`
`};``` | $\sigma$ | ```while(<PREDICATE>)`
`{`
`-    ...`
`    break;`
`    ...`
`};``` | $\sigma$ |

The objective of the **PredRule** task is to challenge LLMs with predicting the ordered sequence of semantic rules that is required to evaluate an $C^\star$ statement when the program state before the execution of that statement is given. Ideally, we want to avoid requiring the LLMs from needing to track program state since that capability is specifically tested for in the **PredTrace** task, and we want to avoid any overlaps/redundancies. This is trivial for statements that are self-contained, such as `declaration`, `assignment`, and `halt`. However statements such as `while`, `if-else`, `break`, require some processing to make them suitable for this task.

Table 18 shows how each type of statement is processed to make it suitable for the **PredRule** task. The primary objective behind processing is to make edits to the sampled statements such that they can be completely evaluated by requiring the least amount of program state updates. The first, second, and third columns lists the type of the sampled statement, its minimal representative skeleton, and the program state captured before its evaluation respectively. The fourth and the fifth columns list the sampled statement after processing and the corresponding processed program state which can now be used in the **PredRule** task. For the sampled `declaration`, `assignment`, and `halt` statements, the statements and the collected program state before their executions are used as is in the **PredRule** task because their evaluation does not require tracking program state nor do they require the execution of other statements. For `while` statements, we replace the body with a `halt` statement. This removes any possibility of needing state updates to correctly and completely evaluate the `while` statement. A similar approach is used for processing the `if-else` statement. For the `break` statement, we capture its closest enclosing loop and remove all statements from its body up until the `break` statement.

Since the **PredRule** task is scoped to a statement level of granularity, it is relatively agnostic to the complexity of the program as a whole.

## G.3.2. MOST MISPREDICTED RULES

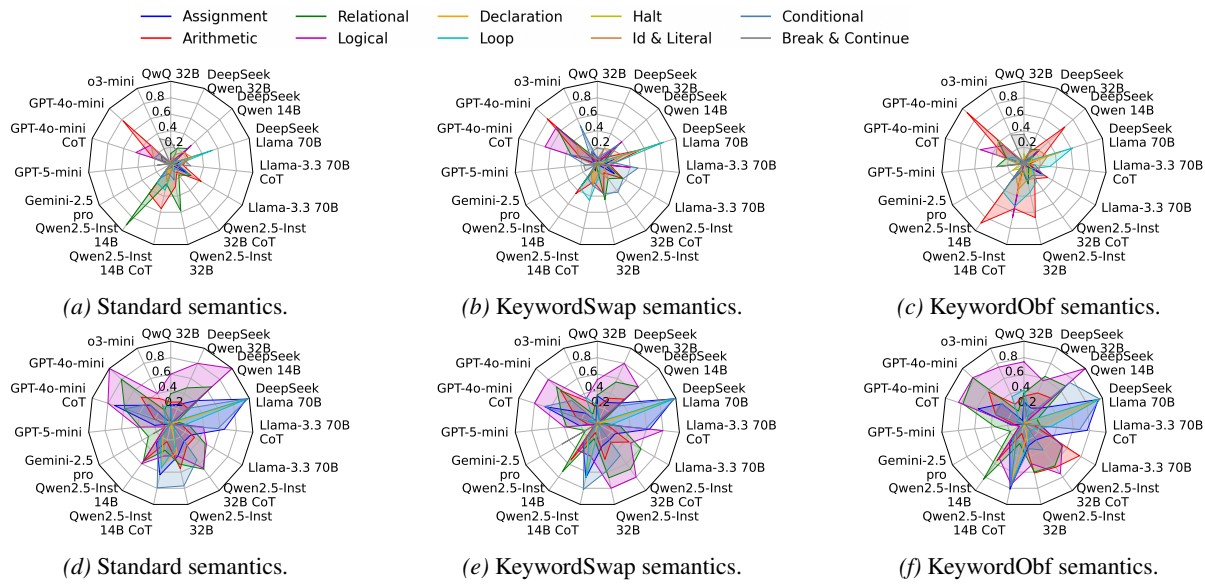

*(a)* Standard semantics.      *(b)* KeywordSwap semantics.      *(c)* KeywordObf semantics.

*(d)* Standard semantics.      *(e)* KeywordSwap semantics.      *(f)* KeywordObf semantics.

*Figure 10.* First-point-of-mismatch rate by category for the **PredRule** task with the $\mathbb{K}$-semantics (top) and $\mathbb{S}$ (bottom) on the Human-Written dataset.

To identify the semantic rules that models struggle with, we compute the *first-point-of-mismatch rate* for each rule, which is the frequency of the rule as the first mismatch between ground truth and the model prediction, relative to its total number of occurrences in the **PredRule** dataset. We group the rules into the following categories: *Assignment*, *Relational*, *Declaration*, *Halt*, *Conditional*, *Arithmetic*, *Logical*, *Loop*, *Id*, and *Break & Continue*. The mapping between the semantic rules and these categories for the $\mathbb{K}$-semantics and $\mathbb{S}$ is shown in Table 19.

The first-point-of-mismatch rate for a category is the maximum across all the rules within this category. Figure 10 shows the first-point-of-mismatch rate across categories for all the models on the Human-Written dataset for the standard and nonstandard semantics, for both their $\mathbb{K}$-semantics (top) and $\mathbb{S}$ (bottom) formalizations.

*Table 19.* Rule categorization for **PredRule** analysis.

| Category | $\mathbb{K}$ | $\mathbb{S}$ |
|---|---|---|
| Assignment | Rule 21 | Rules 4 - 6 |
| Arithmetic | Rules 3 - 11 | Rules 7 - 27 |
| Relational | Rules 12 - 17 | Rules 28 - 51 |
| Logical | Rules 18 - 20 | Rules 52 - 62 |
| Declaration | Rule 36 | Rule 3 |
| Loop | Rules 24 - 25 | Rules 67 - 70 & Rule 77 |
| Break & Continue | Rules 27 - 35 | Rules 71 - 76 |
| Halt | Rule 26 | Rule 78 |
| Id | Rules 1 - 2 | Rules 1 - 2 |
| Conditional | Rules 22 - 23 | Rules 64 - 66 |

Firstly, we observe that the models in general mispredict rules to a larger extent for $\mathbb{S}$ relative to when provided with the $\mathbb{K}$-semantics. Furthermore, categories such as `Declaration`, `Id & Literal`, and `Halt` that generally require one or at most two rules are almost never mispredicted significantly by any model across all the different cases. This is also observed for the `Assignment` category under $\mathbb{K}$-semantics which is formalized by just one rule and we see that its misprediction rate is low across models. In contrast, the `Assignment` category is heavily mispredicted under $\mathbb{S}$ formalization for standard and nonstandard semantics for a large number of models. We see a similar story with the `Logical` category where models mispredict it more significantly under $\mathbb{S}$ than $\mathbb{K}$-semantics. The `Logical` category contains the three logical operators (`AND`, `OR`, and `NOT`) and we see that exactly three rules are required under $\mathbb{K}$-semantics thus one rule per operator whereas $\mathbb{S}$ requires ten rules, almost 4x more rules per operator than $\mathbb{K}$-semantics. Similar trends are observed in the `Relational` category.

# H. Use of External Assets

In this work, we make use of several external assets, including datasets, and pretrained models. We acknowledge and credit the original creators of these assets as follows:

## H.1. Data

We construct the Human-Written dataset by rewriting the existing code solutions from the following sources:

1. HumanEval-X

   (a) License: Apache 2.0
   (b) URL: https://huggingface.co/datasets/THUDM/humaneval-x

2. BabelCode MBPP

   (a) License: CC 4.0
   (b) URL: https://huggingface.co/datasets/gabeorlanski/bc-mbpp

3. CodeContests

   (a) License: CC 4.0
   (b) URL: https://github.com/google-deepmind/code_contests

4. Leetcode

   (a) We scrape only the ground-truth solutions and public test cases from leetcode. We use the collected problems for academic purposes only.
   (b) URL: https://leetcode.com/

We construct the LLM-Translated dataset by using QWEN2.5-INST 32B to translate the C++ solutions to problems from:

1. CodeForces

   (a) License: CC 4.0
   (b) URL: https://huggingface.co/datasets/open-r1/codeforces

## H.2. Models

We evaluate LLMs designed for coding tasks and enhanced reasoning ability on our PLSEMANTICSBENCH:

1. LLAMA-3.3 70B (Grattafiori et al., 2024),

   (a) License: llama3.3
   (b) URL:
       https://huggingface.co/meta-llama/Llama-3.3-70B-Instruct

2. Qwen2.5-Coder Models (Hui et al., 2024),

   (a) License: Apache 2.0
   (b) URLs:
       https://huggingface.co/Qwen/Qwen2.5-Coder-32B-Instruct
       https://huggingface.co/Qwen/Qwen2.5-Coder-14B-Instruct

3. DeepSeek-R1 distilled models (Guo et al., 2025)

   (a) License: MIT
   (b) URLs:
       https://huggingface.co/deepseek-ai/DeepSeek-R1-Distill-Llama-70B
       https://huggingface.co/deepseek-ai/DeepSeek-R1-Distill-Qwen-32B
       https://huggingface.co/deepseek-ai/DeepSeek-R1-Distill-Qwen-14B

4. QwQ 32B (Team, 2025b)

   (a) License: Apache 2.0
   (b) URL: https://huggingface.co/Qwen/QwQ-32B

5. GEMINI-2.5-pro. In this study, we utilized the GEMINI-2.5-pro model provided by Google AI. The use of this model is subject to the Generative AI Preview Terms and Conditions, as outlined in the Google Cloud Service Specific Terms for Pre-GA Offerings.

    (a) URL: https://cloud.google.com/terms/service-terms

6. OpenAI Models. In this study, the use of OpenAI's models is subject to the term of use.

    (a) URL: https://openai.com/policies/row-terms-of-use/

### H.3. Icons

We use several icons from https://www.flaticon.com which we attribute here.

- Document icons created by Roman Káčerek - https://www.flaticon.com/free-icons/document

- Robot icons created by Kiranshastry - https://www.flaticon.com/free-icons/robot

- Xml icons created by Dimitry Miroliubov - https://www.flaticon.com/free-icons/xml

- Diff icons created by brajaomar_j - https://www.flaticon.com/free-icons/diff

- Matrix icons created by meaicon - https://www.flaticon.com/free-icons/matrix

- Logistic regression icons created by raidolicon - https://www.flaticon.com/free-icons/logistic-regression

- Game chart icons created by Arslan Haider - https://www.flaticon.com/free-icons/game-chart

- Gears icons created by sonnycandra - https://www.flaticon.com/free-icons/gears

- Message icons created by Freepik - https://www.flaticon.com/free-icons/message

```
1   int L;
2   int p;
3   int y;
4   int d;
5   int K;
6   int h;
7   int T;
8   int Y;
9   int ble0;
10  int ble1;
11  int ble2;
12  int ble3;
13  int ble4;
14  int ble5;
15  int ble6;
16  int ble7;
17  int ble8;
18  int ble9;
19  int ble10;
20  int ble11;
21  int ble12;
22  int ble13;
23  int ble14;
24  int ble15;
25  int ble16;
26  int ble17;
27  int ble18;
28  int ble19;
29  int ble20;
30  int ble21;
31  int ble22;
32  int ble23;
33  int ble24;
34  int ble25;
35  int ble26;
36  int ble27;
37  int ble28;
38  int ble29;
```

```
39   int ble30;
40   int ble31;
41   int ble32;
42   int ble33;
43   int ble34;
44   int ble35;
45   int ble36;
46   int ble37;
47   int ble38;
48   L = (((- y) / 4) - p);
49   T = ((((3 + K) + 1) - 3) + (8 / 8));
50   L = ((((((- p) % 1) * 7) - (- 1)) - (- 9)) - 3);
51   K = (((((d * (- 5)) + y) + (- 5)) - (- K)) - (- 5));
52   Y = (((9 + 3) - T) + 5);
53   K = ((7 / 7) * L);
54   y = ((((L + L) + 8) + 3) + 1);
55   p = ((((- 8) * 9) - ((- 6) % (- 8))) - T);
56   ble0 = (- 1);
57   ble1 = (- 1);
58   ble2 = (- 1);
59   ble3 = (- 1);
60   ble4 = (- 1);
61   ble5 = (- 1);
62   ble6 = (- 1);
63   ble7 = (- 1);
64   ble8 = (- 1);
65   ble9 = (- 1);
66   ble10 = (- 1);
67   ble11 = (- 1);
68   ble12 = (- 1);
69   ble13 = (- 1);
70   ble14 = (- 1);
71   ble15 = (- 1);
72   ble16 = (- 1);
73   ble17 = (- 1);
74   ble18 = (- 1);
75   ble19 = (- 1);
76   ble20 = (- 1);
77   ble21 = (- 1);
78   ble22 = (- 1);
79   ble23 = (- 1);
80   ble24 = (- 1);
81   ble25 = (- 1);
82   ble26 = (- 1);
83   ble27 = (- 1);
84   ble28 = (- 1);
85   ble29 = (- 1);
86   ble30 = (- 1);
87   ble31 = (- 1);
88   ble32 = (- 1);
89   ble33 = (- 1);
90   ble34 = (- 1);
91   ble35 = (- 1);
92   ble36 = (- 1);
93   ble37 = (- 1);
94   ble38 = (- 1);
95   if(((((y - K) - 2) == ((y % 8) % 5)) || ((L + y) < ((0 / 1) * (- K)))))
96   {
97      while((((((7 % 6) % 2) > (h + (- d))) || ((T % 1) != ((T * d) * (- 3)))) && (ble0 < 0))
98      {
99         while((((y + (- K)) <= (((- 1) + p) + L)) || (((p - (- L)) - 9) > (T - p))) && (ble1 <= 5))
100        {
101           h = ((h - (4 % 2)) + (h * K));
102           ble1 = (ble1 + 2);
103        };
104        while(((((h - p) + 2) > ((9 * h) % 3)) || (((K + 4) - L) <= ((0 - h) + Y))) && (ble2 < 20))
105        {
106           while((((!(((0 - d) + Y) != (h + L))) || ((d + (- d)) >= ((p - 1) - p))) && (ble3 < 15))
107           {
108              T = (((((1 + (- Y)) - 0) + h) - 4) - 1);
109              ble3 = (ble3 + 5);
110           };
111           Y = (1 + (5 / 6));
112           while((((!((d - L) <= ((1 * p) + L))) || ((h - K) >= ((9 - d) - (- h)))) && (ble4 < 9))
113           {
114              if(((!(((- T) - ((- Y) % 8)) == (L + T))) || (((4 - y) + p) > (p * Y))))
115              {
116                 T = ((9 - Y) + (p % 1));
117                 while((((((- L) + y) <= ((6 * h) - K)) || ((1 + (Y % 9)) != ((p * y) + (- 7)))) && (ble5 <= 17))
118                 {
119                    K = ((9 + 9) + (5 * 9));
```

```
120                 ble5 = (ble5 + 3);
121             };
122             T = (((5 - 5) - 1) - (3 * 1));
123         }
124         else
125         {
126             p = ((7 / (- 3)) + 8);
127             break;
128         };
129         if(((d - L) < ((5 % 9) % 1)) && (!((T * K) <= (Y - K))))
130         {
131             while((((T / 4) > ((- Y) - T)) && (((- 4) - ((- y) * (- T))) < (K + (3 / 3)))) && (ble6 < 13))
132             {
133                 Y = (7 + (Y / 9));
134                 ble6 = (ble6 + 1);
135             };
136             if((((d + h) - 2) <= ((L - T) + 8)) && (((7 + (- h)) - h) == (L + T)))
137             {
138                 while(((!(((((- y) - p) + 7) <= (d + d))) && ((Y + y) > (Y + h))) && (ble7 > (- 12)))
139                 {
140                     break;
141                     ble7 = (ble7 + (- 3));
142                 };
143             }
144             else
145             {
146                 h = ((8 - ((- 0) / 4)) + 2);
147             };
148             d = (((2 + (Y % 4)) - 8) - K);
149         }
150         else
151         {
152             d = (((Y + Y) - L) - 2);
153         };
154         while((((T / (- 6)) < (p % 6)) || (((T + d) - 9) == ((9 + K) + h))) && (ble8 > (- 20)))
155         {
156             while((((T - d) > (T + p)) && (((h - 9) + p) == ((p + 1) - p))) && (ble9 > (- 15)))
157             {
158                 p = (((8 / (- 6)) + 0) + (4 / 8));
159                 K = ((((d % 5) + 7) + (9 / 2)) - 7);
160                 ble9 = (ble9 + (- 4));
161             };
162             ble8 = (ble8 + (- 1));
163         };
164         ble4 = (ble4 + 3);
165     };
166     ble2 = (ble2 + 6);
167     };
168     ble0 = (ble0 + 2);
169     };
170 }
171 else
172 {
173     p = (((L - y) - 4) + 5);
174     while((((p + p) <= (d + h)) && (((L - L) - (- 1)) <= (((- h) + 7) - p))) && (ble10 <= 20))
175     {
176         while(((((y / 7) >= ((5 - p) + (- Y))) || ((Y + (- Y)) >= ((4 * d) + (- Y)))) && (ble11 >= (- 9)))
177         {
178             if(((d - y) >= (h - K)) && ((T * (- T)) != ((- T) + (- Y))))
179             {
180                 break;
181                 break;
182                 if(((((- Y) % 9) > (p + y)) && (!(((3 % 6) + y) != (L + K))))
183                 {
184                     break;
185                     K = (6 - ((- 6) % 5));
186                     if((((K + 6) + L) <= ((4 / 8) + Y)) || ((T * T) > (y + K)))
187                     {
188                         L = (d + (3 * 6));
189                         while(((((T - L) + 2) > (Y + T)) || (!((K + h) <= ((d + y) - 5)))) && (ble12 < 11))
190                         {
191                             y = ((Y * h) - ((3 * 8) / 8));
192                             break;
193                             while(((((3 + d) - y) != (p / (- 6))) && (((- T) - h) >= (L / 1))) && (ble13 >= (- 17)))
194                             {
195                                 L = (((Y / 2) * T) + (8 % (- 9)));
196                                 y = ((- y) + ((- p) * T));
197                                 p = (((- 6) + (8 * 5)) - 8);
198                                 ble13 = (ble13 + (- 3));
199                             };
200                             ble12 = (ble12 + 3);
```

```
201              };
202              p = ((((K * (- 6)) * 3) - 2) + 7);
203          }
204          else
205          {
206              Y = (((8 % 4) - (p * 4)) - 8);
207              if(((((d + (- d)) + 0) == (((- d) / 4) * K)) || ((Y * y) >= ((1 % 8) + (- L))))
208              {
209                  y = (((5 * p) + T) - d);
210                  p = ((0 * 0) - (((0 / 3) % 1) / 9));
211              }
212              else
213              {
214                  while((((K - (d * 2)) != (p / 2)) || (((L / 9) - y) < (Y - T))) && (ble14 < 20))
215                  {
216                      y = ((p - (0 * (- h))) + L);
217                      p = (((((- 4) - 6) - y) + L) + T);
218                      break;
219                      ble14 = (ble14 + 1);
220                  };
221                  h = ((T - 4) + 9);
222                  T = (((((- 5) - 3) + 2) - 1) + 8);
223              };
224              y = ((((2 + (9 / 9)) + 4) - (- 0)) + 1);
225          };
226      }
227      else
228      {
229          break;
230      };
231  }
232  else
233  {
234      while((((((- h) + 4) + K) <= (h - Y)) || (((6 * p) + d) < (T / (- 2)))) && (ble15 < 9))
235      {
236          L = (Y - (((d * h) / (- 8)) % 1));
237          K = (((((- 4) * 9) + (7 * 2)) + 1) + 1);
238          d = (((6 - (L * 5)) - 0) + 8);
239          ble15 = (ble15 + 3);
240      };
241      while((((!(((- 1) * p) - y) != (y - h))) || ((((- 4) / (- 5)) + d) <= (y + (- h)))) && (ble16 > (-
                7)))
242      {
243          break;
244          K = (((d * h) + 5) - 7);
245          while((((d % (- 1)) <= ((- p) * K)) && ((h - y) == (p - h))) && (ble17 > (- 13)))
246          {
247              T = ((((((0 - T) + 0) + 6) + 2) - 2);
248              break;
249              break;
250              ble17 = (ble17 + (- 3));
251          };
252          ble16 = (ble16 + (- 2));
253      };
254      while((((Y + (T * 3)) == (((- d) - (- 6)) + p)) || (((L % 8) - L) == (T + h))) && (ble18 >= (- 12)))

255      {
256          d = (((6 % (- 6)) - (K * T)) + L);
257          if((((5 * d) + h) > ((L * 4) / 9)) && ((K + L) > (Y - y)))
258          {
259              K = ((8 + ((7 * 1) / 7)) - (- 3));
260              d = ((p - ((- 2) / 2)) - (1 * 6));
261          }
262          else
263          {
264              break;
265          };
266          if((!((0 - (d % 5)) != (L * d))) || ((d + p) >= ((8 + d) + (- K))))
267          {
268              d = (((K / 8) % 6) - (T % (- 9)));
269              if(((h / 6) >= ((- K) / 1)) || (((h - d) + (- 7)) >= ((y + 1) - h)))
270              {
271                  Y = ((((T - (p * T)) - 2) - 4) + 4);
272              }
273              else
274              {
275                  K = (6 + ((- 0) % 2));
276                  h = (((8 + T) + 8) - 2);
277              };
278          }
279          else
```

```
280                         {
281                             if(((((- 3) + h) - p) < ((9 + d) + L)) && ((K + d) > (d * y)))
282                             {
283                                 if(((h / 6) < ((y - d) - 4)) || ((h + L) != (y - (T / 3))))
284                                 {
285                                     Y = ((- 8) + ((- 5) % 7));
286                                 }
287                                 else
288                                 {
289                                     d = (((6 + 8) + 4) - (- 6));
290                                 };
291                                 while((((p + (T % 4)) <= ((6 + L) - p)) || (((0 * p) - (- K)) >= (((- 1) + (- K)) + Y)))
292                                     && (ble19 < 12))
293                                 {
294                                     break;
295                                     T = (Y + (T / 5));
296                                     ble19 = (ble19 + 1);
297                                 };
298                                 Y = ((L - (5 % 8)) + ((T % 1) % 2));
299                             }
300                             else
301                             {
302                                 while((((p * d) != (((- L) + T) + 9)) && (!(((8 + d) - y) < (p + y)))) && (ble20 <= 18))
303                                 {
304                                     if(((d - (3 * L)) < ((p + d) - 6)) || ((T - d) <= ((T % 1) + (- L))))
305                                     {
306                                         break;
307                                     }
308                                     else
309                                     {
310                                         break;
311                                         h = ((((5 + 9) - 1) - (4 / 3)) - 3);
312                                         L = (((p + (0 * 8)) + 0) + 8);
313                                     };
314                                     ble20 = (ble20 + 5);
315                                 };
316                             };
317                             ble18 = (ble18 + (- 3));
318                         };
319                     };
320             if(((Y * (- K)) == (T - h)) || (((L % 6) - (- K)) >= ((K / (- 2)) / 6)))
321             {
322                 h = ((((4 % 6) + (6 % 1)) + 3) + (- 1));
323                 while((((K + y) == ((- T) - (- L))) && ((Y / 9) != (((- p) * h) + 7))) && (ble21 >= (- 8)))
324                 {
325                     if((!((((4 % 6) % 6) >= (p / 8))) && (((2 + y) - K) >= ((1 + y) - (- y))))
326                     {
327                         T = ((((- L) * 1) - (5 * (- 8))) + 6);
328                     }
329                     else
330                     {
331                         h = (((((0 - y) + L) + 5) + T) + 5);
332                     };
333                     if(((Y * p) <= ((4 % 2) + T)) && ((d + L) >= ((y - (- 1)) + h)))
334                     {
335                         h = ((((6 + 2) - (- Y)) + y) - 1);
336                     }
337                     else
338                     {
339                         d = ((((L * (- 5)) - T) - (- L)) - (2 / 8));
340                         while(((((K + (- K)) + 8) != (d + h)) || ((Y * (- K)) < (T - (- K)))) && (ble22 > (- 20)))
341                         {
342                             while((((y % 3) >= ((6 - (- Y)) + T)) && ((y + (- K)) >= ((K + 8) + p))) && (ble23 < 18))
343                             {
344                                 K = (((4 % (- 9)) + (- K)) + y);
345                                 h = ((2 * 7) - (7 % 7));
346                                 break;
347                                 ble23 = (ble23 + 1);
348                             };
349                             y = (((T - 8) + ((9 % 3) % 1)) + (- 8));
350                             ble22 = (ble22 + (- 2));
351                         };
352                     };
353                     y = ((T + 2) - ((((- y) / 9) % 6) / (- 9)));
354                     ble21 = (ble21 + (- 3));
355                 };
356                 y = ((d - 4) - 7);
357             }
358             else
359             {
```

```
360          while((((y * K) != (L + (- Y))) || (((3 / 2) - p) > (K / 4))) && (ble24 >= (- 4)))
361          {
362             while((((d + Y) < (h - L)) || (((Y + (- h)) + 7) >= (T - d))) && (ble25 <= 5))
363             {
364                while((((K - T) <= (T - (- Y))) || ((y + d) >= ((p % 9) * T))) && (ble26 > (- 15)))
365                {
366                   break;
367                   ble26 = (ble26 + (- 2));
368                };
369                ble25 = (ble25 + 2);
370             };
371             d = (((((- L) - h) + K) + (3 * 2)) + 7);
372             while(((!(((T - L) + (- 5)) >= ((2 * (- K)) + T))) || (((8 + (- T)) - (- y)) <= (y / 4))) && (
                       ble27 > (- 19)))
373             {
374                while((((8 + (L * y)) >= (d / 8)) || (((y / 3) + T) < ((7 * h) + (- Y)))) && (ble28 > (- 9)))
375                {
376                   break;
377                   while(((((Y + (Y * (- 5))) >= ((p * (- h)) - 4)) && ((p % 3) > (Y - h))) && (ble29 >= (- 10)
                          ))
378                   {
379                      break;
380                      ble29 = (ble29 + (- 2));
381                   };
382                   ble28 = (ble28 + (- 1));
383                };
384                K = (((((p + d) + 0) + (- 9)) - 0) + 9);
385                if((((h * 2) + T) == (y - h)) || ((y % 7) < (L + p)))
386                {
387                   while(((((y - (Y % 5)) == ((L * (- L)) + 0)) || (((- 7) + (p * T)) > (L / 5))) && (ble30 >=
                          (- 2)))
388                   {
389                      h = ((3 + (- d)) - ((- T) * Y));
390                      T = (((((- 6) * L) - y) + 7) + 6);
391                      ble30 = (ble30 + (- 2));
392                   };
393                   if((((h * 5) % 7) <= ((d - (- Y)) - 4)) && ((L + Y) <= ((T * 5) % 4)))
394                   {
395                      Y = (((3 + 7) + 3) - 9);
396                      T = ((3 + L) - ((y % (- 6)) * (- h)));
397                      h = ((h * T) - (7 / 4));
398                   }
399                   else
400                   {
401                      Y = ((((0 - (- 8)) + 5) + 6) - (5 * 9));
402                      T = ((Y + (- 5)) + (- 6));
403                   };
404                }
405                else
406                {
407                   Y = (((3 - K) - Y) + (0 / 1));
408                   while((((L - h) < (y + y)) && ((p * y) == (((- h) * K) % (- 2)))) && (ble31 <= 17))
409                   {
410                      K = (((y * y) - d) + 7);
411                      while((((L / 7) != ((- p) / 6)) && (!(((4 - p) - T) == ((5 * K) / (- 2))))) && (ble32 >
                             (- 6)))
412                      {
413                         L = ((((- K) + (- 4)) - p) + (- d));
414                         d = ((y + 9) - 1);
415                         h = ((((K / 4) - 8) - (- 4)) + ((- 6) * 5));
416                         ble32 = (ble32 + (- 2));
417                      };
418                      K = (((6 + 6) - 6) + (p / 2));
419                      ble31 = (ble31 + 6);
420                   };
421                   while((((8 - (h * (- p))) != (((- Y) / 2) + d)) && (((h - T) - (- 6)) > (y * d))) && (
                          ble33 <= 0))
422                   {
423                      T = (((((- 2) + 7) + (- 9)) + 0) + 0);
424                      break;
425                      ble33 = (ble33 + 2);
426                   };
427                };
428                ble27 = (ble27 + (- 5));
429             };
430             ble24 = (ble24 + (- 2));
431          };
432       };
433       while(((((L * 2) - T) != (d + L)) || (((2 + (- T)) - d) == ((6 - h) - L))) && (ble34 > (- 7)))
434       {
435          d = ((K * Y) + L);
```

```
436              L = ((4 % 8) % 4);
437              ble34 = (ble34 + (- 1));
438          };
439          ble11 = (ble11 + (- 3));
440      };
441      T = ((((- 3) * (- K)) + 4) - (- 7));
442      h = (((((7 + 4) + 4) - (- 9)) + p) - 2);
443      ble10 = (ble10 + 5);
444  };
445  if((((Y * (- 5)) + L) > (((- 9) - Y) - T)) || (((L / 8) % 4) == (K - K)))
446  {
447      while((((!((h + T) != (K + ((- Y) * 7)))) || (!(((L / 9) - T) < ((Y + T) + (- 8))))) && (ble35 <= (- 3))))
448      {
449          Y = ((0 / 7) - 2);
450          Y = ((T + 2) + 8);
451          ble35 = (ble35 + 2);
452      };
453      while(((((3 + K) + L) != ((3 - h) + d)) && ((d + (- L)) > ((K - 8) - y))) && (ble36 < (- 1)))
454      {
455          if(((K + T) > (d + K)) || ((y - K) == ((L + 1) + p)))
456          {
457              L = (((8 * p) * p) * L);
458          }
459          else
460          {
461              if((!(((((- K) * T) + 8) != (L + K))) || (((6 % 1) - p) != ((y + 1) - K)))
462              {
463                  y = ((((Y - 0) + 5) - h) + (- L));
464                  break;
465                  while((((((- Y) - h) + 9) <= (Y * p)) || ((((- 6) - K) + Y) <= (Y - y))) && (ble37 < 20))
466                  {
467                      K = ((p % 7) + ((d / 6) * (- 3)));
468                      while((((y + (K * 7)) == ((- d) + T)) || ((h * h) >= (6 + (L * Y)))) && (ble38 > (- 18)))
469                      {
470                          Y = (3 - (L % 9));
471                          T = ((p + Y) + L);
472                          ble38 = (ble38 + (- 2));
473                      };
474                      K = ((p - Y) - (((- T) * 1) / 3));
475                      ble37 = (ble37 + 6);
476                  };
477              }
478              else
479              {
480                  h = ((9 + (- p)) + 8);
481                  y = ((((K * 7) % 6) / 4) + T);
482              };
483          };
484          h = (((L + (2 * 6)) - 1) + (- 3));
485          ble36 = (ble36 + 2);
486      };
487  }
488  else
489  {
490      Y = (((6 - p) - (4 * (- Y))) - L);
491  };
492  };
```

*Figure 11.* An example $C^\star$ program (`fuzz_100.imp`) from the Fuzzer-Generated dataset. Its code-complexity metric profile is: control-flow complexity ($\Omega_{CC} = 62$, $\Omega_{If} = 5$, $\Omega_{Loop} = 6$, $\hat{\Omega}_{If} = 3$, $\hat{\Omega}_{Loop} = 5$), data-flow complexity ($\Omega_{DD} = 2603$, $\hat{\Omega}_{Assign} = 86$), and program-size complexity ($\Omega_{Loc} = 492$, $\Omega_{Vol} = 37140$, $\Omega_{Voc} = 91$, $\hat{\Omega}_{Trace} = 249$). The GEMINI-2.5-pro model successfully predicted the final program-state of this program in the **PredState** task.

