# OpenReview forum: "LLMs Lean on Priors, Not Programming Language Semantics"
_ICML.cc/2026/Conference — ICML 2026 regular_

### Official Review · Reviewer_QTih · 2026-03-11

**Soundness:** 3
**Presentation:** 3
**Significance:** 2
**Originality:** 3
**Overall Recommendation:** 3
**Confidence:** 3

**Summary:**

This paper presents a benchmark intended to assess whether Large Language Models (LLMs) can condition on explicitly provided formal operational semantics, via execution traces and rules, rather than relying on pretrained lexical associations.
The authors generate controlled semantic alterations (e.g., swapping/obfuscating operator), and evaluate several tasks across multiple LLMs.

**Compliance With Llm Reviewing Policy:**

Affirmed.

**Final Justification:**

I thank the authors for the detailed rebuttal and for extending the experimental evaluation with more recent models. I appreciate the additional results and clarifications.

### 1. Experimental robustness and consistency (major concern)

A key issue is the lack of a consistent and clearly defined evaluation protocol across models.

From the paper, it appears that:
- Non-reasoning models are evaluated with temperature = 0,
- While reasoning models (and GPT-4o-mini) are averaged over three runs,
This raises a major concern:
- **Comparability**: Using different temperatures and evaluation procedures across models makes it difficult to draw fair comparisons between them.
Given that many conclusions rely on relatively fine-grained differences in accuracy, a unified evaluation protocol (e.g., consistent temperature across models and reporting of mean ± std) would be important to support the empirical claims.


### 2. Benchmark realism and significance
The motivation around semantic mutations (e.g., operator overloading, DSLs) is better articulated in the rebuttal. However, the connection to realistic downstream scenarios could still be strengthened. The benchmark appears primarily diagnostic, and it would be helpful to further clarify how the measured capability translates to practical settings.

### 3. Benchmark scope
The additional experiments with newer models are helpful and show that some settings remain challenging. At the same time, **some simpler settings appear close to saturation**, which may limit the headroom of the benchmark in its current form.

---
Overall, I acknowledge the strengths of the paper, including a clear and well-structured benchmark design and interesting empirical observations. However, the **concerns regarding experimental consistency and evaluation protocol reduce my confidence in the strength of the empirical conclusions**.

**Key Questions For Authors:**

1. Why not evaluate this work on with LiveCodeBench?
2. Why not compare to semantics-oriented or code models like SemCoder?
3. Why were more recent, and strongly performing coding models not included? (e.g. Qwen3, GPT-5.2)

**Limitations:**

Yes

**Strengths And Weaknesses:**

# Strengths

- Clear, well-motivated diagnostic goal.

- Benchmark design is structured with multiple tasks.

- Findings are plausibly useful to the program semantics community.

# Weaknesses

- Evaluation set feels outdated. All the LLMs chosen are not SOTA anymore, which weakens the paper’s impact.

- All the benchmarks might have been used in the training of the evaluated models (e.g., MBPP, HumanEval).

- Missing comparisons to relevant related work.

# Soundness

The core idea is seems sound.

Moreover, there is no threats-to-validity section. There should be one explaining the limitations of this work, like the scope of the C language used for this study, and the limitations of all the tools used.

# Significance

A clean semantics-controlled benchmark can be very useful for future research on program semantics, although this paper's significance is currently limited by non-SOTA models.
For instance, the authors Qwen2.5, however Qwen3 has been available since May 2025.

The paper should discuss and position itself relative to recent work on evaluating the robustness of LLMs against syntactic code mutations. There is some missing recent works:
- "What can Large Language Models Capture about Code Functional Equivalence?" https://arxiv.org/abs/2408.11081
- "Are Large Language Models Robust in Understanding Code Against Semantics-Preserving Mutations?" https://arxiv.org/abs/2505.10443
- "Data Augmentation by Program Transformation", https://doi.org/10.1016/j.jss.2022.111304

# Presentation

The manuscript is well-written, and the paper is easy to follow.

---

> ### Author Rebuttal · Authors · 2026-03-31
>
> ### Q1.
> > Why were more recent, strongly performing coding models not included?
> > (e.g., Qwen3, GPT-5.2)
>
> We have extended our evaluation with three additional frontier models spanning three distinct provider families: Qwen3-Coder-30B-A3B-Instruct (Alibaba), GPT-5.4-mini (OpenAI), and Claude Sonnet 4.6 (Anthropic, PredState only due to cost). Previously evaluated models are included for comparison.
>
> **PredState (Human-Written)**
>
> | Models | NA | S-Std | S-Swap | S-Obf | K-Std | K-Swap | K-Obf |
> |---|---|---|---|---|---|---|---|
> | Gemini-2.5-Pro | 93 | 99 | 98 | 100 | 100 | 97 | 94 |
> | GPT-5-mini | 100 | 99.8 | 78.6 | 98.7 | 99.4 | 78.8 | 93.6 |
> | Qwen2.5-Coder-32B | 50 | 33 | 4 | 19 | 29 | 4 | 12 |
> | GPT-5.4-mini (New) | 87 | 98.1 | 38.3 | 74.1 | 96.3 | 75.3 | 86.4 |
> | Qwen3-Coder-30B (New) | 41.4 | 29.6 | 4.3 | 15.4 | 30.2 | 3.7 | 14.2 |
> | Sonnet 4.6 (New) | 100 | 100 | 58.6 | 98.8 | 100 | 99.4 | 93.2 |
>
> **PredState (Fuzzer-Generated)** — Qwen models not applicable.
>
> | Models | NA | S-Std | S-Swap | S-Obf | K-Std | K-Swap | K-Obf |
> |---|---|---|---|---|---|---|---|
> | Gemini-2.5-Pro | 73 | 69 | 39 | 47 | 69 | 26 | 49 |
> | GPT-5-mini | 56.9 | 54.5 | 16.9 | 23.4 | 51.3 | 13.5 | 22.8 |
> | GPT-5.4-mini (New) | 34.5 | 30.3 | 11.5 | 4.8 | 33.3 | 7.8 | 4.8 |
> | Sonnet 4.6 (New) | 92.7 | 86.7 | 60 | 51.5 | 86.7 | 69.1 | 52.1 |
>
> - **KeywordSwap degradation persists across all new models and providers.** Sonnet 4.6: 100%→58.6% (S-Swap, 41-pt drop). GPT-5.4-mini: 98.1%→38.3% (S-Swap, 60-pt drop). Qwen3-Coder-30B: 29.6%→4.3%. This holds across three independent provider families, confirming reliance on pretrained symbol-semantics associations is a fundamental capability gap.
>
> - **Newer generations do not close the gap.** GPT-5.4-mini performs *worse* than GPT-5-mini under S-KeywordSwap (38.3% vs. 78.6%). Qwen3-Coder-30B shows no improvement over Qwen2.5-Coder-32B. Standard training advances do not automatically improve rule-conditioned reasoning — PLSemanticsBench measures a capability axis orthogonal to existing benchmarks.
>
> - **S vs K asymmetry emerges.** Sonnet 4.6 is robust to K-KeywordSwap (100%→99.4%) but drops sharply under S-KeywordSwap (100%→58.6%) on the same programs. This formalization-dependent behavior validates evaluating under both semantic frameworks.
>
> - **Structural complexity exposes further fragility**: even Sonnet 4.6 drops to 60% (S-Swap) and GPT-5.4-mini to 11.5% on Fuzzer-Generated. The benchmark remains far from saturated.
>
> **PredTrace**
>
> | Models | S-Std | S-Swap | S-Obf | K-Std | K-Swap | K-Obf |
> |---|---|---|---|---|---|---|
> | GPT-5-mini | 17 | 15 | 17 | 20 | 14 | 17 |
> | GPT-5.4-mini (New) | 17 | 12.3 | 8.6 | 20 | 14.2 | 19.8 |
>
> On **PredRule** (not shown), GPT-5.4-mini regresses relative to GPT-5-mini under S-semantics (75% vs 80%). Qwen3-Coder-30B shows near-identical performance across all S-semantic variants (22.3/22.3/23.1), suggesting it may not be conditioning on supplied S-rules at all. On **PredTrace**, GPT-5.4-mini achieves only 17–20% (standard), dropping to 8.6–14.2% under interventions. Long-horizon rule conditioning remains unsolved.
>
> These results confirm that rule-conditioned reasoning under semantic shift is a fundamental capability gap, robust across model generations and providers.
>
> ### Q2.
> > Why not evaluate with LiveCodeBench? Why not compare to SemCoder?
>
> LiveCodeBench evaluates code generation/test-passing, not formal-semantics-conditioned reasoning. SemCoder  (cited in the paper)  is in fact a key motivation for our work — see
> our response to `EWww Q2, Points 1–3`, where we position PLSemanticsBench
> as the missing diagnostic for the hypothesis that SemCoder and related
> work advance. We do not compare against it because fundamentally, we differ
> by centering on *rule-conditioned reasoning* with systematic semantic mutations,
> rather than incorporating semantic structure into training. Direct comparison is methodologically unsound: SemCoder uses informal operational semantics (natural language monologues) with arbitrary granularity and no mechanically verifiable ground truth, whereas PLSemanticsBench uses formal semantics with exactly one correct rule sequence per execution, enabling principled step-level metrics (PredRule, PredTrace).
>
> ### Q3.
> > Missing related work.
>
> We will incorporate all three: "What can LLMs Capture about Code Functional Equivalence?" (arXiv:2408.11081) — functional equivalence, complementary to our focus; "Are LLMs Robust Against Semantics-Preserving Mutations?" (arXiv:2505.10443) — perturbs surface-level program-semantics (we alter semantics of the programming language itself); "Data Augmentation by Program Transformation" (JSS 2022) — orthogonal training-data contribution.
>
> ### Q4.
> > No threats-to-validity section.
>
> We will add one discussing: scope limited to C* syntax; tokenization effects of KeywordObf (addressed in Appendix E); prompt sensitivity; reliance on K-framework for ground truth; and synthetic nature of fuzzer-generated programs.

---

> > ### Author Rebuttal · Reviewer_QTih · 2026-04-03
> >
> > Thank you for the detailed rebuttal and for extending the experimental evaluation with more recent models. I appreciate the additional results and clarifications.
> >
> > After carefully considering the rebuttal and re-examining the paper, I still have **some concerns that affect my overall assessment**.
> >
> > ### 1. Experimental robustness and consistency (major concern)
> >
> > A key issue is the lack of a consistent and clearly defined evaluation protocol across models.
> >
> > From the paper, it appears that:
> > - Non-reasoning models are evaluated with temperature = 0,
> > - While reasoning models (and GPT-4o-mini) are averaged over three runs,
> > This raises a major concern:
> > - **Comparability**: Using different temperatures and evaluation procedures across models makes it difficult to draw fair comparisons between them.
> > Given that many conclusions rely on relatively fine-grained differences in accuracy, a unified evaluation protocol (e.g., consistent temperature across models and reporting of mean ± std) would be important to support the empirical claims.
> >
> > ### 2. Integration of new results
> > The rebuttal introduces additional results with newer models, which is appreciated. However, these results are not fully integrated into the experimental section of the paper.
> > In particular, the evaluation setup is not consistently updated to include these models across all tasks and settings, and it remains unclear whether the same evaluation protocol is used for both original and newly introduced models.
> >
> > ### 3. Benchmark realism and significance
> > The motivation around semantic mutations (e.g., operator overloading, DSLs) is better articulated in the rebuttal. However, the connection to realistic downstream scenarios could still be strengthened. The benchmark appears primarily diagnostic, and it would be helpful to further clarify how the measured capability translates to practical settings.
> >
> > ### 4. Benchmark scope
> > The additional experiments with newer models are helpful and show that some settings remain challenging. At the same time, **some simpler settings appear close to saturation**, which may limit the headroom of the benchmark in its current form.
> >
> > ---
> > Overall, I acknowledge the strengths of the paper, including a clear and well-structured benchmark design and interesting empirical observations. However, the **concerns regarding experimental consistency and evaluation protocol reduce my confidence in the strength of the empirical conclusions**.
> >
> > Based on this, I am currently leaning toward lowering my score to **3 (weak reject)**.

---

> > > ### Author Response · Authors · 2026-04-03
> > >
> > > ### 1.
> > > > Non-reasoning models are evaluated with temperature = 0
> > >
> > >  - We already confirmed that the results are deterministic across two or more runs for non-reasoning models when the temperature is set to 0; hence, there is no need to average.
> > >  - We believe that deterministic results (when possible to enforce) are important for reporting on a benchmark, which is the reason we set the temperature to 0.
> > >
> > > > averaged over three runs... Using different temperatures and evaluation procedures across models...
> > >
> > > > Using different temperatures and evaluation procedures across models...
> > >
> > > - Several models (e.g., GPT-5 family, O3-mini family) unfortunately do not allow us to set temperature since it's an unsupported parameter. It is an API constraint outside our control. The only responsible alternative was to average over multiple trials.
> > > - The **per-model standard deviation** across all combinations (tasks x models x semantic-shifts) is within 0.052 and is
> > >   insignificant relative to the percentage-point changes we report with semantic/code structural complexity shifts.
> > >
> > >
> > > ### 2.
> > > > However, these results are not fully integrated into the experimental section of the paper.
> > >
> > > - ICML policy does **not** permit paper updates during the author-response period, so we were unable to incorporate them into the manuscript.
> > > - The updated version of the paper will include the results we shared with you in the rebuttal. Note: The new results reinforce every finding in the paper.
> > >
> > > > it remains unclear whether the same evaluation protocol is used for both original and newly introduced models.
> > >
> > > The same evaluation protocol is used for all models, old and new. Identical prompts, identical one-shot examples, identical parsing and scoring scripts, identical semantic variants.
> > >
> > > ### 3.
> > > > The benchmark appears primarily diagnostic, and it would be helpful to further clarify how the measured capability translates to practical settings
> > >
> > > PLSemanticsBench is the only benchmark to date that explicitly supplies formal semantics rules to models and evaluates their ability to condition their reasoning on the supplied rules and when those rules are changed to challenge their learned priors.
> > >
> > > ### 4.
> > > > some simpler settings appear close to saturation, which may limit the headroom of the benchmark in its current form.
> > >
> > > **Near-ceiling accuracy is reached only under standard semantics on structurally simple programs** — the easiest combination. Performance drops sharply along *either* axis of difficulty: semantic interventions or structural complexity, and collapses when both are applied. For instance, Sonnet 4.6 goes from 100% (Standard, Human-Written) to 58.6% under S-KeywordSwap alone, and to 60% under S-KeywordSwap on Fuzzer-Generated. GPT-5.4-mini drops from 98.1% to 38.3% under S-KeywordSwap on Human-Written, and to 11.5% on Fuzzer-Generated. On PredTrace, the best model achieves only 35%. The benchmark is far from saturated.
> > >
> > > **The near-ceiling standard settings are the control condition (line 308-317), not the benchmark.** They exist precisely to establish what a model can do when priors align with supplied rules. Without this baseline, performance under semantic/code structural complexity shifts has no reference for comparison.
> > >
> > >
> > > We hope that the reviewer will agree that we addressed all the points already.

---

### Official Review · Reviewer_K9gN · 2026-03-14

**Soundness:** 3
**Presentation:** 3
**Significance:** 3
**Originality:** 3
**Overall Recommendation:** 4
**Confidence:** 4

**Summary:**

This paper introduces a benchmark PLSemanticsBench to assess whether large language models can reason about program behaviour by grounding their reasoning in formal semantics rather than lexical cues. The authors define a simple imperative language C∗ and derive two interpreters: small‑step semantics S and a K‑framework semantics K. The authors evaluate 11 frontier LLMs in standard, chain‑of‑thought, and tool‑assisted modes. Results show that while some models achieve moderate accuracy on simple final‑state prediction, performance collapses under semantic mutations. Final‑state accuracy drops 40-60 %, and long‑horizon trace prediction yields near‑zero accuracy. The study concludes that current LLMs rely heavily on learned lexical associations and struggle to condition on formal rules.

**Compliance With Llm Reviewing Policy:**

Affirmed.

**Key Questions For Authors:**

Q1. Can you provide more concrete examples of programs and their mutated semantics to illustrate how models are expected to adapt?

Q2. How generalizable is the benchmark to more realistic languages? Could similar tasks be constructed for languages like Python or simple languages?

**Limitations:**

Yes

**Strengths And Weaknesses:**

+ The benchmark addresses a foundational question: can LLMs apply formal semantics rather than pattern matching? The severe performance drops highlight a gap that could motivate new research directions.
+ The tasks defined in this paper are carefully designed to isolate semantic reasoning. Mutated semantics and obfuscations break superficial word associations, forcing models to rely on formal rules.
+ The paper is clearly written by first stating hypotheses (rule composition, rule selection, long‑horizon reasoning, robustness under semantic shift) and then mapping tasks to these hypotheses, which helps the reader to understand the methodology.

- Some tasks (e.g., PredTrace) are extremely difficult for current models; the benchmark does not explore whether failure is due to task design or inherent model limitations.
- Examples of tasks and mutated programs are limited in the main text; including more illustrative examples in the appendix would help readers grasp the challenges.
- The work is diagnostic; it does not propose methods to improve semantic reasoning, and immediate practical impact is limited by the synthetic nature of the language.
- Extending the benchmark to real languages (with rather simple grammar, e.g., python)  is recommended for broader relevance.

---

> ### Author Rebuttal · Authors · 2026-03-31
>
> ### Q1.
> > Can you provide more concrete examples of programs and their
> > mutated semantics?
>
> Yes — we will add a dedicated "Worked Examples" subsection in the
> appendix showing:
>
> 1. a sample program under Standard, KeywordSwap, and KeywordObf semantics;
>
> 2. the corresponding execution traces under each; and
>
> 3. representative model predictions illustrating common failure modes.
>
> Space permitting, we will include a condensed example in the main text.
>
> ### Q2.
> > How generalizable is the benchmark to more realistic languages?
>
> The benchmark framework is language-agnostic by design. Extending to
> other languages requires:
>
> 1. formalizing the target language's semantics in S or K. The
>    K-framework already has formalizations of substantial subsets
>    of C (Ellison and Rosu, POPL 2012), Java (Bogdanas and Rosu, POPL 2015),
>    and Python (Rosu and Chen, CPP 2020);
>
> 2. constructing corresponding semantic interventions; and
>
> 3. generating ground-truth traces.
>
> The primary challenge for relatively more complex languages is ground-truth
> scale (longer traces, larger state) and fitting larger semantics formalization
> within a model's context window, not a methodological limitation.
> We view C* as the right starting point because it captures the
> essential foundations — complex control-flow and mutable state — of
> the imperative programming language paradigm and its formal tractability
> enables rigorous experimental control. We will add a discussion section on
> extension to other languages (including Python) as immediate future work.
>
> ### Q3.
> > The work is diagnostic; it does not propose methods to improve semantic
> > reasoning.
>
> We respectfully note that influential benchmarks such as CRUXEval (Gu et al., ICML 2024), CRUXEval-X (Xu et al., ACL 2025), LiveCodeBench (Jain et al., ICLR 2025),
> Bigcodebench (Zhuo et al., ICLR 2024), and SURGE (Lyu et al., EMNLP 2025) etc.,
> are also diagnostic, and their value lies in identifying capability gaps that
> motivate subsequent methodological advances.
>
> Our benchmark isolates a specific, previously unmeasured
> capability axis (rule-conditioned reasoning under formal semantics).
> Please see our response to `EWww-Q2` for our justification on the
> importance of measuring this capability axis and hence PLSemanticsBench.
>
> The fine-grained failure analysis (e.g., first-point-of-mismatch
> analysis in Appendix F.3.2) and performance analysis under semantics
> shifts provide actionable signals for future method development — for
> instance, the finding that assignment and relational rules under S-semantics
> are disproportionately mispredicted or that model performances are more
> adversely affected under KeywordSwap than under KeywordObf suggests
> targeted training interventions.

---

### Official Review · Reviewer_JQ7s · 2026-03-16

**Soundness:** 3
**Presentation:** 3
**Significance:** 2
**Originality:** 2
**Overall Recommendation:** 4
**Confidence:** 3

**Summary:**

The paper builds a benchmark to test whether LLMs are following formal semantics rules or just leaning on their prior knowledge of how code usually executes. They use a tiny C-like language with slightly specialized syntax (in BNF) and semantics (in Small-Step) and test LLM's capability on the tasks such as final-state prediction, rule prediction, and full execution traces, and then stress-test them with semantic mutations and more complicated programs. The evaluation shows that models often look strong on the normal setting, but become much less reliable once the semantics change or the reasoning required is more complicated.

**Compliance With Llm Reviewing Policy:**

Affirmed.

**Key Questions For Authors:**

Please see weaknesses.

**Limitations:**

- The syntax is still largely C-based and semantics are well-studied, LLMs may have be already saturated in this space.
- The altered semantics can hardly match with real-life use cases. Therefore the motivation is somewhat weak.

**Strengths And Weaknesses:**

Strengths:
- The formal model is interesting and sound
- The evaluation is cut straight to the point
- The result aligns with what people (including me) would expect
- The writing is strong, formalism is good (while arguably many more symbols are introduced and is somewhat complicated)

Weaknesses
- The main weakness for me is that the benchmark is more treated as a diagnostic stress test and weaker as a measure of code reasoning. The story depends on semantic mutation rules, where familiar operators are reassigned with new meanings. That is a clean way to distill pure reasoning capabilities from priors, but it is still fairly artificial, so the motivation needs to be extremely well thought-out.
- While I really like the PL-centric formalism (e.g., BNF, small-step, evaluation model) but the broader concept of obfuscating the semantic meaning to stress test LLM's reasoning capabilities have already been studied extensively in NLP domain. And with that, the result obtained from the evaluation is largely expected and unsurprising. I'm wondering if there is any better claim of contributions and novelty related to PL.
- I do think that even with the altered semantics and increased complexity, the LLMs are already performing quite well with the task.
- I am not fully convinced that the paper is cleanly disentangling structural complexity from bended semantics. From the look of evaluations I don't see the two concepts being separated and it's hard to interpret whether the lose in accuracy is coming more from the semantic rules or the increased complexity.

Minor
- Line 400, the formatting surrounding hyphens need to be unified.

---

> ### Author Rebuttal · Authors · 2026-03-31
>
> ### Q1.
> > The broader concept of obfuscating semantic meaning to stress test
> > LLMs has been studied extensively in NLP.
>
> PLSemanticsBench shares the motivation of exposing heuristic reliance but differs fundamentally from NLP perturbation work — CheckList (Ribeiro et al., ACL 2020), Contrast Sets (Gardner et al., EMNLP Findings 2020), HANS (McCoy et al., ACL 2019), Stress Tests for NLI (Naik et al., COLING 2018), Semantic Sensitivities (Arakelyan et al., EACL 2024):
>
> 1. **What is perturbed.** Prior NLP work perturbs *inputs* — modifying words or surface forms. PLSemanticsBench does not perturb programs; we perturb the *formal semantic rules* that define execution — programs remain syntactically identical while supplied semantics change. This tests a different capability — whether models can condition on *externally supplied formal specifications*. Code-domain works like EquiBench (Wei et al., EMNLP 2025) and SeqCoBench (Maveli et al., NAACL 2025), which perturb *input programs* under semantics-preserving mutations, are more analogous to NLP perturbation than PLSemanticsBench.
>
> 2. **Rules are explicitly supplied, not implicitly learned.** In NLP perturbation work, rules governing correct behavior are learned implicitly. In PLSemanticsBench, complete formal semantic rules are *explicitly provided* as mathematically specified transition rules, and we test whether models follow them. Our auxiliary NL→Rule and Rule→NL tasks confirm models understand the notation, so downstream failures reflect reasoning limitations, not comprehension difficulties.
>
> 3. **Connection to a specific research hypothesis.** PLSemanticsBench is the missing diagnostic for the hypothesis that models internalize formal semantics through execution-aware training (SemCoder, NExT, TRACED, CodeI/O, Jin & Rinard — see EWww Q2). Our semantic interventions directly test this. Operator meaning conflicts with priors arise frequently via *operator overloading* in C++, Python, etc. (see Q2).
>
> A closer NLP analogue would be explicitly redefining task inference rules at test time (e.g., "entails" now means "contradicts" in HANS) and supplying these rules to the model — rather than only perturbing input wording.
>
> ### Q2.
> > still fairly artificial, so the motivation needs to be extremely
> > well thought-out.
>
> Operators frequently have nonstandard meanings in practice:
>
> 1. **Operator overloading** in C++, Python, Scala, Haskell, Julia allows redefining existing operators or creating new symbols. E.g., Python's `__add__` can redefine `+` as subtraction: `class X: def __add__(self, o): return X(self.v - o.v)` so `X(10)+X(3)` yields 7. Widely used in scientific computing (PyLops, SoftwareX 2020) and DSL development (Rompf & Odersky, GPCE 2010).
> 2. **DSLs** where operators may have nonstandard semantics.
> 3. **Proof assistants** (Lean4, Coq) where LLMs must condition on user-defined nonstandard axioms.
>
> ### Q3.
> > the result obtained from the evaluation is largely expected and unsurprising
>
> Several findings are non-obvious (see also EWww Q1):
>
> - **KeywordSwap > KeywordObf drops** (Table 4): If models were pattern-matching on surface forms, KeywordObf (novel symbols) should be harder. The reverse reveals models actively *misapply* learned semantics.
> - **Formalization-dependent asymmetry**: Sonnet 4.6 (New) drops to 58.6% under S-KeywordSwap but stays at 99.4% under K-KeywordSwap on the same programs (see QTih Q1 for full results).
> - **CoT fails under KeywordSwap** (Table 4): CoT nearly triples non-reasoning model performance under Standard/KeywordObf but has zero effect under KeywordSwap.
> - **PredTrace near-zero** (Table 6): Models at ~100% on PredState collapse to 0–5% on PredTrace.
> - **Formal rules hurt non-reasoning models** (negative Δ_cnd, Table 4).
>
> ### Q4.
> > not fully convinced that the paper is cleanly disentangling
> > structural complexity from bended semantics.
>
> We address this with existing data:
>
> 1. **PredRule controls for structural complexity.** PredRule (§5.2) isolates individual statement evaluation without long-horizon state tracking — loop bodies are replaced with `halt`, eliminating state propagation and thus normalizing code-structural complexity. Even here, KeywordSwap causes significant drops (Table 4, right), showing *semantic shift alone* degrades performance.
>
> 2. **Same programs, different semantics.** Within each split (fixed complexity), we compare Standard vs. KeywordSwap vs. KeywordObf on identical programs. Gemini-2.5-Pro drops threefold under KeywordSwap on Fuzzer-Generated (Table 5), while its KeywordSwap performance *exceeded* No-Semantics on Human-Written — confirming semantic intervention and structural complexity are separable.
>
> 3. **Regression analysis (Appendix F.2.1).** Elastic Net with PLS dimensionality reduction confirms complexity metrics and semantic intervention are significant, separable predictors.
>
> We will make this disentanglement more explicit in the revision.

---

### Official Review · Reviewer_EWww · 2026-03-22

**Soundness:** 3
**Presentation:** 4
**Significance:** 2
**Originality:** 2
**Overall Recommendation:** 3
**Confidence:** 5

**Summary:**

This paper proposes a new benchmark, PLSemanticsBench, which is designed to assess whether LLMs can understand formal semantics of featherweight C programs. Both small-step operational semantics and K semantics are considered and four concrete hypotheses are validated through careful experiment design. The evaluations show that small or non-reasoning models behave much worse than large or reasoning models, and the frontier model like Gemini-2.5 Pro achieves nearly perfect accuracy. The performance drop for K semantics is more significant when semantic mutations present.

**Compliance With Llm Reviewing Policy:**

Affirmed.

**Final Justification:**

Please refer to the Rebuttal Acknowledgement.

**Key Questions For Authors:**

What makes the featherweight C particularly suitable for PLSemanticsBench? Or what are fundamentally missing for other languages?

**Limitations:**

yes

**Strengths And Weaknesses:**

Strengths
- this work concerns an interesting formal reasoning challenge about code, more concretely, the small step operational semantics and K semantics of lightweight C programs
- the writing is very well-done with careful visual illustrations about syntax, semantics, and dataset constructions
- detailed experimental evaluations regarding the four hypotheses are conducted and takeaway messages are clearly summarized

Weaknesses
- although PLSemanticsBench is a new dataset, the empirical findings regarding performance variations in terms of model sizes and reasoning/non-reasoning models are well-expected
- the motivation of assessing whether LLMs understand formal semantics through predicting executing traces or final state is a bit weak, since well-defined computational tasks with structured inputs and outputs can be easily carried out by existing tools. For instance, even large number arithmetics can be challenging for LLMs, however, there is NO need for LLMs to _directly_ solve such kind of tasks. For similar reasons, LLMs don't have to act as parsers, compilers, or runtime interpreters.
- programming languages are vastly diverse, while PLSemanticsBench only considers a simplified dialect of C
- the new dataset is immediately saturated, since even slightly outdated frontier models like Gemini-2.5 Pro achieve nearly perfect accuracy on human-written programs

---

> ### Author Rebuttal · Authors · 2026-03-31
>
> ### Q1.
> > empirical findings regarding performance variations in terms of
> > model sizes and reasoning/non-reasoning models are well-expected
>
> Several findings are non-obvious:
>
> - **KeywordSwap causes far larger drops than KeywordObf (Table 4).** If models were pattern-matching on surface forms, KeywordObf (novel symbols) should be harder. The reverse reveals models actively *misapply* learned operator semantics — with implications for operator overloading, DSLs, and proof assistants (see Q2, point 5).
> - **Formalization-dependent asymmetry.** New rebuttal results (QTih Q1) show Sonnet 4.6 drops to 58.6% under S-KeywordSwap but stays at 99.4% under K-KeywordSwap on the same programs.
> - **CoT fails under KeywordSwap (Table 4).** CoT nearly triples non-reasoning model performance under Standard/KeywordObf but has zero effect under KeywordSwap.
> - **PredTrace near-zero (Table 6).** Models at ~100% on PredState collapse to 0–5% on PredTrace, revealing that final-state accuracy masks inability to sustain step-level rule conditioning.
> - **Formal rules hurt non-reasoning models (negative Δ_cnd, Table 4)** — supplied rules act as confounders for weaker models.
> - **NL→Rule/Rule→NL validation** confirms models understand the notation itself; failures reflect reasoning, not comprehension limitations.
>
> ### Q2.
> > LLMs don't have to act as parsers, compilers, or runtime interpreters.
>
> PLSemanticsBench does not position LLMs as interpreter replacements. Instead, we address: *can LLMs condition reasoning on externally supplied formal rules when those conflict with learned priors?* We motivate in 5 points:
>
> 1. **Execution-aware training improves auxiliary task performance.**
>     - Chen et al. (ICLR'19) boosted synthesis 77%→92%.
>     - NExT (ICML'24) improved program repair.
>     - SemCoder (NeurIPS'24) improved generation.
>     - TRACED (ICSE'24) improved clone/vulnerability detection by 12–25%;
>     - CodeI/O (ICML'25 Spotlight) improved reasoning across symbolic, scientific, logic and math domains.
>
> 2. **The prevailing hypothesis: models internalize semantics when trained on execution traces.** Jin & Rinard (ICML'24) showed formal trace semantics emerge in Transformer hidden states with R²=0.968 correlation to correctness.
>
> 3. **This hypothesis has never been directly tested.** No prior work supplies formal inference rules to frontier models and measures rule-level compliance especially when rules conflict with priors. PLSemanticsBench fills this gap.
>
> 4. **Our results: models rely on surface correlations.** KeywordSwap causes 40–60% drops even at near-perfect standard accuracy; CoT fails under KeywordSwap; PredTrace collapses to near-zero; formal rules hurt non-reasoning models (see Q1 for details).
>
> 5. **Direct practical implications.** Gains from execution-aware training may be brittle when operator meanings differ from training data. This matters:
>     - *Operator overloading* in C++, Python, Scala, Haskell, Julia — e.g., Python's `__add__` can redefine `+` as subtraction, widely used in scientific computing (PyLops, SoftwareX 2020) and DSL development (Rompf & Odersky, GPCE 2010).
>     - *DSLs* with nonstandard semantics.
>     - *Proof assistants* (Lean4, Coq) where LLMs must condition on user-defined nonstandard axioms.
>
> ### Q3.
> > The benchmark is immediately saturated since Gemini-2.5-Pro
> > achieves nearly perfect accuracy.
>
> 1. New model results (see QTih Q1 for full results) show:
>    - GPT-5.4-mini *regresses* vs. GPT-5-mini under S-KeywordSwap (38.3% vs. 78.6%).
>    - Sonnet 4.6 shows formalization-dependent asymmetry (58.6% S-Swap vs. 99.4% K-Swap).
>    - Non-reasoning models perform poorly across the board; CoT helps only under Standard/KeywordObf.
>
> 2. Gemini-2.5-Pro achieves near-perfect accuracy only on PredState/Human-Written/Standard — the easiest combination:
>
> | Setting | Gemini-2.5-Pro |
> |---|---|
> | PredState, Human-Written, Standard (K) | 100% |
> | PredState, Fuzzer-Generated, Standard (K) | 69% |
> | PredState, Fuzzer-Generated, KeywordSwap (K) | 26% |
> | PredTrace, Human-Written, Standard (S) | 32% |
>
> The benchmark has substantial headroom.
>
> ### Q4.
> > What makes featherweight C particularly suitable?
>
> - Explicit block delimiters `{}` avoid conflating indentation-based syntax recovery with semantic reasoning, unlike Python.
> - C-like syntax maximizes pretraining exposure, creating a strong baseline of learned priors against which semantic interventions can be measured.
> - The language supports complex control flow and mutable state while remaining formally tractable.
>
> ### Q5.
> > What are fundamentally missing for other languages?
>
> The framework is language-agnostic. Extension requires formalizing semantics in S or K (K-framework has formalizations for C, Java, Python —  see K9gN Q2), constructing semantic interventions, and generating traces. The challenge is trace scale and fitting formalizations within context windows, not methodology. We will discuss extensions in the revision.

---

> > ### Author Rebuttal · Reviewer_EWww · 2026-04-03
> >
> > I appreciate the careful discussions and updated results. My main concern is not whether the authors have done sufficient analyses of their findings (which I do believe are well-done), but whether the new benchmark indeed matters to the machine learning community, which is the part I am not yet convinced. Random permutations and keyword swaps can be confusing to human programmers. I feel the current contribution may be similar to random SAT instances, which still remain hard for practical solvers to solve but meanwhile matters little for practical SAT solvers to improve. I believe this is a borderline work and would not strongly argue for rejection or acceptance.

---

> > > ### Author Response · Authors · 2026-04-04
> > >
> > > >  I feel the current contribution may be similar to random SAT instances, which still remain hard for practical solvers to solve but meanwhile matters little for practical SAT solvers to improve.
> > >
> > > We thank the reviewer for the thoughtful analogy on random SAT and practical SAT solvers, but the causal mechanism differs: progress on random SAT does not transfer to real SAT because random instances lack the modular structure (variable locality, hierarchical dependencies) present in real instances. Our setting is fundamentally different: the complete inferential structure — every rule, every compositional relationship — is explicitly supplied in the prompt and is preserved identically across all semantic variants. KeywordSwap and KeywordObf change only the mapping between surface symbols and rule meanings; the rules themselves, their dependencies, and their composition are unchanged. Our NL→Rule/Rule→NL validation confirms models understand this structure under all variants. The failure is therefore not due to missing/diluted structure, but the model ignoring supplied specifications in favor of pretrained associations — the same failure mechanism that arises with operator overloading, DSLs, and proof assistants in practice. A closer precedent for our approach is HANS (McCoy et al., ACL 2019), which used synthetic NLI examples and interventions to expose model reliance on lexical heuristics over compositional reasoning. Similarly, Stylized ImageNet (Geirhos et al., ICLR 2019) used synthetic texture-shape conflicts to expose CNN texture bias. In both cases, structurally grounded synthetic interventions revealed real failure mechanisms. PLSemanticsBench follows the same logic for the code domain.
> > >
> > > > but whether the new benchmark indeed matters to the machine learning community
> > >
> > > The failure mechanism we expose — defaulting to priors over supplied specifications — is not specific to programming languages. It is an instance of a general question the ML community cares about: do models faithfully condition on in-context information when it conflicts with pretraining priors? Formal semantics provides a uniquely controlled setting to study this because the rules are unambiguous, atomic, and mechanically verifiable — properties that are difficult to achieve with natural-language. We believe the finding that models systematically fail to override priors even when given complete, unambiguous formal specifications has implications beyond PL, for any deployment setting where models must follow supplied specifications that diverge from their training distribution.

---

### Decision · Program_Chairs · 2026-04-30

**Decision:**

Accept (regular)

**Comment:**

Reviewers agreed that this paper studies an interesting research problem in LLM-guided formal reasoning: how robust are LLM at program semantics reasoning. The paper introduces a methodology for evaluating the capabilities of LLMs to use program formal semantics, to reason about program execution. Through a detailed empirical evaluation, the paper shows that the performance of several state of the art LLMs on program semantics reasoning tasks decrease considerably when mutations or increase in structural complexity occur. This gives empirical evidence that current LLMs performances are based on lexical association rather than on formal rule conditioning. This finding supports the call for strengthening the capabilities of LLMs to reason using specified formal systems. The reviewers and the area chair agree that the results of this paper are good contributions for the ICML community.